# UNITS: A Unified Multi-Task Time Series Model

**Shanghua Gao**
Harvard University
shanghua_gao@hms.harvard.edu

**Teddy Koker**
MIT Lincoln Laboratory
tekoker@mit.edu

**Owen Queen**
Harvard University
owen_queen@hms.harvard.edu

**Thomas Hartvigsen**
University of Virginia
hartvigsen@virginia.edu

**Theodoros Tsiligkaridis**
MIT Lincoln Laboratory
ttsili@ll.mit.edu

**Marinka Zitnik**
Harvard University
marinka@hms.harvard.edu

## Abstract

Although pre-trained transformers and reprogrammed text-based LLMs have shown strong performance on time series tasks, the best-performing architectures vary widely across tasks, with most models narrowly focused on specific areas, such as time series forecasting. Unifying predictive and generative time series tasks within a single model remains challenging. We introduce UNITS, a unified multi-task time series model that utilizes task tokenization to integrate predictive and generative tasks into a single framework. UNITS employs a modified transformer block to capture universal time series representations, enabling transferability from a heterogeneous, multi-domain pre-training dataset—characterized by diverse dynamic patterns, sampling rates, and temporal scales—to a wide range of downstream datasets with varied task specifications and data domains. Tested on 38 datasets across human activity sensors, healthcare, engineering, and finance, UNITS achieves superior performance compared to 12 forecasting models, 20 classification models, 18 anomaly detection models, and 16 imputation models, including adapted text-based LLMs. UNITS also demonstrates strong few-shot and prompt capabilities when applied to new domains and tasks. In single-task settings, UNITS outperforms competitive task-specialized time series models. Code and datasets are available at `https://github.com/mims-harvard/UniTS`.

## 1 Introduction

Foundation models, particularly large language models (LLMs), have transformed deep learning by enabling a single pre-trained model to support multiple tasks, eliminating the need for task-specific models. Language and vision models [9, 101, 92, 50, 32] can be adapted to new tasks with minimal additional training through approaches such as multi-task learning [125], few-shot learning [108, 86], and prompting [66]. Beyond language and vision, there is a growing need for similarly versatile models in time series that can accommodate data from diverse domains—including medicine [34], engineering [102], and science [48]—and support a wide range of tasks, such as forecasting, classification, imputation, and anomaly detection.

Developing multi-task time series models that unify predictive and generative tasks under a single framework remains an open challenge. Time series datasets span multiple domains and exhibit varied temporal scales, sampling rates, and dynamic patterns, making them complex to manage [124, 78].

38th Conference on Neural Information Processing Systems (NeurIPS 2024).

Existing models often fall short in adaptability, as they either struggle to handle samples with varying numbers of variables [112, 67, 14] or treat each variable as independent, overlooking important interdependencies [82]. Time series tasks are also highly diverse, encompassing distinct objectives and specifications across generative and predictive tasks. For example, generative forecasting tasks aim to produce future values within a time series, while predictive tasks may involve making discrete predictions for entire samples. Additionally, task requirements can vary significantly even within the same task type; for instance, generative tasks may involve different forecast lengths, and predictive tasks may feature multiple classification categories. As a result, time series models have mainly remained task-specific, with unique architectures typically designed and trained from scratch for forecasting [67, 82, 119], classification [30, 113], or other specialized tasks [116, 112]. Recent efforts to pre-train unified models [36, 22] or adapt LLMs for time series [118, 12, 129, 47, 97, 100] still heavily depend on extensive fine-tuning or the addition of task- and dataset-specific modules. Some models have explored generative pre-training transformers specifically for time series forecasting [10, 118, 47, 28], reporting strong results but focusing exclusively on forecasting without addressing other types of time series tasks. Consequently, these approaches require users to design and train new modules for each task or limit their application to a single type of tasks. To achieve a versatile, unified time series model—akin to foundational models in vision and language that operate across unified task spaces—a model must accommodate both *generative* and *predictive* tasks. Such a unified model would leverage a single set of weights for multiple tasks, removing the need to develop task-specific models from scratch. This approach would support a broad range of tasks and facilitate rapid adaptation to new datasets.

**Present work.** To address these challenges, we introduce UNITS, a unified multi-task time series model capable of handling a broad spectrum of time series tasks. We rigorously compare UNITS against 12 forecasting methods, 20 classification methods, 18 anomaly detection methods, and 16 imputation methods, including transformer-based, LLM-based, RNN-based, and traditional approaches, to highlight UNITS's generalizability to new tasks. This capability is achieved through the following model design: 1) *Task tokenization:* UNITS encodes task specifications into a unified token representation, enabling universal task specification without post-hoc architectural modifications. 2) *Unified time series architecture:* UNITS processes heterogeneous time series data with varying numbers of variables and sequence lengths without altering its network structure. To accomplish this, UNITS employs self-attention across time and variable dimensions to adapt to diverse temporal dynamics. We introduce a dynamic linear operator to model complex relationships between data points along the time dimension and a module to reduce interference in the feature space of heterogeneous data. 3) *Support for generative and predictive tasks:* The combination of universal task specification and a unified time series architecture allows UNITS to share weights across tasks by co-training on multiple datasets. We use a masked reconstruction pre-training approach, enabling UNITS to be jointly optimized for generative and predictive tasks.

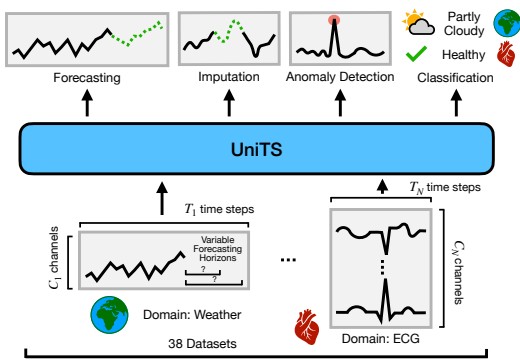

Figure 1: UNITS is a unified multi-task time series model for predictive and generative tasks.

In the single-task setting, where models are trained individually for each dataset, UNITS outperforms task-specialized time series models and repurposed LLMs across forecasting, classification, anomaly detection, and imputation. In a challenging multi-domain, multi-task setting, we find that a single shared-weight UNITS model successfully handles 38 tasks, demonstrating its versatility as a multi-task time series model. UNITS surpasses top baselines that rely on data- and task-specific modules, achieving the highest average performance across tasks and excelling in 27 out of 38 tasks. Additionally, UNITS supports prompt-based learning and direct multi-step forecasting with flexible sequence lengths, capabilities not offered by models using task- and data-specific heads. In direct multi-step forecasting, UNITS outperforms the strongest baseline (which uses a sliding-window approach) by 10.5%. UNITS can also adapt to new tasks through parameter-efficient prompting, achieving results comparable to its fully fine-tuned counterpart. For example, across 20 forecasting datasets, prompted UNITS slightly outperforms the fully fine-tuned model, reducing MAE from 0.381

to 0.376. Furthermore, UNITS demonstrates effective few-shot transfer, successfully addressing tasks like imputation, anomaly detection, and out-of-domain forecasting and classification without requiring specialized modules. For instance, UNITS improves on the strongest baseline by 12.4% in MSE on imputation and 2.3% in F1-score on anomaly detection. UNITS paves the way toward unified time series models, offering strong performance and adaptability across tasks and domains.

## 2   Related Work

**Traditional time series modeling.** Time series analysis has been extensively explored in both the statistics and machine learning communities for many years [45, 103, 123, 18, 80]. Numerous neural architectures have been developed for specific time series tasks such as forecasting [114, 65, 68, 67, 107], classification [115, 71, 70], anomaly detection [25, 56, 16], and imputation [17, 49, 3]. Task-specific models are typically trained via supervised learning on individual datasets, necessitating specialized modules. For example, a classification model requires a classification head with a specific number of classes, while data processing modules must handle a predetermined number of variables. In contrast, UNITS aims to unify various tasks into a universal task specification, enabling the handling of diverse data with a single, unified network architecture. This approach facilitates training a multi-task model capable of addressing multiple time series tasks.

**General time series modeling.** Foundation models, including language models [9, 101] and vision models [62, 50], are trained on broad data at scale to address diverse tasks with no or minimal additional training [8]. Recent studies in time series analysis have sought to develop models with similar capabilities. This includes developing novel architectures to capture diverse time series signals. For instance, TimesNet [112] uses multiple frequency-based features obtained through Fourier transform to capture complex time series signals. There have been several efforts to reprogram LLMs for time series tasks [81, 12, 129, 47, 10]. Models such as GPT4TS [129] and Time-LLM [47] adapt LLMs by fine-tuning their embedding layers or aligning time series samples with LLM-based text prototypes (e.g., GPT-2 [89]). Unlike these models, UNITS is trained exclusively on time series data rather than relying on LLM architectures. Another approach, Lag-Llama [90], pre-trains a model on time series data from multiple domains specifically for forecasting tasks. Similarly, the Moment model [36] is pre-trained on a diverse range of time series data. However, these approaches still require task-specific modules and tuning for each task. In contrast, our UNITS model supports generative and predictive tasks without requiring extensive task-specific model adjustments.

**Prompt learning.** Prompt learning has emerged as an efficient method for task adaptation in large models [55, 88, 121, 13, 42]. Some approaches construct prompts directly in the model's input domain, such as text prompts for LLMs [2]. Other methods involve tuning soft token inputs to frozen language models [58]. In time series, PromptCast [118] and LLMTime [81] convert time series data into prompts for LLMs to facilitate forecasting. TEMPO [10] is another prompt-based approach that uses a learned set of prompts for LLM-based forecasting applications, while GPT4MTS [46] integrates both textual and numerical data to fine-tune LLMs for forecasting. In contrast, UNITS is trained exclusively on time series data, eliminating the need for computationally expensive pre-trained LLMs. Moreover, the universal task tokenization enables a frozen UNITS to adapt to new tasks beyond forecasting, such as classification and imputation. Further discussion of related work can be found in Appendix A.

## 3   Problem Formulation

**Notation.** We are given a set of multi-domain datasets $\mathcal{D} = \{\mathcal{D}_i | i = 1, \ldots, n\}$, where each dataset $\mathcal{D}_i$ can have a varying number of time series samples; samples can be of varying time lengths and have varying numbers of sensors/variables. Each dataset is described as $\mathcal{D}_i = (\mathcal{X}_i, \mathcal{Y}_i)$, where $\mathcal{X}_i$ denotes time series samples and $\mathcal{Y}_i$ specifies a task defined on $\mathcal{X}_i$. Let $\mathcal{X}$ and $\mathcal{Y}$ be collections, defined as $\mathcal{X} = \{\mathcal{X}_i | i = 1, \ldots, n\}$ and $\mathcal{Y} = \{\mathcal{Y}_i | i = 1, \ldots, n\}$, respectively. A time series sample in datasets is denoted as $\mathbf{x} \in \mathbb{R}^{t \times v}$, where $t$ and $v$ are the length of the time series sample and the number of variables, respectively. We use *time dimension* and *variable dimension* to indicate the row and column dimensions in $\mathbf{x}$. $\mathcal{Y}_i$ contains four common time series tasks: forecasting, classification, anomaly detection, and imputation. Further, each task type can be instantiated in numerous ways, e.g., forecasting over different time lengths and classification with varying numbers of classes. We use $F(\mathcal{X}, \theta)$ to denote a multi-task model trained on $\mathcal{X}$. See Table 12 for notation details.

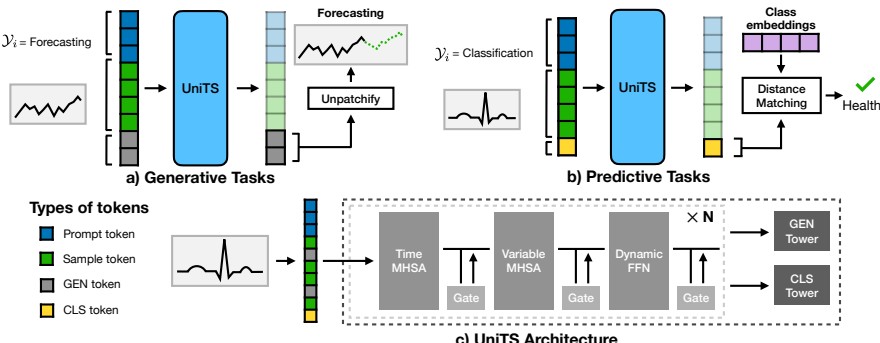

Figure 2: **a**) UNITS for forecasting; input is tokenized, and GEN tokens are un-patchified to infer the forecast horizon. **b)** UNITS for classification; a CLS token is used to represent class information and then compared to class tokens to get prediction class. **c)** Architecture of UNITS model.

**Desiderata for a unified multi-task time series model.** Unlike specialized time series models designed and separately trained for each specific dataset $\mathcal{D}_i$, a unified time series model $F(\mathcal{X}, \theta)$ is a single model with weights $\theta$ that are shared across all types of tasks and satisfies the following three desiderata: 1) *Heterogeneous time series:* To process time series from all sources, the model $F$ must be agnostic with any input samples in $\mathcal{X}$, given the heterogeneity in time series lengths $t$ and variable counts $v$ in time series samples $\mathbf{x}$ from various sources. 2) *Universal task specification:* For easy multi-task support and swift adaption to new tasks, the model $F$ should adopt a universal task specification $F(\mathcal{X}, \theta) \to \mathcal{Y}$ applicable across all type of tasks $\mathcal{Y}$. 3) *One shared model:* Sharing weights $\theta$ across tasks enables the unified model $F$ to handle multiple tasks simultaneously. It contrasts with existing methods that typically train separate models on task-specific datasets, often involving elaborately tuned training parameters.

To realize the above desiderata, UNITS supports multi-task, prompt-based, and few-shot learning. **Multi-task learning**: UNITS specifies a single model $F(\mathcal{X}, \theta) \to \mathcal{Y}$ for tasks $\mathcal{Y}$ defined on datasets $\mathcal{X}$. Multi-task learning showcases the flexibility of the model to learn across time series domains and tasks. **Prompt learning**: By leveraging prompt tokens, UNITS supports prompt learning, *Prompting*$\{F(\mathcal{X}, \theta), \text{token}\} \to \mathcal{Y}$, across tasks while keeping the model frozen. Additionally, UNITS can be trained in a single-task manner, following the same setup as used by many existing models. Other settings are described in Appendix C.1.

# 4 UNITS Model

UNITS is a multi-task model with a unified network architecture. It uses a token-based format to describe tasks and time series from different domains. We introduce a novel approach with three distinct token types: sample, prompt, and task tokens, each serving a unique purpose in time series analysis. The input time series sample is tokenized into sample tokens. Prompt tokens provide essential context for the task, guiding the model to accomplish the user-specified task. Task tokens (GEN and CLS) are combined with other tokens and used for generative and predictive tasks. UNITS then converts task tokens into task predictions to produce the final model output. Unlike transformers such as PatchTST [82], UNITS introduces new token types: sample tokens allow for modeling of multivariate time series, prompt tokens enable efficient multi-task and prompt learning [101], and task tokens unify predictive and generative tasks into one format.

## 4.1 Prompting UNITS with Unified Time Series Data Tokens

We introduce how to use unified tokens to unify different task types and data for inference. Tokens on different network layers have the same shape, so we omit the layer index for simplicity.

**Sample tokens.** We divide time series input sample $\mathbf{x} \in \mathbb{R}^{t \times v}$ into patches along the time dimension using a non-overlapping patch size of $k$. A linear layer projects each patch into an embedding vector of length $d$, obtaining sample tokens $\mathbf{z_x} \in \mathbb{R}^{s \times v \times d}$, where $s = t/k$. Since $v$ and $s$ vary across time

series data domains, we keep the variable and time dimension in tokens. $\mathbf{z_x}$ are then added with learnable positional embeddings.

**Prompt tokens.** Prompt tokens $\mathbf{z}_p \in \mathbb{R}^{p \times v \times d}$ are defined as learnable embeddings, where $p$ is the number of tokens. In a multi-task setting, each dataset has its own set of prompt tokens. These tokens incorporate the specific context related to the data and the task the model needs to complete. For each sample in the dataset, these prompt tokens are appended to the sample tokens and sent to the network to provide context information about the current sample. For prompt learning, with the pre-trained model weights being frozen, UNITS adapts to new tasks by utilizing prompt tokens learned with the prompt tuning. Prompt learning is more efficient than tuning new data/task-specific heads and achieves comparable performance to full model fine-tuning, as shown by few-shot learning experiments on new tasks (Tables 4 and 5) and new datasets (Table 3).

**Task tokens.** In Figure 2ab, we categorize task tokens into two types: 1) GEN (Generation) tokens used in forecasting, imputation, and anomaly detection, and 2) CLS (Classification) tokens, which are used for classification tasks (in a given task, the number of CLS tokens corresponds to the number of classes in the task). Task tokens define a general format for representing tasks and support flexible adaptation to new tasks. For tasks involving forecasting, in Figure 2a, the GEN token $\mathbf{z}_m \in \mathbb{R}^{1 \times v \times d}$, is replicated $f$-times based on desired forecasting length to get $\hat{\mathbf{z}}_m \in \mathbb{R}^{f \times v \times d}$. These tokens $\hat{\mathbf{z}}_m$ are then concatenated with the sample and prompt tokens and fed into the UNITS network:

$$\mathbf{z}_{\text{Fore}} = \text{CA}(\mathbf{z}_p, \mathbf{z_x}, \hat{\mathbf{z}}_m) \in \mathbb{R}^{(p+s+f) \times v \times d}, \tag{1}$$

where CA is the concatenation operation along the time dimension. At the output of the model, embedding vectors with length $d$ in $\hat{\mathbf{z}}_m$ are unpatchified to patches with size $e$ to obtain the forecasting sample $\hat{\mathbf{x}}$, i.e. $\hat{\mathbf{x}} = \text{Proj}(\hat{\mathbf{z}}_m) \in \mathbb{R}^{(f \times e) \times v}$. This approach allows the UNITS model to perform direct multi-step forecasting [99, 76, 119] over arbitrary time lengths, as illustrated in Figure 3. For classification, in Figure 2b, CLS token $\mathbf{z}_c \in \mathbb{R}^{1 \times v \times d}$ is concatenated along the time dimension with the prompt and sample tokens, resulting in:

$$\mathbf{z}_{\text{Pred}} = \text{CA}(\mathbf{z}_p, \mathbf{z_x}, \mathbf{z}_c) \in \mathbb{R}^{(p+s+1) \times v \times d}, \tag{2}$$

which is then fed into the model. We define class embeddings $\mathbf{z}_e \in \mathbb{R}^{e \times v \times d}$ for each of $e$ classes in the task. These class embeddings are either trained or generated by averaging CLS tokens of training samples in each class. Finally, the class for sample $\mathbf{x}$ is predicted by finding the class embedding vector in $\mathbf{z}_e$ that is the closest to the CLS token $\mathbf{z}_c$ from the model output:

$$\text{Class} = \underset{i}{\text{argmin}} \, ||\mathbf{z}_c - \mathbf{z}_{e_i}||^2, i \in [0, e). \tag{3}$$

For imputation, missing values are imputed using the GEN tokens. For anomaly detection, the model takes a time series sample containing any number of potentially anomalous values, generates the output sample by reading out the sample tokens, and then determines anomalous values based on the reconstruction error between the input sample and the generated sample. Details on using tokens for imputation and anomaly detection are in Appendix C.2. All tokens and embeddings are trained to achieve their functions.

## 4.2 Unified Network Architecture in UNITS

Time series samples can have varying numbers of variables, temporal dynamics, and time lengths across different domains and types of tasks. UNITS uses a modified transformer architecture [104] to handle heterogeneous multi-domain data with varying dynamics and the number of variables (Figure 2c). In the following, we describe key modules of UNITS architecture. Note that UNITS can also be used with other backbones, such as Mamba [38].

**Time and variable self-attention.** We use a two-way self-attention to both variable and time dimensions. This approach contrasts with previous methods that apply self-attention to either time [82] or variable dimension [67], but not to both dimensions. Time and variable self-attention effectively handle time series samples with various numbers of variables $v$ and different time lengths $t$.

**DyLinear.** We modify the transformer block by adding a dynamic operator (DyLinear) into the feed-forward network layer (FFN). This modification enables the FFN to capture dependencies between tokens. In contrast to the standard FFN, which processes embedding vectors on a point-wise basis, DyLinear uses weight interpolation to accommodate varying time lengths. Given a sequence of

sample tokens $\mathbf{z}_t \in \mathbb{R}^{l_{\text{in}} \times d}$, DyLinear interpolates weights $\mathbf{w} \in \mathbb{R}^{w_{\text{out}} \times w_{\text{in}}}$ to accommodate varying time lengths as follows:

$$\text{DyLinear}(\mathbf{z}_t, \mathbf{w}) = \mathbf{W}_{\text{Interp}} \mathbf{z}_t \in \mathbb{R}^{l_{\text{out}} \times d}; \mathbf{W}_{\text{Interp}} = \text{Interp}(\mathbf{w}) \in \mathbb{R}^{l_{\text{out}} \times l_{\text{in}}}, \qquad (4)$$

where Interp is a bi-linear interpolation to resize $\mathbf{w}$ from shape $w_{\text{out}} \times w_{\text{in}}$ to $l_{\text{out}} \times l_{\text{in}}$ to match the input and output length. DyLinear captures dependency patterns across time series samples, which leads to improved performance on generative tasks (Table 23).

**Gating module.** We add a gating module after each layer to mitigate interference in the latent representation space caused by multi-domain and multi-task datasets (Figure 2). This module dynamically re-scales features in layer-wise latent spaces and promotes the stability of latent representations.

**Generative and predictive towers.** We design a shared GEN tower ($H_{\text{GEN}}$) and CLS tower ($H_{\text{CLS}}$) for transferring GEN/CLS tokens to generate time series samples and classification classes, as introduced in Section 4.1. Unlike existing works that use standalone, task-specific heads for individual datasets, our approach leverages GEN tower and CLS tower for all generative and predictive tasks, respectively, ensuring a more unified and efficient model architecture.

The UNITS architecture includes the backbone network composed of $N$ modified transformer blocks described above, a CLS tower, and a GEN tower. Implementation details are in Appendix C.3. Ablations in Appendix F verify the effectiveness of this architecture.

## 4.3  UNITS Model Training

**Unified masked reconstruction pre-training.** To enhance UNITS's abilities to 1) learn general features applicable to both generative and predictive tasks and 2) efficiently adapt to downstream tasks via prompt learning, we introduce a unified mask reconstruction pre-training scheme. It leverages the semantics of both prompt and CLS tokens (Section 4.1) for masked reconstruction pre-training, therefore learning representations for both generative and predictive capabilities. This is distinct from pre-training strategies that use either generative [82, 120, 26, 54] or predictive [72, 109, 117, 29, 124, 87] approach. Unlike these approaches that pre-train only the model backbone, our strategy pre-trains all components of UNITS, including the backbone and GEN/CLS towers (Section 4.2), enabling prompt and zero-shot learning over a frozen pre-trained model. For each time-series sample $\mathbf{x}$, a handful of sample tokens get masked and replaced with GEN tokens. These masked sample tokens is then concatenated with prompt tokens and CLS tokens, sent to the UNITS backbone network. In the unified pre-training loss, tokens from the backbone network output are sent to the CLS/GEN towers to reconstruct the input sample $\mathbf{x}$, formulating as follows:

$$L_{\text{pretrain}} = L_{\text{MSE}}(H_{\text{GEN}}(\mathbf{z}_p, \mathbf{z}_\mathbf{x}), \mathbf{x}) + L_{\text{MSE}}(\hat{H}_{\text{GEN}}(H_{\text{CLS}}(\mathbf{z}_{\text{Pred}}), \mathbf{z}_\mathbf{x}), \mathbf{x}). \qquad (5)$$

$L_{\text{MSE}}$ is the MSE loss to predict the full sample $\mathbf{x}$. For the left side of the loss, prompt token $\mathbf{z}_p$ is sent along with sample token $\mathbf{z}_\mathbf{x}$ to GEN tower $H_{\text{GEN}}$ to help with the reconstruction. For the right side of the loss, to leverage the semantics of the CLS token and train the CLS tower $H_{\text{CLS}}$ for predictive tasks, $\mathbf{z}_{\text{Pred}}$ (Eq. 2) from the model output is processed by the CLS tower $H_{\text{CLS}}$ to get classification-related embedding vectors $\hat{\mathbf{z}}_{\text{Pred}} = H_{\text{CLS}}(\mathbf{z}_{\text{Pred}})$, and another GEN tower $\hat{H}_{\text{GEN}}$ takes in $\hat{\mathbf{z}}_{\text{Pred}}$ and $\mathbf{z}_\mathbf{x}$ to predict the full sample. $\hat{H}_{\text{GEN}}$ is only used for pre-training and will be removed for downstream tasks. This unified pre-training strategy involves pre-training both tokens, the backbone network, and the GEN/CLS towers for both generative and predictive abilities.

**Training UNITS models.** We implement and evaluate two UNITS models, each trained in a different regime. We start with a pre-trained UNITS that is optimized using self-supervised $L_{\text{pretrain}}$ in Eq. 5 and trained across a collection of multi-domain datasets. Given a self-supervised pre-trained UNITS whose weights are frozen, we consider a fine-tuned model where only tokens for predictive or generative tasks are fine-tuned (denoted as UNITS-*PMT* in Experiments). We also consider a standard multi-task supervised learning regime, where a single UNITS model is trained from scratch to simultaneously perform many tasks (denoted as UNITS-*SUP* in Experiments). These two regimes use a multi-task setup, where a single model is trained and tested on multiple tasks and datasets. During multi-task training, we sample batches of time series samples and aggregate dataset-centric loss values: $L_{\text{total}} = \sum_{i=1}^{I} \lambda_i L_i(D_i)$, where $L_i$ is the loss of batch $i$, $\lambda_i$ is the weight for each loss, and $I$ denotes the number of batches. We follow [112] and use the MSE loss for forecasting and cross-entropy loss for classification. For fair comparison with models trained in a single-task manner, we follow the experimental setup of [112, 67] and benchmark UNITS in a single-task setting (denoted as UNITS-*ST* in Experiments), where the model is trained separately on each dataset/task.

**Forecasting (36 datasets)**

| Forecasting 36 datasets | UniTS-*ST* (Ours) | | iTransformer [67] | | RLinear [59] | | PatchTST [82] | | Crossformer [126] | | TiDE [21] | | TimesNet [112] | | DLinear [119] | | SCINet [64] | | FEDformer [128] | | Stationary [69] | | Autoformer [114] | |
|---|---|---|---|---|---|---|---|---|---|---|---|---|---|---|---|---|---|---|---|---|---|---|---|---|
| Metric | MSE | MAE | MSE | MAE | MSE | MAE | MSE | MAE | MSE | MAE | MSE | MAE | MSE | MAE | MSE | MAE | MSE | MAE | MSE | MAE | MSE | MAE | MSE | MAE |
| ETTm1 | **0.377** | **0.395** | 0.407 | 0.410 | 0.414 | 0.407 | 0.387 | 0.400 | 0.513 | 0.496 | 0.419 | 0.419 | 0.400 | 0.406 | 0.403 | 0.407 | 0.485 | 0.481 | 0.448 | 0.452 | 0.481 | 0.456 | 0.588 | 0.517 |
| ETTm2 | **0.275** | **0.323** | 0.288 | 0.332 | 0.286 | 0.327 | 0.281 | 0.326 | 0.757 | 0.610 | 0.358 | 0.404 | 0.291 | 0.333 | 0.350 | 0.401 | 0.571 | 0.537 | 0.305 | 0.349 | 0.306 | 0.347 | 0.327 | 0.371 |
| ETTh1 | **0.403** | **0.424** | 0.454 | 0.447 | 0.446 | 0.434 | 0.469 | 0.454 | 0.529 | 0.522 | 0.541 | 0.507 | 0.458 | 0.450 | 0.456 | 0.452 | 0.747 | 0.647 | 0.440 | 0.460 | 0.570 | 0.537 | 0.496 | 0.487 |
| ETTh2 | **0.366** | **0.395** | 0.383 | 0.407 | 0.374 | 0.398 | 0.387 | 0.407 | 0.942 | 0.684 | 0.611 | 0.550 | 0.414 | 0.427 | 0.559 | 0.515 | 0.954 | 0.723 | 0.437 | 0.449 | 0.526 | 0.516 | 0.450 | 0.459 |
| ECL | **0.163** | **0.258** | 0.178 | 0.270 | 0.219 | 0.298 | 0.205 | 0.290 | 0.244 | 0.334 | 0.251 | 0.344 | 0.192 | 0.295 | 0.212 | 0.300 | 0.268 | 0.365 | 0.214 | 0.327 | 0.193 | 0.296 | 0.227 | 0.338 |
| Exchange | **0.297** | **0.376** | 0.360 | 0.403 | 0.378 | 0.417 | 0.367 | 0.404 | 0.940 | 0.707 | 0.370 | 0.413 | 0.416 | 0.443 | 0.354 | 0.414 | 0.750 | 0.626 | 0.519 | 0.429 | 0.461 | 0.454 | 0.613 | 0.539 |
| Traffic | 0.452 | 0.289 | **0.428** | **0.282** | 0.626 | 0.378 | 0.481 | 0.304 | 0.550 | 0.304 | 0.760 | 0.473 | 0.620 | 0.336 | 0.625 | 0.383 | 0.804 | 0.509 | 0.610 | 0.376 | 0.624 | 0.340 | 0.628 | 0.379 |
| Weather | **0.235** | **0.266** | 0.258 | 0.278 | 0.272 | 0.291 | 0.259 | 0.281 | 0.259 | 0.315 | 0.271 | 0.320 | 0.259 | 0.287 | 0.265 | 0.317 | 0.292 | 0.363 | 0.309 | 0.360 | 0.288 | 0.314 | 0.338 | 0.382 |
| Solar-Energy | **0.225** | **0.254** | 0.233 | 0.262 | 0.369 | 0.356 | 0.270 | 0.307 | 0.641 | 0.639 | 0.347 | 0.417 | 0.301 | 0.319 | 0.330 | 0.401 | 0.282 | 0.375 | 0.291 | 0.381 | 0.261 | 0.381 | 0.885 | 0.711 |
| Best Count | 28 | 27 | 4 | 4 | 0 | 1 | 0 | 0 | 0 | 0 | 0 | 0 | 0 | 0 | 0 | 0 | 0 | 0 | 0 | 0 | 0 | 0 | 0 | 0 |

**Classification (10 datasets, Accuracy↑)**

| Classification | | Freq. | MLP | | Transformers | | | | | | | | | TCN | RNN | | | Classic methods | | |
|---|---|---|---|---|---|---|---|---|---|---|---|---|---|---|---|---|---|---|---|---|
| 10 datasets Accuracy↑ | UniTS-*ST* (Ours) | TimesNet [112] | LightTS [122] | DLinear [119] | Flow. [113] | ETS. [111] | FED. [128] | Station. [69] | Auto. [114] | Pyra. [65] | In. [127] | Re. [51] | Trans. [104] | TCN [30] | LSSL [39] | LSTNet [53] | LSTM [41] | Rocket [24] | XGBoost [15] | DTW [6] |
| Avg. | **75.0** | 73.6 | 70.4 | 67.5 | 73.0 | 71.0 | 70.7 | 72.7 | 71.1 | 70.8 | 72.1 | 71.5 | 71.9 | 70.3 | 70.9 | 71.8 | 48.6 | 72.5 | 66.0 | 67.0 |

**Anomaly Det. (F1↑)**

| Anomaly Det. (F1↑) | UniTS-*ST* (Ours) | TimesNet [112] | FED [128] | LightTS [122] | ETS. [111] | DLinear [119] | Station. [69] | LSSL [39] | Auto. [114] | Pyra. [65] | Anomaly [116] | Info. [127] | Refo. [51] | TCN [30] | LogTrans [57] | Trans. [104] | LSTM [41] |
|---|---|---|---|---|---|---|---|---|---|---|---|---|---|---|---|---|---|
| SMD | **88.09** | 84.62 | 85.08 | 82.53 | 83.13 | 77.10 | 84.62 | 71.31 | 85.11 | 83.04 | 85.49 | 81.65 | 75.32 | 81.49 | 76.21 | 79.56 | 71.41 |
| MSL | 83.46 | 81.80 | 78.57 | 78.95 | 85.03 | 84.88 | 77.50 | 82.53 | 79.05 | 84.86 | 83.31 | 84.06 | 84.40 | 78.60 | 79.57 | 78.68 | 81.93 |
| SMAP | **83.80** | 69.50 | 70.76 | 69.21 | 69.50 | 69.26 | 71.09 | 66.90 | 71.12 | 71.09 | 71.18 | 69.92 | 70.40 | 70.45 | 69.97 | 69.70 | 70.48 |
| SWaT | 93.26 | 93.00 | 93.19 | **93.33** | 84.91 | 87.52 | 79.88 | 85.76 | 92.74 | 91.78 | 83.10 | 81.43 | 82.80 | 85.09 | 80.52 | 80.37 | 84.34 |
| PSM | **97.43** | 97.38 | 97.23 | 97.15 | 91.76 | 93.55 | 97.29 | 77.20 | 93.29 | 82.08 | 79.40 | 77.10 | 73.61 | 70.57 | 76.74 | 76.07 | 81.67 |
| Avg. | **89.21** | 85.26 | 84.97 | 84.23 | 82.87 | 82.46 | 82.08 | 76.74 | 84.26 | 82.57 | 80.50 | 78.83 | 77.31 | 77.24 | 76.60 | 76.88 | 77.97 |

**Impu.**

| Impu. | UniTS-*ST* (Ours) | | TimesNet [112] | | ETS. [111] | | LightTS [122] | | DLinear [119] | | FED. [128] | | Station. [69] | | Auto. [114] | | Pyra. [65] | | In. [127] | | LogTrans [57] | | Re. [51] | | LSTM [41] | | TCN [30] | | LSSL [39] | |
|---|---|---|---|---|---|---|---|---|---|---|---|---|---|---|---|---|---|---|---|---|---|---|---|---|---|---|---|---|---|---|
| Metric | MSE | MAE | MSE | MAE | MSE | MAE | MSE | MAE | MSE | MAE | MSE | MAE | MSE | MAE | MSE | MAE | MSE | MAE | MSE | MAE | MSE | MAE | MSE | MAE | MSE | MAE | MSE | MAE | MSE | MAE |
| ETTm1 | **0.019** | **0.087** | 0.027 | 0.107 | 0.120 | 0.253 | 0.104 | 0.218 | 0.093 | 0.206 | 0.062 | 0.177 | 0.036 | 0.126 | 0.051 | 0.150 | 0.717 | 0.570 | 0.071 | 0.188 | 0.050 | 0.154 | 0.055 | 0.166 | 0.989 | 0.786 | 0.516 | 0.497 | 0.113 | 0.254 |
| ETTh1 | **0.043** | **0.136** | 0.078 | 0.187 | 0.202 | 0.329 | 0.284 | 0.373 | 0.201 | 0.306 | 0.117 | 0.246 | 0.094 | 0.201 | 0.103 | 0.214 | 0.842 | 0.682 | 0.161 | 0.279 | 0.219 | 0.332 | 0.122 | 0.245 | 1.225 | 0.873 | 0.621 | 0.571 | 0.424 | 0.481 |
| ECL | **0.038** | **0.124** | 0.092 | 0.210 | 0.214 | 0.339 | 0.131 | 0.262 | 0.132 | 0.260 | 0.130 | 0.259 | 0.100 | 0.218 | 0.101 | 0.225 | 0.297 | 0.382 | 0.222 | 0.328 | 0.175 | 0.303 | 0.200 | 0.313 | 0.277 | 0.365 | 0.582 | 0.597 | 0.222 | 0.293 |
| Weather | **0.026** | **0.045** | 0.030 | 0.054 | 0.076 | 0.171 | 0.055 | 0.117 | 0.052 | 0.110 | 0.099 | 0.203 | 0.032 | 0.059 | 0.031 | 0.057 | 0.152 | 0.235 | 0.045 | 0.104 | 0.039 | 0.076 | 0.038 | 0.087 | 0.365 | 0.434 | 0.183 | 0.291 | 0.045 | 0.108 |
| Best Count | 16 | 16 | 0 | 0 | 0 | 0 | 0 | 0 | 0 | 0 | 0 | 0 | 0 | 0 | 0 | 0 | 0 | 0 | 0 | 0 | 0 | 0 | 0 | 0 | 0 | 0 | 0 | 0 | 0 | 0 |

Table 1: Single-task comparison with existing methods on forecasting, classification, anomaly detection, and imputation tasks where each model is separately trained on each dataset. Full results are shown in Table 30, Table 31, Table 32, and Table 33.

## 5 Experiments

**Datasets.** For multi-task learning on forecasting and classification, we compiled 38 datasets from several sources [79, 33, 82]. These datasets span domains including human activity, healthcare, mechanical sensors, and finance domains and include 20 forecasting tasks of varying forecast lengths ranging from 60 to 720, as well as 18 classification tasks featuring from 2 to 52 categories. Time series samples have varying numbers of readouts (from 24 to 1,152) and sensors (from 1 to 963). Details are in Table 7. When evaluating multi-task few-shot learning on new datasets, a novel dataset collection comprising 6 classification tasks and 9 forecasting tasks (Table 8) is utilized. For multi-task few-shot learning on new tasks, we use the 6 datasets (Table 10) for imputation tasks and 5 datasets (Table 11) for anomaly detection tasks. On the single-task setting, we following existing works [112, 67] to use 36 datasets for forecasting (Table 30), 10 datasets for classification (Table 31), 4 datasets for imputation (Table 10), and 5 datasets for anomaly detection (Table 11).

**Baselines.** We conduct an extensive comparison between UNITS and 12 time series forecasting methods, 20 classification methods, 18 anomaly detection methods, and 16 imputation methods, as listed in Table 13. For comparison on the challenging multi-task setting, we excluded methods that overly rely on task-specific modules and lack a shared backbone, and we select 6 strong time series methods: iTransformer [67], TimesNet [82], PatchTST [82], Pyraformer [65], Autoformer [114], and the LLM-reprogrammed method GPT4TS [129]. Many of these methods are designed and evaluated only for one type of tasks, e.g., GPT4TS and iTransformer are forecasting models. To include these methods in our benchmarking, when necessary, we add task-specific input/output modules to support multiple tasks. Training and evaluation details are shown in Appendix D.2.

### 5.1 Benchmarking UNITS on Single-Task Learning

**Setup.** For fair comparisons with baseline methods, we benchmark single-task UNITS-*ST* on forecasting, classification, anomaly detection, and imputation. Models are separately trained from scratch with configuration tailored to datasets. Details are in Appendix K.

**MULTI-TASK CLASSIFICATION (ACCURACY↑)** — left part: **MULTI-TASK FORECAST**

| Multi-task Forecast | UniTS-SUP MSE↓ | MAE↓ | UniTS-PMT MSE↓ | MAE↓ | iTrans. MSE↓ | MAE↓ | TimesNet MSE↓ | MAE↓ | PatchTST MSE↓ | MAE↓ | Pyraformer MSE↓ | MAE↓ | Autoformer MSE↓ | MAE↓ | GPT4TS MSE↓ | MAE↓ |
|---|---|---|---|---|---|---|---|---|---|---|---|---|---|---|---|---|
| NN5$_{P112}$ | .611 | .549 | .622 | .546 | .623 | .554 | .629 | .541 | .634 | .568 | 1.07 | .791 | 1.23 | .903 | .623 | .545 |
| ECL$_{P96}$ | .167 | .271 | .157 | .258 | .204 | .288 | .184 | .289 | .212 | .299 | .390 | .456 | .262 | .364 | .198 | .285 |
| ECL$_{P192}$ | .181 | .282 | .173 | .272 | .208 | .294 | .204 | .307 | .213 | .303 | .403 | .463 | .34 | .421 | .200 | .288 |
| ECL$_{P336}$ | .197 | .296 | .185 | .284 | .224 | .310 | .217 | .320 | .228 | .317 | .417 | .466 | .624 | .608 | .214 | .302 |
| ECL$_{P720}$ | .231 | .324 | .219 | .314 | .265 | .341 | .284 | .363 | .270 | .348 | .439 | .483 | .758 | .687 | .254 | .333 |
| ETTh1$_{P96}$ | .386 | .409 | .390 | .411 | .382 | .399 | .478 | .448 | .389 | .400 | .867 | .702 | .505 | .479 | .396 | .413 |
| ETTh1$_{P192}$ | .429 | .436 | .432 | .438 | .431 | .426 | .561 | .504 | .440 | .43 | .931 | .751 | .823 | .601 | .458 | .448 |
| ETTh1$_{P336}$ | .466 | .457 | .480 | .460 | .476 | .449 | .612 | .537 | .482 | .453 | .96 | .763 | .731 | .580 | .508 | .472 |
| ETTh1$_{P720}$ | .494 | .483 | .542 | .508 | .495 | .487 | .601 | .541 | .486 | .479 | .994 | .782 | .699 | .590 | .546 | .503 |
| Exc.$_{P192}$ | .243 | .351 | .200 | .320 | .175 | .297 | .259 | .370 | .178 | .301 | 1.22 | .916 | .306 | .409 | .177 | .300 |
| Exc.$_{P336}$ | .431 | .476 | .346 | .425 | .322 | .409 | .478 | .501 | .328 | .415 | 1.22 | .917 | .462 | .508 | .326 | .414 |
| ILI$_{P60}$ | 1.99 | .878 | 2.372 | .945 | 1.99 | .905 | 2.367 | .966 | 2.307 | .970 | 4.791 | 1.46 | 3.812 | 1.33 | 1.90 | .868 |
| Traf.$_{P96}$ | .47 | .318 | .465 | .298 | .606 | .389 | .611 | .336 | .643 | .405 | .845 | .465 | .744 | .452 | .524 | .351 |
| Traf.$_{P192}$ | .485 | .323 | .484 | .306 | .592 | .382 | .643 | .352 | .603 | .387 | .883 | .477 | 1.09 | .638 | .519 | .346 |
| Traf.$_{P336}$ | .497 | .325 | .494 | .312 | .600 | .384 | .662 | .363 | .612 | .389 | .907 | .488 | 1.19 | .692 | .530 | .350 |
| Traf.$_{P720}$ | .53 | .34 | .534 | .335 | .633 | .401 | .678 | .365 | .652 | .406 | .974 | .522 | 1.34 | .761 | .562 | .366 |
| Wea.$_{P96}$ | .158 | .208 | .157 | .206 | .193 | .232 | .169 | .220 | .194 | .233 | .239 | .323 | .251 | .315 | .182 | .222 |
| Wea.$_{P192}$ | .207 | .253 | .208 | .251 | .238 | .269 | .223 | .264 | .238 | .268 | .323 | .399 | .289 | .335 | .228 | .261 |
| Wea.$_{P336}$ | .264 | .294 | .264 | .291 | .291 | .306 | .279 | .302 | .290 | .304 | .333 | .386 | .329 | .356 | .282 | .299 |
| Wea.$_{P720}$ | .341 | .344 | .344 | .344 | .365 | .354 | .359 | .355 | .363 | .35 | .424 | .447 | .39 | .387 | .359 | .349 |
| Best Count | 8/20 | 2/20 | 9/20 | 12/20 | 3/20 | 5/20 | 0/20 | 1/20 | 1/20 | 1/20 | 0/20 | 0/20 | 0/20 | 0/20 | 1/20 | 1/20 |
| Average | .439 | .381 | .453 | .376 | .466 | .394 | .525 | .412 | .488 | .401 | .931 | .623 | .809 | .571 | .449 | .386 |
| Shared | ✓ | ✓ | ✓ | ✓ | × | × | × | × | × | × | × | × | × | × | × | × |

**MULTI-TASK CLASSIFICATION (ACCURACY↑)**

| Class./Num. | UniTS-SUP | UniTS-PMT | iTra.[67] | Tim.[82] | Pat.[82] | Pyra.[65] | Aut.[114] | GPT[129] |
|---|---|---|---|---|---|---|---|---|
| 2/7 | 73.1 | 73.1 | 72.4 | 73.0 | 70.8 | 61.5 | 66.2 | 73.1 |
| 3/1 | 79.7 | 81.4 | 79.4 | 78.0 | 79.2 | 81.4 | 69.9 | 79.4 |
| 4/1 | 96.0 | 99.0 | 79.0 | 91.0 | 77.0 | 74.0 | 60.0 | 96.0 |
| 5/1 | 92.8 | 92.4 | 93.3 | 92.6 | 94.3 | 91.4 | 91.9 | 93.0 |
| 6/1 | 95.1 | 95.8 | 93.6 | 90.6 | 75.8 | 88.7 | 30.2 | 96.2 |
| 7/2 | 72.7 | 72.6 | 70.2 | 63.5 | 71.6 | 74.3 | 67.7 | 71.1 |
| 8/1 | 82.2 | 85.3 | 82.2 | 84.4 | 81.9 | 72.2 | 42.2 | 81.9 |
| 9/1 | 92.2 | 90.3 | 95.9 | 97.6 | 94.1 | 85.4 | 94.1 | 94.6 |
| 10/2 | 92.2 | 89.7 | 93.5 | 97.2 | 88.9 | 72.2 | 86.1 | 95.8 |
| 52/1 | 89.6 | 80.8 | 88.2 | 88.9 | 86.5 | 21.4 | 21.7 | 89.7 |
| Best | 3/18 | 7/18 | 0/18 | 4/18 | 3/18 | 4/18 | 0/18 | 2/18 |
| Avg. | 81.6 | 81.2 | 80.3 | 80.9 | 78.1 | 68.8 | 65.6 | 82.0 |
| Shared | ✓ | ✓ | × | × | × | × | × | × |

Table 2: Multi-task benchmarking across 20 forecasting tasks and 18 classification tasks. Both UNiTS-*SUP* and UNiTS-*PMT* process all 38 tasks using a single model. GPT4TS reprograms a pre-trained LLM (GPT-2) to time series and has dataset/task-specific modules, thus, it is excluded from best count evaluations to ensure fair comparisons.

"$_P$" is forecasting length. "Class./Num." denotes the "number of classes in each task"/"number of datasets".

**Results.** Table 1 shows the single-task performance for four types of tasks. On forecasting tasks with forecasting lengths of 92, 196, 336, and 720, compared with 11 forecasting methods, UNiTS-*ST* achieves the best results on 28 out of 32 datasets for MSE and 27 out of 32 for MAE, surpassing the previous best method, iTransformer, by a clear margin. In Table 34, we demonstrate that UNiTS-*ST* outperforms the concurrent MOMENT [36] model, which was trained on a large and diverse collection of time series data. Additionally, UNiTS-*ST* achieves stronger performance than LLM-reprogrammed methods that are pre-trained with extensive natural language data, e.g. GPT4TS [129], TEST [97], LLM4TS [12], and TEMPO [10]. On 10 classification datasets, UNiTS-*ST* outperforms 19 classification methods on the average accuracy, such as the transformer/MLP/frequency-based methods. It has a gain of 1.4% compared to the previous best TimesNet model. On 5 anomaly detection datasets, UNiTS-*ST* has a clear gain of 3.95% in F1 score compared to the TimesNet and also beat other 15 anomaly detection methods, such as Anomaly Transformer [116]. On 16 imputation datasets with a mask ratio of 12.5%, 25%, 37.5%, UNiTS-*ST* has the best results on all datasets in terms of MSE and MAE, outperforming 14 baseline methods. UNiTS-*ST* has the SoTA performance on these single-task benchmarks, showing its effectiveness.

## 5.2 Benchmarking UNiTS for Multi-Task Learning

**Setup.** In a multi-task setting, we benchmark a single UNiTS model co-trained and evaluated on 38 datasets, comprising 20 forecasting tasks and 18 classification tasks, with variations in the number of variables/sensors, classification classes, and forecasting lengths. We consider two variants of UNiTS; the fully supervised UNiTS-*SUP* and the more challenging UNiTS-*PMT* with prompting, as introduced in Section 4.3. Baselines use the same fully supervised multi-task training as our approach but cannot handle differences across data types and task specifications with a single model. To benchmark them, a shared backbone is used for all tasks, augmented by data-specific input modules and task-specific output modules.

**Results: Model benchmarking.** Table 2 shows multi-task learning performance. UNiTS consistently outperforms baseline methods, achieving the best results in 17 out of 20 forecasting tasks (MSE) and 10 out of 18 classification tasks (accuracy). Performance gains are especially remarkable because UNiTS has one fully shared model, whereas all existing methods require task or dataset-specific modules. We find that baseline methods encounter difficulties performing well across different types of tasks. For example, TimesNet, which excels in classification tasks, underperforms in forecasting tasks. Conversely, iTransformer, the top-performing forecaster, struggles with classification tasks. In contrast, the UNiTS model exhibits robust performance across classification and forecasting. On forecasting, UNiTS-*SUP* surpasses the leading baseline, iTransformer, by 5.8% (0.439 vs. 0.466) in MSE and 3.3% (0.381 vs. 0.394) in MAE. On classification, UNiTS-*SUP* has an average gain of 0.7% accuracy (81.6% vs. 80.9%) over the strongest baseline (TimesNet). UNiTS shows promising potential to unify data and task diversity across time series domains.

Recent research has adapted pre-trained LLMs to time series [47, 12, 129, 37]. Most approaches [47, 12, 129], such as GPT4TS, incorporate additional task-specific modules to align the modalities of time series and natural language. We compare UNiTS with GPT4TS that reprograms pre-trained GPT-2 model [89]. Despite the substantial data amount and model scale gap, e.g., GPT4TS is $48\times$ larger than UNiTS-*SUP* (164.5M vs. 3.4M), UNiTS-*SUP* still compares favorably to GPT4TS. On forecasting tasks, UNiTS-*SUP* even outperforms GPT4TS by 2.2% (0.439 vs. 0.449; MSE).

**Results: Prompting is competitive with supervised training.** Using tokens to prompt a frozen UNiTS, the SSL-pre-trained UNiTS achieves performance comparable to its fully supervised counterpart (Table 2). UNiTS-*PMT* even outperforms the supervised model in forecasting, with a lower MAE score (0.379 vs. 0.381), highlighting the effectiveness of prompt learning in UNiTS. Furthermore, prompt learning with UNiTS surpasses the performance of supervised baseline methods with separate modules. This indicates that the SSL-pre-trained model captures valuable time series representations and that prompt learning allows the model to efficiently adapt to target tasks.

## 5.3  UNiTS for Direct Multi-Step Forecasting

**Setup.** Direct multi-step forecasting predicts across varying time horizons by adjusting from the original trained length, with offsets ranging from 0 to 384. We use 14 out of 20 forecasting datasets with varying lengths. UNiTS achieves this flexibility by repeating the `GEN` token, as described in Section 4.1, a capability not supported by existing methods. For comparison with baseline models, we implement a sliding-window approach for forecasting. In this method, predictions are made over a fixed window size, which then shifts forward incrementally to cover progressively extended time horizons. This sliding mechanism allows us to adapt the model to forecast over new, unseen time periods while maintaining consistency with the evaluation setup used by baseline methods.

**Results: Direct multi-step inference outperforms sliding window approach.** In Figure 3, UNiTS demonstrates improved performance over baseline models across various forecasting lengths when using the sliding-window approach. For example, in the longest forecasting extension of +384, UNiTS outperforms the iTransformer by 8.7% in MSE, achieving a score of 0.451 compared to 0.494. When using direct multi-step inference, UNiTS gains an even larger advantage over the iTransformer, reducing MSE by 10.5% (0.442 vs. 0.494). This approach also reduces the average number of inference steps from 3.66 to 1, resulting in a 3× speedup.

## 5.4  UNiTS for Few-Shot Learning on New Datasets and Tasks

For transfer learning on new tasks and datasets, we load the model weights pre-trained on 38 datasets and apply them in a multi-task setting. We evaluate two approaches: the fully fine-tuned UNiTS-*FT* model and the prompted UNiTS-*PMT* model, in which task-specific tokens are trained.

**Setup: Few-shot classification and forecasting.** Pre-trained models, undergo fine-tuning using 5%, 15%, and 20% of the 11 training set shown in Table 8. Average performance is reported.

**Results.** UNiTS achieves superior performance compared to iTransformer across all training data ratios (Table 3). At the 20% data ratio, UNiTS-*FT* achieves a gain of 8.8% in classification accuracy and a reduction of 5.7% in forecasting MSE. UNiTS-*PMT* surpasses the fully supervised iTrans-

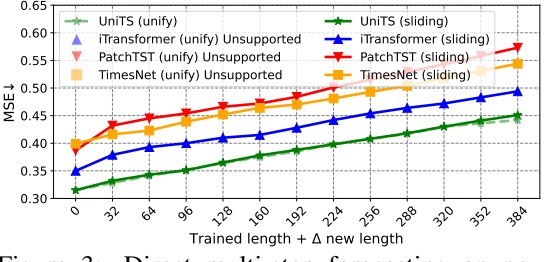

Figure 3: Direct multi-step forecasting on new lengths. UNiTS achieves any new forecasting length with unified direct multi-step inference. Baseline methods use the sliding windows inference as they do not support direct multi-step inference.

Table 3: Few-shot multi-task learning on 9 forecasting and 6 classification tasks on out-of-domain datasets. Ratio is the data ratio of the dataset used for training. Full results in Table 29.

| Model | Ratio | Acc↑ | MSE↓ | MAE↓ | Best Count | Shared |
|---|---|---|---|---|---|---|
| iTransformer-*FT* | 5% | 56.4 | 0.598 | 0.487 | 1/24 | ✗ |
| UNiTS-*PMT* | 5% | 55.7 | **0.508** | **0.440** | 16/24 | ✓ |
| UNiTS-*FT* | 5% | **57.4** | 0.530 | 0.448 | 7/24 | ✓ |
| iTransformer-*FT* | 15% | 56.5 | 0.524 | 0.447 | 4/24 | ✗ |
| UNiTS-*PMT* | 15% | 59.5 | 0.496 | 0.435 | 4/24 | ✓ |
| UNiTS-*FT* | 15% | **61.8** | **0.487** | **0.428** | 16/24 | ✓ |
| iTransformer-*FT* | 20% | 59.9 | 0.510 | 0.438 | 4/24 | ✗ |
| UNiTS-*PMT* | 20% | 63.6 | 0.494 | 0.435 | 3/24 | ✓ |
| UNiTS-*FT* | 20% | **65.2** | **0.481** | **0.425** | 17/24 | ✓ |

Table 4: Few-shot multi-task learning for block-wise imputation on 6 datasets. Full results are in Table 28.

| Impu. (MSE) | Ratio | ECL | ETTh1 | ETTh2 | ETTm1 | ETTm2 | Weather | Avg | Best | Shared |
|---|---|---|---|---|---|---|---|---|---|---|
| TimesNet-*FT* | 25% | 0.245 | 0.369 | 0.193 | 0.442 | 0.119 | 0.106 | 0.246 | 0/6 | × |
| | 50% | 0.258 | 0.412 | 0.211 | 0.607 | 0.140 | 0.125 | 0.292 | 0/6 | × |
| PatchTST-*FT* | 25% | 0.195 | 0.315 | 0.147 | 0.309 | 0.092 | 0.089 | 0.191 | 0/6 | × |
| | 50% | 0.230 | 0.353 | 0.175 | 0.442 | 0.111 | 0.105 | 0.236 | 0/6 | × |
| iTrans-*FT* | 25% | 0.174 | 0.301 | 0.185 | 0.254 | 0.113 | 0.087 | 0.186 | 0/6 | × |
| | 50% | 0.203 | 0.332 | 0.205 | 0.372 | 0.136 | 0.106 | 0.226 | 0/6 | × |
| UNITS-*PMT* | 25% | **0.117** | 0.281 | **0.177** | 0.247 | 0.095 | 0.075 | 0.165 | 2/6 | ✓ |
| | 50% | **0.135** | 0.323 | **0.246** | 0.343 | 0.131 | **0.093** | 0.212 | 3/6 | ✓ |
| UNITS-*FT* | 25% | 0.143 | **0.277** | 0.194 | **0.204** | **0.088** | **0.074** | **0.163** | 4/6 | ✓ |
| | 50% | 0.161 | **0.313** | 0.252 | **0.295** | **0.119** | 0.096 | **0.206** | 3/6 | ✓ |

Table 5: Few-shot multi-task learning on anomaly detection tasks on 5 datasets.

| Anomaly (F1↑) | MSL | PSM | SMAP | SMD | SWAT | Avg | Best | Shared |
|---|---|---|---|---|---|---|---|---|
| Anomaly Trans. | 78.0 | 90.2 | 68.3 | 77.8 | 81.5 | 79.2 | 0/5 | × |
| TimesNet-*FT* | 33.9 | 91.0 | 68.5 | 84.0 | **93.4** | 74.2 | 1/5 | × |
| iTransformer-*FT* | 80.4 | 96.5 | 67.2 | 82.4 | 89.0 | 83.1 | 0/5 | × |
| PatchTST-*FT* | 79.9 | 96.6 | 68.7 | 83.8 | 92.6 | 84.3 | 0/5 | × |
| UNITS-*PMT* | 75.4 | 95.5 | 65.8 | 82.3 | 92.5 | 82.3 | 0/5 | ✓ |
| UNITS-*FT* | **81.2** | **97.3** | **76.0** | **84.7** | 92.5 | **86.3** | 4/5 | ✓ |

former, leading to 6.2% increase in classification accuracy and 3.1% decrease in forecasting MSE. When trained under a 5% data ratio, UNITS-*PMT* exceeds UNITS-*FT* performance for forecasting, suggesting that prompt learning is effective for transfer learning when training data is scarce.

**Setup: Few-shot imputation.** Models are fine-tuned with 10% of 6 imputation training data listed in Table 10, asked to impute 25% and 50% of missing data points.

**Results.** A unified UNITS-*FT* outperforms models that use separate task-specific modules (Table 4), indicating that UNITS has robust few-shot imputation performance. Specifically, on a 25% masking ratio, UNITS-*FT* exceeds the top-performing baseline iTransformer by 12.4% in MSE and 7.9% in MAE. The margin remains notable at a 50% masking ratio, where UNITS-*FT* surpasses iTransformer by 8.8% in MSE and 6.8% in MAE. UNITS-*PMT*, the fixed model with appropriate prompt tokens, outperforms all baseline methods and achieves results comparable to its fully fine-tuned counterpart, suggesting that prompting can adapt UNITS for imputation.

**Setup: Few-shot anomaly detection.** The pre-trained models have been fine-tuned using 5% of five training datasets as listed in Table 10. The average F1-score is used as the metric.

**Results.** UNITS outperforms the top-performing baseline (PathTST) across all metrics (Table 5). UNITS-*FT* achieves an F1-score of 86.3 compared to PathTST's F1-score of 84.3. UNITS-*PMT* also outperforms specialized models (Anomaly Transformer) trained from scratch.

**Additional results and ablations.** Zero-shot learning is significantly more challenging than few-shot learning. Our work primarily focuses on few-shot learning, with some initial exploration of zero-shot learning for forecasting tasks of UniTS on new datasets in Appendix G. Additional analysis and ablation results are in Appendix F and Appendix E.

# 6 Conclusion

We have developed UNITS, a unified model for time series that uses a universal specification of time series tasks. UNITS handles multi-domain time series data with heterogeneous representations, outperforming task-specific models and reprogrammed LLMs on 38 multi-domain and multi-task datasets. UNITS also shows strong few-shot and prompt-based performance and can generalize to new domains and tasks. The unified token scheme in UNITS allows it to represent data and tasks in a general manner. UNITS uses a transformer architecture, and we plan to explore other types of backbones, such MLP-based blocks [107, 14] and Mamba [38], to further enhance UNITS. Limitations and future directions are discussed in Appendix M.

# Acknowledgments

S.G., O.Q., and M.Z. gratefully acknowledge the support of NIH R01-HD108794, NSF CAREER 2339524, US DoD FA8702-15-D-0001, awards from Harvard Data Science Initiative, Amazon Faculty Research, Google Research Scholar Program, AstraZeneca Research, Roche Alliance with Distinguished Scientists, Sanofi iDEA-iTECH, Pfizer Research, Chan Zuckerberg Initiative, John and Virginia Kaneb Fellowship at Harvard Medical School, Biswas Computational Biology Initiative in partnership with the Milken Institute, Harvard Medical School Dean's Innovation Fund for the Use of Artificial Intelligence, and Kempner Institute for the Study of Natural and Artificial Intelligence at Harvard University. T.H. acknowledges the support of the National Security Data & Policy Institute, Contracting Activity 2024-24070100001. Any opinions, findings, conclusions or recommendations expressed in this material are those of the authors and do not necessarily reflect the views of the funders.

DISTRIBUTION STATEMENT: Approved for public release. Distribution is unlimited. This material is based upon work supported by the Under Secretary of Defense for Research and Engineering under Air Force Contract No. FA8702-15-D-0001. Any opinions, findings, conclusions or recommendations expressed in this material are those of the author(s) and do not necessarily reflect the views of the Under Secretary of Defense for Research and Engineering.

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

# A  Extended Related Work

**Comparison of the abilities required by a unified time series model.** We evaluate whether existing works in time series possess the necessary capabilities for constructing a unified time series model, as outlined in Table 6. Most methods fail to support these requirements. For instance, PatchTST [82] processes each variable independently, enabling it to handle multi-domain time series datasets without the need for data-specific heads. However, it still requires task-specific heads for tasks like making forecasts over a fixed length or performing classifications within a predetermined number of classes.

Table 6: Key features of a unified multi-task time series model include the capability to handle heterogeneous time series samples with different numbers of variables and time lengths. Additionally, it should support both generative and predictive time series tasks within the same model.

| Method | Multi-domain time series | Universal task specification | One model |
|---|---|---|---|
| TimesNet [112] | ✗ | ✗ | ✗ |
| PatchTST [82] | ✓ | ✗ | ✗ |
| iTransformer [67] | ✗ | ✗ | ✗ |
| Dlinear [119] | ✗ | ✗ | ✗ |
| FEDFormer [128] | ✗ | ✗ | ✗ |
| MICN [106] | ✗ | ✗ | ✗ |
| Pyraformer [65] | ✗ | ✗ | ✗ |
| Autoformer [114] | ✗ | ✗ | ✗ |
| **UNITS** | ✓ | ✓ | ✓ |

# B  Datasets

**Dataset details.** We introduce the details of the multi-task dataset collection used by our work in Table 7. The dataset collection used for few-shot learning on classification and forecasting are listed in Table 8, the collection used for zero-shot forecasting are listed in Table 9, the collection used for imputation is listed in Table 10, and the collection used for anomaly detection is listed in Table 11. Datasets were aggregated from the Monash Forecasting Repository [33], Time Series Classification Website [79], and Time Series Library [112]. The combined training set consists of over 35 million timesteps and over 6,000 variables. For subsets of a dataset such as ETTh1, we start by splitting the data into training and testing sets based on distinct time intervals of a long time series sequence, following splits in [112]. Within these training and testing intervals, we generate samples using various sliding windows, ensuring that there is no data leakage between the training and testing sets.

**Dataset for direct multi-step forecasting on new forecasting lengths.** For evaluating zero-shot learning capabilities over new forecasting lengths, we initially consider 20 forecasting datasets utilized in the multi-task setting, as detailed in Table 7. However, to adapt to 384 additional forecasting lengths that the model was not trained on, we exclude specific datasets that are incompatible with this requirement. These datasets include $NN5_{P112}$, $ECL_{P720}$, $ETTh1_{P720}$, $ILI_{P60}$, $Traffic_{P720}$, and $Weather_{P720}$. Consequently, our analysis is conducted using 14 remaining forecasting datasets.

# C  Further information on UNITS

## C.1  All learning settings supported by UNITS

UNITS incorporates multi-task, prompt, few-shot, and zero-shot learning, as well as the single-task learning same to existing methods. We introduce the multi-task and prompt learning in the manuscript, here we introduce the other settings supported by UNITS.

**Notations for zero-shot/few-shot learning.** $\hat{\mathcal{X}}$ is an out-of-domain dataset collection not included in $\mathcal{X}$, and $\hat{\mathcal{Y}}$ is used to denote a new type of tasks not contained in $\mathcal{Y}$.

**Zero-shot learning.** UNITS has zero-shot learning ability where model $F(\mathcal{X}, \theta)$ trained on all datasets in $\mathcal{D}$ is tested on multiple types of new tasks that are not trained for, i.e. $F(\mathcal{X}, \theta) \rightarrow \hat{\mathcal{X}}, \hat{\mathcal{X}} \notin \mathcal{X}$. New zero-shot learning tasks include direct multi-step forecasting with a new length and forecasting on out-of-domain datasets with a new number of variables. Zero-shot learning shows the adaptability of UNITS to different time series tasks.

Table 7: Multi-task datasets for classification and forecasting. Prediction length or number of classes are indicated in parenthesis for Forecast and Classification respectively.

| Name | Train Size | Sequence Length | Variables | Task | Class |
|---|---|---|---|---|---|
| NN5$_{P112}$ [98] | 409 | 112 | 111 | Forecast (112) | Finance |
| ECL$_{P96}$ [102] | 18221 | 96 | 321 | Forecast (96) | Electricity |
| ECL$_{P192}$ [102] | 18125 | 96 | 321 | Forecast (192) | Electricity |
| ECL$_{P336}$ [102] | 17981 | 96 | 321 | Forecast (336) | Electricity |
| ECL$_{P720}$ [102] | 17597 | 96 | 321 | Forecast (720) | Electricity |
| ETTh1$_{P96}$ [127] | 8449 | 96 | 7 | Forecast (96) | Electricity |
| ETTh1$_{P192}$ [127] | 8353 | 96 | 7 | Forecast (192) | Electricity |
| ETTh1$_{P336}$ [127] | 8209 | 96 | 7 | Forecast (336) | Electricity |
| ETTh1$_{P720}$ [127] | 7825 | 96 | 7 | Forecast (720) | Electricity |
| Exchange$_{P192}$ [53] | 5024 | 96 | 8 | Forecast (192) | Finance |
| Exchange$_{P336}$ [53] | 4880 | 96 | 8 | Forecast (336) | Finance |
| ILI$_{P60}$ [11] | 581 | 36 | 7 | Forecast (60) | Illness |
| Traffic$_{P96}$ [85] | 12089 | 96 | 862 | Forecast (96) | Traffic |
| Traffic$_{P192}$ [85] | 11993 | 96 | 862 | Forecast (192) | Traffic |
| Traffic$_{P336}$ [85] | 11849 | 96 | 862 | Forecast (336) | Traffic |
| Traffic$_{P720}$ [85] | 11465 | 96 | 862 | Forecast (720) | Traffic |
| Weather$_{P96}$ [110] | 36696 | 96 | 21 | Forecast (96) | Weather |
| Weather$_{P192}$ [110] | 36600 | 96 | 21 | Forecast (192) | Weather |
| Weather$_{P336}$ [110] | 36456 | 96 | 21 | Forecast (336) | Weather |
| Weather$_{P720}$ [110] | 36072 | 96 | 21 | Forecast (720) | Weather |
| SharePriceIncrease [79] | 965 | 60 | 1 | Classification (2) | Finance |
| JapaneseVowels [52] | 270 | 29 | 12 | Classification (9) | Audio |
| SpokenArabicDigits [5] | 6599 | 93 | 13 | Classification (10) | Audio |
| Heartbeat [61] | 204 | 405 | 61 | Classification (2) | Audio |
| ECG5000 [35] | 500 | 140 | 1 | Classification (5) | ECG |
| NonInvasiveFetalECGThorax1 [95] | 1800 | 750 | 1 | Classification (52) | ECG |
| Blink [19] | 500 | 510 | 4 | Classification (2) | EEG |
| FaceDetection [40] | 5890 | 62 | 144 | Classification (2) | EEG |
| SelfRegulationSCP2 [7] | 200 | 1152 | 7 | Classification (2) | EEG |
| ElectricDevices [60] | 8926 | 96 | 1 | Classification (7) | Sensors |
| Trace [93] | 100 | 275 | 1 | Classification (4) | Sensors |
| FordB [23] | 3636 | 500 | 1 | Classification (2) | Sensors |
| MotionSenseHAR [75] | 966 | 200 | 12 | Classification (6) | Human Activity |
| EMOPain [27] | 968 | 180 | 30 | Classification (3) | Human Activity |
| UWaveGestureLibrary [63] | 120 | 315 | 3 | Classification (8) | Human Activity |
| Chinatown [23] | 20 | 24 | 1 | Classification (2) | Traffic |
| MelbournePedestrian [23] | 1194 | 24 | 1 | Classification (10) | Traffic |
| PEMS-SF [20] | 267 | 144 | 963 | Classification (7) | Traffic |

Table 8: Datasets for few-shot learning on classification and forecasting tasks. Prediction length or number of classes are indicated in parenthesis for Forecast and Classification respectively.

| Name | Train Size | Sequence Length | Variables | Task | Class |
|---|---|---|---|---|---|
| ECG200 [84] | 100 | 96 | 1 | Classification (2) | ECG |
| SelfRegulationSCP1 [7] | 268 | 896 | 6 | Classification (2) | EEG |
| RacketSports [4] | 151 | 30 | 6 | Classification (4) | Human Activity |
| Handwriting [94] | 150 | 152 | 3 | Classification (26) | Human Activity |
| Epilepsy [105] | 137 | 207 | 3 | Classification (4) | Human Activity |
| StarLightCurves [91] | 1000 | 1024 | 1 | Classification (3) | Sensor |
| ETTh2$_{P96}$ [127] | 8449 | 96 | 7 | Forecast (96) | Electricity |
| ETTh2$_{P192}$ [127] | 8353 | 96 | 7 | Forecast (192) | Electricity |
| ETTh2$_{P336}$ [127] | 8209 | 96 | 7 | Forecast (336) | Electricity |
| ETTh2$_{P720}$ [127] | 7825 | 96 | 7 | Forecast (720) | Electricity |
| ETTm1$_{P96}$ [127] | 34369 | 96 | 7 | Forecast (96) | Electricity |
| ETTm1$_{P192}$ [127] | 34273 | 96 | 7 | Forecast (192) | Electricity |
| ETTm1$_{P336}$ [127] | 34129 | 96 | 7 | Forecast (336) | Electricity |
| ETTm1$_{P720}$ [127] | 33745 | 96 | 7 | Forecast (720) | Electricity |
| SaugeenRiverFlow [73] | 18921 | 48 | 1 | Forecast (24) | Weather |

**Few-shot learning.** UNITS model $F(\mathcal{X}, \theta)$ pre-trained on $\mathcal{X}$, can be fine-tuned on a few samples on new data $\hat{\mathcal{X}}$ and new tasks $\hat{\mathcal{Y}}$, i.e., *Few-Shot*$\{F(\mathcal{X}, \theta), \hat{\mathcal{X}}\} = F(\hat{\mathcal{X}}, \hat{\theta}) \rightarrow \hat{\mathcal{Y}}$. We verify the few-shot learning ability of UNITS on forecasting and classification tasks on new, out-of-domain datasets and on new types of tasks, including imputation and anomaly detection.

**Single-task learning.** UNITS model can also conduct the single-task learning same as the existing works, where each model is separately trained on each dataset $\mathcal{D}_i = (\mathcal{X}_i, \mathcal{Y}_i)$, i.e., $F(\mathcal{X}_i, \theta_i) \rightarrow \mathcal{Y}_i$.

Table 9: Datasets for zero-shot forecasting. Prediction length is indicated in parenthesis. Note that only the first 500 variables are used for the Web Traffic and Temperature Rain datasets.

| Name | Sequence Length | Variables | Task | Class |
|---|---|---|---|---|
| Solar [83] | 128 | 137 | Forecast (64) | Electricity |
| SaugeenRiverFlow [73] | 256 | 1 | Forecast (128) | Weather |
| Hospital [44] | 32 | 767 | Forecast (16) | Healthcare |
| Web Traffic [74] | 160 | 500 | Forecast (80) | Web |
| Temperature Rain [33] | 96 | 500 | Forecast (48) | Weather |

Table 10: Datasets for imputation tasks.

| Name | Sequence Length | Variables | Task | Mask ratio | Class |
|---|---|---|---|---|---|
| ETTm1 [127] | 96 | 7 | Imputation | 12.5%, 25%, 37.5%,50% | Electricity |
| ETTh1 [127] | 96 | 7 | Imputation | 12.5%, 25%, 37.5%,50% | Electricity |
| ECL[102] | 96 | 321 | Imputation | 12.5%, 25%, 37.5%,50% | Electricity |
| Weather [110] | 96 | 21 | Imputation | 12.5%, 25%, 37.5%,50% | Weather |

Table 11: Datasets for anomaly detection tasks.

| Name | Sequence Length (Multi-task) | Sequence Length (Single-task) | Variables | Task | Class |
|---|---|---|---|---|---|
| SMD [96] | 96 | 100 | 38 | Anomaly detection | Machine |
| MSL [43] | 96 | 100 | 55 | Anomaly detection | Spacecraft |
| SMAP [43] | 96 | 100 | 25 | Anomaly detection | Spacecraft |
| SWaT [77] | 96 | 100 | 51 | Anomaly detection | Infrastructure |
| PSM [1] | 96 | 100 | 25 | Anomaly detection | Machine |

Table 12: Additional notation.

| Variable | Description |
|---|---|
| $\mathcal{D}$ | Multi-domain dataset collection |
| $n$ | Number of datasets in $\mathcal{D}$ |
| $\mathcal{D}_i$ | The $i_{\text{th}}$ dataset in $\mathcal{D}$ |
| $\mathcal{X}_i$ | All time series samples in the dataset $\mathcal{D}_i$ |
| $\mathcal{X}$ | A collection of $\mathcal{X}_i$ |
| $\mathcal{Y}_i$ | A time series task defined on $\mathcal{X}_i$ |
| $\mathcal{Y}$ | A collection of tasks $\mathcal{Y}_i$ |
| $\mathbf{x}$ | One time series sample from the dataset |
| $t$ | The length of time series sample $\mathbf{x}$ |
| $v$ | The number of variables/sensors of sample $\mathbf{x}$ |
| $F(\mathcal{X}, \theta)$ | A multi-task model with weights $\theta$ trained on collection of samples $\mathcal{X}$ |
| $k$ | Patch size of a sample token |
| $d$ | Number of embedding dimension of tokens |
| $\mathbf{z_x}$ | Sample tokens converted from input sample $\mathbf{x}$ |
| $s$ | Number of sample tokens, and $s = t/e$ |
| $\mathbf{z}_p$ | Prompt tokens with number of $p$ |
| $p$ | Number of prompt tokens |
| $\mathbf{z}_m$ | A GEN token |
| $f$ | Desired number of prediction tokens of forecasting tasks |
| $\hat{\mathbf{z}}_m$ | Replicated GEN tokens with the number of $f$ |
| $\hat{\mathbf{x}}$ | The foretasted time series data points projected from the output $\hat{\mathbf{z}}_m$ |
| $\mathbf{z}_c$ | A CLS token |
| $\mathbf{z}_e$ | Class embeddings for $e$ classes of a classification task |
| $H_{\text{GEN}}$ | The GEN tower in UNITS |
| $H_{\text{CLS}}$ | The CLS tower in UNITS |

## C.2 Generalizing Task Tokens to Various Tasks

We introduce how to use tokens for forecasting and classification tasks in the manuscript. Here we present the implementation of using tokens for imputation and anomaly detection tasks.

**Imputation task.** In tasks that require imputation, GEN token $\mathbf{z}_m$ is inserted in the positions where sample tokens $\mathbf{z_x}$ are missing. This process creates an augmented sequence of tokens represented by $\hat{\mathbf{z}}_\mathbf{x}$. These augmented tokens are then concatenated along the time dimension with prompt tokens, forming the input tokens for the network:

$$\mathbf{z}_{\text{Imp}} = \text{CA}(\mathbf{z}_p, \hat{\mathbf{z}}_\mathbf{x}) \in \mathbb{R}^{(p+s) \times v \times d}, \tag{6}$$

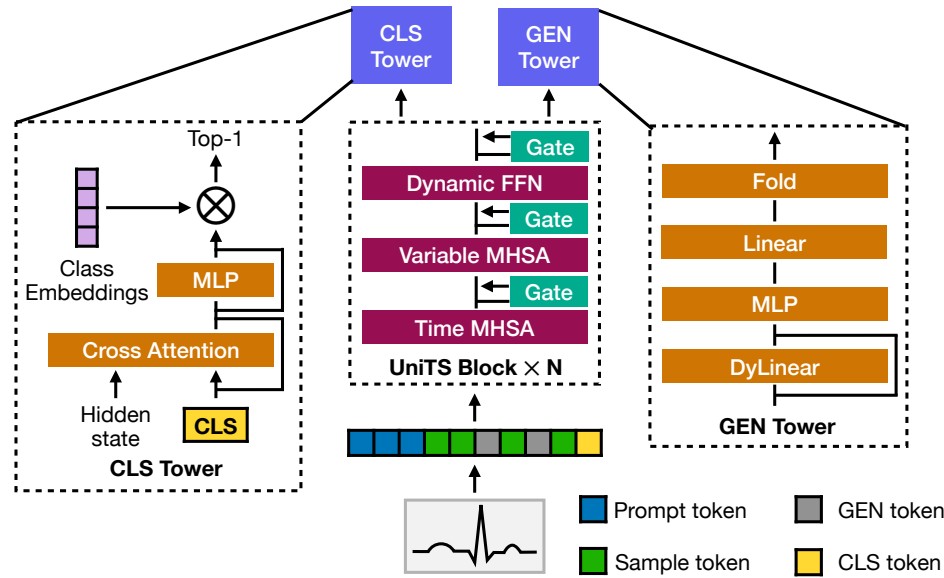

Figure 4: The network architecture of UNITS. Shared `GEN` tower and `CLS` tower transform task tokens to the prediction results of generative and predictive tasks.

where CA denotes the concatenation operation along the time dimension. Similar to the approach in forecasting tasks, the output for augmented sample tokens $\hat{\mathbf{z}}_{\mathbf{x}}$ are unpatchified to obtain the imputed sample $\hat{\mathbf{x}}$, i.e. $\hat{\mathbf{x}} = \text{Proj}(\hat{\mathbf{z}}_{\mathbf{x}})$.

**Anomaly detection task.** For the anomaly detection task, we follow TimesNet [112] to form it as a generative task, where the model is trained to reconstruct the time series sample using reconstruction error as the anomaly criterion. The prompt tokens and the sample tokens are concatenated along the time dimension to form the input tokens for the network:

$$\mathbf{z}_{\text{Ano}} = \text{CA}(\mathbf{z}_p, \mathbf{z}_{\mathbf{x}}) \in \mathbb{R}^{(p+s) \times v \times d}. \tag{7}$$

The output for sample tokens $\mathbf{z}_{\mathbf{x}}$ is unpatchified to obtain the predicted sample $\hat{\mathbf{x}}$. During inference, following the approach in [112], we determine a threshold of reconstruction error from the training and testing data, which is then used to detect anomalous time series points. Specifically, we sort the reconstruction errors between the input and output samples from our model across all training and testing sets. A predefined anomaly ratio is then applied to determine the threshold that distinguishes normal from anomalous data points.

### C.3 Implementation of UNITS Network Architecture

The UNITS network architecture is composed of $N$ UNITS blocks, one CLS tower, and one GEN tower. We introduce more implementation details of UNITS network architecture, including the Time MHSA, Variable MHSA, Dynamic FFN, and Gate Module in the UNITS block, as well as the `GEN`/`CLS` towers shared for generative and predictive tasks.

**UNITS block: time and variable MHSA.** For attention across the time dimension, the standard MHSA is applied as done by [82]. For variable MHSA, to capture relations among variables across all time points while minimizing the computational overhead associated with long time lengths, we average the $Q$ and $K$ over the time dimension to get shared $\hat{Q}$ and $\hat{K}$ as follows:

$$\hat{Q}, \hat{K} = \text{mean}_t(Q, K); Q, K, V = \text{Linear}(\mathbf{z}_{\text{in}}), \tag{8}$$

where $\text{mean}_t$ is the mean along the time dimension. Then, $\text{Output} = \text{Attn}_v V = \text{Softmax}\left(\frac{\hat{Q}\hat{K}^T}{\sqrt{d}}\right) V$ is obtained where $\text{Attn}_v \in \mathbb{R}^{v \times v}$ is the attention map among variables, which is shared for all time points. The notations for multi-head attention are omitted for simplicity. We show the effectiveness of both time and variable MHSA in Table 22.

**UNITS block: Dynamic FFN.** By argument the FFN layer in transformers with the proposed DyLinear operator, we present the Dynamic FFN module, as shown in Figure 5. In the Dynamic FFN, we replace the first linear layer in the standard FFN layer with a 3-kernel convolution across the time dimension to capture the local details. The second linear layer is kept the same as the standard FFN layer, and the DyLinear is inserted in between the input convolution and the output linear layer. Specifically, after processed by the convolution layer, the embeddings with $d$ dimension are split into two groups, resulting in $(\mathbf{z}_{\text{mid}}^1, \mathbf{z}_{\text{mid}}^2) \in \mathbb{R}^{s \times v \times d/2}$. $\mathbf{z}_{\text{mid}}^1$ and $\mathbf{z}_{\text{mid}}^2$ are processed as follows:

$$\mathbf{z}_{\text{out}} = \text{Linear}(\text{Concat}(\text{DyLinear}_M(\mathbf{z}_{\text{mid}}^1), \mathbf{z}_{\text{mid}}^2)), \qquad (9)$$

where $\text{DyLinear}_M$ processes the sample and prompt tokens in $\mathbf{z}_{\text{mid}}^1$ with two DyLinear operators, while CLS token is skipped to ensure consistency for all tasks. $\mathbf{z}_{\text{mid}}^2$ is kept unprocessed. This separation of routes for $\mathbf{z}_{\text{mid}}^1$ and $\mathbf{z}_{\text{mid}}^2$ leads to a scale combination effect, enhancing multi-scale processing ability [31].

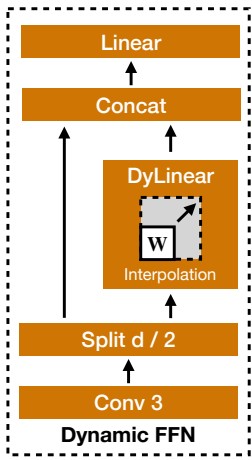

Figure 5: The dynamic FFN in UNITS.

**UNITS block: gate module.** The gate module is placed as the output of each component in the UNITS block, including time MHSA, variable MHSA, and Dynamic FFN. Specifically, given an input $\mathbf{z}_{\text{in}} \in \mathbb{R}^{s \times v \times d}$, a linear layer maps it to a scaling factor $\mathbf{x}_g \in \mathbb{R}^{s \times v \times 1}$ along the embedding dimension. This is followed by a Sigmoid function to ensure the scaling factor lies between 0 and 1. The final gating operation involves element-wise multiplication of the input by the Sigmoid-activated scaling factor, i.e.,

$$\mathbf{z}_{\text{out}} = \text{Sigmoid}(\mathbf{x}_g) \cdot \mathbf{z}_{\text{in}}, \mathbf{x}_g = \text{Linear}(\mathbf{z}_{\text{in}}). \qquad (10)$$

**GEN tower.** The GEN tower $H_{\text{GEN}}$ is designed to transform tokens into time points prediction results. One GEN tower is shared by all generative tasks, including forecasting, imputation, and anomaly detection. As shown in Figure 4, take the forecasting task as an example, the $\mathbf{z}_{\text{Fore}} \in \mathbb{R}^{(p+s+f) \times v \times d}$ from Eq. 1 is processed by the GEN tower to get the full time-series sample as follows:

$$\hat{\mathbf{x}} = \text{Proj}(\text{MLP}((\mathbf{z}_{\text{Fore}} + \text{DyLinear}(\mathbf{z}_{\text{Fore}})))), \qquad (11)$$

where the MLP is composed of two linear layers with an activation layer in between, and Proj is the unpatchify operation that transfers the embedding back to the time series patch as introduced in Section 4.1. For imputation and anomaly detection tasks, only the tokens are modified while the GEN tower remains unchanged.

**CLS tower.** The CLS tower $H_{\text{CLS}}$ transforms CLS tokens into classification classes. The CLS tower is shared across all classification tasks from different datasets. As illustrated in Figure 4, the CLS tower processes $\mathbf{z}_{\text{Pred}} \in \mathbb{R}^{(p+s+1) \times v \times d}$ from Eq. 2, which includes the CLS token $\mathbf{z}_c'$, to produce the final CLS token $\mathbf{z}_c$ as follows:

$$\mathbf{z}_c = \mathbf{z}_c'' + \text{MLP}(\mathbf{z}_c''), \quad \mathbf{z}_c'' = \mathbf{z}_c' + \text{CrossAtt}(\text{Query} = \mathbf{z}_c', \text{K} = \text{V} = \mathbf{z}_{\text{Pred}}), \qquad (12)$$

where the CLS token $\mathbf{z}_c'$ serves as a query to perform cross-attention with all tokens in $\mathbf{z}_{\text{Pred}}$. Subsequently, the processed CLS token $\mathbf{z}_c$ is matched with class embeddings to determine the predicted class as described in Eq. 3.

# D  Implementation Details

## D.1  Model Details

By default, in a multi-task setting, the UNITS network comprises three UNITS blocks, one GEN tower, and one CLS tower. For each data source, the prompt tokens and task tokens are defined. Forecasting tasks on the same data source but with different forecast lengths share the same prompt and GEN token. For zero-shot learning on new datasets, we use a shared prompt and GEN token across all data sources to facilitate zero-shot learning. Tokens are trained to achieve their functions. The number of embedding dimensions, $d$, is set to 64 for UNITS-*SUP* and 128 for UNITS-*PMT*. All blocks in UNITS maintain the same feature shape, following the Transformer architecture.

Table 13: Baseline methods used for comparison in this paper.

| Task | Method Types | Method |
|---|---|---|
| Forecasting | LLM-reprogrammed | TEMPO [10] TIME-LLM [47] LLM4TS [12] TEST [97] GPT4TS [129] |
| | Transformer-based | MOMENT [36] iTransformer [67] PatchTST [82] Crossformer [126] FEDformer [128] Stationary [69] Autoformer [114] |
| | MLP-based | TSMixer [14] RLinear [59] DLinear [119] |
| | Frequency-based | TimesNet [112] |
| | Conv-based | TiDE [21] SCINet [64] |
| Classification | LLM-reprogrammed | GPT4TS [129] |
| | Frequency-based | TimesNet [112] |
| | MLP-based | DLinear [119] LightTS [122] |
| | Transformer-based | iTransformer [67] PatchTST [82] Transformer [104] Reformer [51] Informer [127] Pyraformer [65] Autoformer [114] Stationformer [69] FEDformer [128] ETSformer [111] Flowformer [113] |
| | TCN-based | TCN [30] |
| | RNN-based | LSTM [41] LSTNet [53] LSSL [39] |
| | Classical methods | DTW [6] XGBoost [15] Rocket [24] |
| Imputation | Frequency-based | TimesNet [112] |
| | MLP-based | DLinear [119] LightTS [122] |
| | Transformer-based | iTransformer [67] PatchTST [82] Reformer [51] Informer [127] Pyraformer [65] Autoformer [114] Stationformer [69] FEDformer [128] ETSformer [111] LogTransfomer [57] |
| | TCN-based | TCN [30] |
| | RNN-based | LSTM [41] LSSL [39] |
| Anomaly detection | Frequency-based | TimesNet [112] |
| | MLP-based | DLinear [119] LightTS [122] |
| | Transformer-based | iTransformer [67] PatchTST [82] Transformer [104] Reformer [51] Anomaly Transformer [116] Informer [127] Pyraformer [65] Autoformer [114] Stationformer [69] FEDformer [128] ETSformer [111] LogTransfomer [57] |
| | TCN-based | TCN [30] |
| | RNN-based | LSTM [41] LSSL [39] |

## D.2 Training Details

For multi-task settings, all models are jointly trained on multiple tasks following the same training protocol. To match the size of the largest dataset, samples from each dataset are repeated in every training epoch. In each inference step, datasets are randomly sampled with equal probability, utilizing a batch size of 32. Supervised training involves 5 epochs using gradient accumulation for an effective batch size of 1024, starting with a learning rate of 3.2e-2 and adjusted with a multi-step decayed schedule. The $\lambda_i$ in $L_{\text{total}}$ are all set to 1 in this work. For self-supervised pre-training, the models are trained over 10 epochs with an effective batch size of 4096 and an initial learning rate of 6.4e-3, using a cosine decay schedule. All experiments are conducted using A100-40G GPUs. Each experiment is conducted with one or two GPUs, and the maximum running time is under 48 hours.

Since all models are jointly trained across multiple tasks, we report the average performance for each task type. For tasks involving forecasting and imputation, model performance is assessed using Mean Squared Error (MSE) and Mean Absolute Error (MAE). In classification tasks, accuracy is used as the primary evaluation metric. For anomaly detection tasks, performance is measured using precision, recall, and the F1-score.

**No task-specific hyper-parameter tuning.** UNITS is designed for multi-task settings where tasks share the same model weights. In UNITS, we do not need to perform any task-specific hyper-parameter tuning. The baseline methods follow the same training setting as our method to ensure a fair comparisons.

## D.3 Further Information on Pre-training

During the unified pre-training, we introduce two distinct masking schemes: the random masking scheme and the right masking scheme. The time series sample is initially truncated to a length randomly selected within the range of 50% to 100% of its original length. Subsequently, in the random masking scheme, a certain proportion $p_{\text{rand}}$ of tokens are masked at random positions within the time dimension. For the right masking scheme, designed to enhance the model's forecasting ability, a random proportion $p_{\text{right}}$ of tokens on the right side of the sample is masked. Both $p_{\text{rand}}$ and $p_{\text{right}}$ are set to 70%-80%. Each training step randomly utilizes one of these two schemes with equal probability.

### D.4 Implementation Details of Baselines

The baseline methods used in this paper are summarized in Table 13. Unlike UniTS, which can handle diverse data and tasks within a single model, baseline methods cannot be directly used for unified training because: 1) To accommodate data with varying numbers of variables, baseline methods typically use a data-specific input head to project features from the variable count to a fixed number of embedding dimensions. 2) Similarly, to manage different tasks, such as classification with various classes and forecasting with different lengths, baseline methods employ task-specific output heads to transform the features into the appropriate task outputs. Since baseline methods are designed for single-task training, in their original setting, data/task-specific heads are used for each data and task. In the multi-task learning setting, to make baseline methods support unified training, we add separate input heads to project data into a shared embedding space and separate output heads to convert the shared model output into task-specific outputs. However, using separate input and output heads makes it hard to generalize to new datasets and tasks. We employ the same fully supervised multi-task training approach as UniTS. In this setting, model networks are stacked with 3 basic building blocks, except for GPT4TS, which utilizes the prescribed setting of 6 GPT blocks. For both the proposed method and patch-based baseline approaches, the patch size and stride are fixed at 16. The input and output heads of baseline methods are duplicated for each task to create data/task-specific heads tailored for each data source and task. For single-task learning settings, we follow the original settings of baseline methods and compare results reported in their papers.

## E  Additional Results: Prompt Learning and Pre-training

We do more analysis on the prompting and pre-training of UNiTS. The average performance under 38 datasets with the multi-task setting is reported.

**Prompt learning with model scaling.** In Table 14, we further explore the capabilities of prompt learning in the SSL pre-trained UNiTS model across different model sizes. As UNiTS model size grows, we observe consistent improvements in performance for both classification and forecasting, suggesting that larger SSL models contain more robust representations for prompt learning.

Table 14: Enhancing prompt learning capability of pre-trained UNiTS through model scaling. Average performance on 20 forecasting tasks and 18 classification tasks are reported.

| Prompt Learning | Par. | Classification Acc↑ | Forecasting MSE↓ | Forecasting MAE↓ |
|---|---|---|---|---|
| UNiTS-$SUP_{\times 64}$ | 3.41M | 81.6 | 0.439 | 0.381 |
| UNiTS-$PMT_{\times 32}$ | 1.57M | 78.0 | 0.471 | 0.388 |
| UNiTS-$PMT_{\times 64}$ | 3.41M | 79.0 | 0.460 | 0.383 |
| UNiTS-$PMT_{\times 96}$ | 5.67M | 79.2 | 0.458 | 0.382 |
| UNiTS-$PMT_{\times 128}$ | 8.24M | **81.2** | **0.453** | **0.376** |

Table 15: Ablation on the number of prompt tokens.

| Prompt token Num. | $Acc_{Avg}$↑ | $MSE_{Avg}$↓ | $MAE_{Avg}$↓ |
|---|---|---|---|
| No | 81.0 | 0.460 | 0.391 |
| 5 | 81.5 | 0.455 | 0.387 |
| 10 | **81.6** | **0.439** | **0.381** |

Table 16: Ablation on using shared/unshared prompt tokens in UNiTS network.

| | $Acc_{Avg}$↑ | $MSE_{Avg}$↓ | $MAE_{Avg}$↓ |
|---|---|---|---|
| Unshared prompt tokens | **81.6** | **0.439** | **0.381** |
| Shared prompt tokens | 81.4 | 0.450 | 0.387 |

**Effect of prompt tokens.** Prompt tokens learn the contextual information related to the given data source and task types. By default, we use 10 prompt tokens for each task. We present an ablation

study on the use of different numbers of prompt tokens in Table 15. Utilizing prompt tokens leads to notable improvements in both forecasting and classification tasks. The average classification accuracy improves from 81.0% to 81.6%, and the average MSE and MAE improve from 0.460 to 0.439 and 0.391 to 0.381, respectively. Employing 10 instead of 5 prompt tokens results in greater gains in forecasting tasks and a marginal improvement of 0.1% in classification accuracy, indicating that forecasting tasks benefit more from the contextual information provided by the prompt tokens. We also evaluate the case where all prompt tokens are shared among tasks in Table 16. Using shared prompt tokens across different tasks results in a performance decline, yet this approach still surpasses the performance of models that do not utilize prompt tokens.

Table 17: Ablation on the pre-training scheme.

| UNiTS-*PMT* | Acc$_{Avg}$↑ | MSE$_{Avg}$↓ | MAE$_{Avg}$↓ |
|---|---|---|---|
| Unified Pre-training | **78.0** | **0.471** | **0.388** |
| Without CLS token based reconstruction loss | 33.1 | 0.484 | 0.393 |
| Without Prompt token based reconstruction loss | 76.8 | 0.967 | 0.656 |

**Unified pre-training.** In Equation 5, the proposed unified mask reconstruction pre-training loss is detailed, consisting of two components: the mask reconstruction loss associated with prompt tokens and the mask reconstruction loss related to CLS tokens. Table 17 presents the results where either the CLS token-based reconstruction loss or the prompt token-based reconstruction loss is omitted. The performance of prompt learning is reported. The results highlight the impact of each loss component on the learning performance.

Specifically, excluding the CLS token-based loss resulted in a significant decline in classification performance, dropping sharply from 78.0% to 33.1%. This substantial drop underscores the critical role of the CLS token-based pre-training loss in enabling the model's classification capabilities. Conversely, the removal of the prompt token-based loss adversely affected the forecasting performance. For instance, the MSE drops from 0.471 to 0.967. This deterioration in performance demonstrates the importance of prompt token-based pre-training in generative tasks.

**Pre-training with scaled numbers of epochs and data sizes.** To evaluate the effect of scaling effect of pre-training, we conduct experiments of pre-training UniTS by varying the size of the pre-training dataset and the amount of training epochs. As demonstrated in Table 18, increasing the number of pre-training epochs improves performance on both forecasting and classification tasks. Similarly, increasing the size of pre-training dataset improves performance on both forecasting and classification tasks, as shown in Table 19.

Table 18: Performance of UniTS under different pre-training epochs, average performance on 20 forecasting and 18 classification are reported.

| Pre-training steps | 1 epoch | 3 epochs | 5 epochs | 8 epochs | 10 epochs |
|---|---|---|---|---|---|
| Acc$_{Avg}$↑ (Cls.) | 75.1 | 76.8 | 78.2 | 77.0 | 79.0 |
| MSE$_{Avg}$↓ (Fore.) | 0.493 | 0.479 | 0.484 | 0.473 | 0.460 |
| MAE$_{Avg}$↓ (Fore.) | 0.410 | 0.391 | 0.389 | 0.386 | 0.383 |

Table 19: Performance of UniTS under different pre-training data sizes, average performance on 20 forecasting and 18 classification are reported. Pre-training data size refers to the proportion of the total training set used.

| Pre-training data size | 10% | 30% | 50% | 80% | 100% |
|---|---|---|---|---|---|
| Acc$_{Avg}$↑ (Cls.) | 74.2 | 76.3 | 77.6 | 78.8 | 79.0 |
| MSE$_{Avg}$↓ (Fore.) | 0.502 | 0.462 | 0.483 | 0.465 | 0.460 |
| MAE$_{Avg}$↓ (Fore.) | 0.417 | 0.385 | 0.391 | 0.384 | 0.383 |

**Cross-task pre-training.** We evaluate the effect of cross-task pre-training by pre-training a model using our pre-training strategy on either generative tasks (forecasting) or predictive tasks (classification). Table 20 shows that UniTS, pre-trained solely on forecasting datasets, achieves similar performance to the model pre-trained on both forecasting and classification data. Despite not encountering any

classification datasets during pre-training, it still performs well on classification tasks. When the model is pre-trained exclusively on classification datasets, performance on both classification and forecasting tasks drops significantly compared to the model pre-trained on both types of data. Given that the data amount of forecasting datasets is larger than classification datasets (22920 vs. 5022 iterations per epoch), this suggests that the larger amount of data plays a more crucial role in pre-training effectiveness than the data type.

Table 20: Cross-task pre-training evaluation on UniTS, average performance on 20 forecasting and 18 classification tasks are reported.

| | | Evaluation data | |
| Pre-training data type | $\text{Acc}_{Avg}\uparrow$ (Cls.) | $\text{MSE}_{Avg}\downarrow$ (Fore.) | $\text{MAE}_{Avg}\downarrow$ (Fore.) |
|---|---|---|---|
| 20 forecasting datasets | 78.5 | 0.454 | 0.379 |
| 18 classification datasets | 74.1 | 0.583 | 0.807 |
| Full 38 datasets | 79.0 | 0.460 | 0.383 |

**Cross-domain pre-training.** We evaluate the effect of cross-domain data pre-training, where the model is pre-trained on either Weather-domain datasets or Traffic-domain datasets. In Table 21, compared to joint pre-training on both domains, the performance decreases with single-domain pre-training, where pre-training is conducted solely on the downstream dataset's domain, showing the advantage of joint pre-training. For instance, the MSE on Weather datasets goes from 0.253 to 0.259. Compared to single-domain pre-training, cross-domain pre-training leads to larger performance drops, e.g., pre-training on Traffic datasets and then evaluating on Weather datasets results in an MSE increase from 0.259 to 0.289. Interestingly, pre-training on Weather datasets achieves better performance across both domains, suggesting that data from certain domains might be more beneficial for pre-training.

Table 21: Cross-domain pre-training evaluation on UniTS, average performance on 4 Weather or Traffic dataset domains are reported.

| Pre-training data | Weather datasets (4 sets) $\text{MSE}_{Avg}/\text{MAE}_{Avg}\downarrow$ (Fore.) | Traffic datasets (4 sets) $\text{MSE}_{Avg}/\text{MAE}_{Avg}\downarrow$ (Fore.) |
|---|---|---|
| Weather domain (4 datasets) | 0.259 / 0.287 | 1.338 / 0.768 |
| Traffic domain (4 datasets) | 0.289 / 0.314 | 0.680 / 0.438 |
| Weather + Traffic domains (8 sets) | 0.253 / 0.282 | 0.511 / 0.320 |

# F Additional Results: Ablation Studies of UNITS

We conduct an ablation study to verify the effectiveness of the key designs in UNITS. The average performance under 38 datasets with the multi-task setting is reported.

Table 22: Ablation on the MHSA in UNITS.

| | $\text{Acc}_{Avg}\uparrow$ | $\text{MSE}_{Avg}\downarrow$ | $\text{MAE}_{Avg}\downarrow$ |
|---|---|---|---|
| UNITS-*SUP* | **81.6** | **0.439** | **0.381** |
| Without Time MHSA | 80.7 | 0.449 | 0.380 |
| Without Variable MHSA | 80.8 | 0.444 | 0.383 |

**Effect of time and variable MHSA.** In Table 22, we present an ablation study to assess the impact of both Time and Variable MHSA on the UNITS model. When the Time MHSA is removed from the UNITS model, we observe a decrease in performance, where the average accuracy drops to 80.7%, and the MSE drops to 0.449. Similarly, eliminating the Variable MHSA from the UNITS model results in diminished performance. This scenario yields a decreased accuracy of 80.8%, a decrease in MSE to 0.444, and a reduction in MAE to 0.383. These experimental findings highlight the crucial role that both Time and Variable MHSA play in the efficacy of the UNITS model.

**Effect of Dynamic FFN.** In Table 23, we present an ablation study on the Dynamic FFN layer in the UNITS network. The UNITS, which incorporates the Dynamic FFN, achieves the highest performance with an average accuracy of 81.6%, demonstrating effectiveness in handling classification

Table 23: Ablation on the MLP layer in UNITS network.

| | Acc$_{Avg}\uparrow$ | MSE$_{Avg}\downarrow$ | MAE$_{Avg}\downarrow$ |
|---|---|---|---|
| UNITS-*SUP* | **81.6** | **0.439** | **0.381** |
| Dynamic FFN $\rightarrow$ MLP | 81.3 | 0.462 | 0.394 |
| Without Dynamic FFN | 80.8 | 0.465 | 0.396 |

tasks. It also shows superior results in terms of MSE and MAE in forecasting tasks, with scores of 0.439 and 0.381 respectively. The model variant where the Dynamic FFN is replaced with a standard MLP layer exhibits a decrease in performance. The average accuracy dropped to 81.3%, and MSE and MAE dropped to 0.462 and 0.394, respectively. This variation suggests the effect of Dynamic FFN for the UNITS. The performance is observed when the Dynamic FFN is completely removed from the model, highlighting the importance of Dynamic FFN layers in UNITS network.

Table 24: Ablation on the gate module in UNITS network.

| | Acc$_{Avg}\uparrow$ | MSE$_{Avg}\downarrow$ | MAE$_{Avg}\downarrow$ |
|---|---|---|---|
| UNITS-*SUP* | **81.6** | **0.439** | **0.381** |
| Without Gate module | 81.1 | 0.459 | 0.387 |

**Effect of gate module.** In Table 24, we present a comparison of the UNITS model with and without the inclusion of the gate module. Incorporating the gate module yields consistent enhancements relative to the baseline model that lacks it. Specifically, the addition of the gate module results in an increase in classification accuracy, moving from 81.1% to 81.6%. For the forecasting task, the MSE sees an improvement from 0.459 to 0.439, and the MAE decreases from 0.387 to 0.381. These results show the effectiveness of the gate module in mitigating task interference by adjusting the scaling of embedding vectors.

Table 25: Zero-shot multi-task learning on forecasting tasks on 5 out-of-domain data with new forecasting length and new number of variables. We set shared prompt tokens and GEN tokens for UNITS. One sample from each dataset is used following [81].

| | Var. | Pred. | UNITS-*Zero-shot* | | LLMTime | |
|---|---|---|---|---|---|---|
| | | | MSE$\downarrow$ | Inf. Time | MSE$\downarrow$ | Inf. Time |
| Solar | 137 | 64 | 0.030 | $6.8e^{-3}$ | 0.265 | $2.0e^3$ |
| River | 1 | 128 | 0.456 | $1.4e^{-2}$ | 0.832 | $3.5e^1$ |
| Hospital | 767 | 16 | 1.045 | $5.9e^{-3}$ | 1.319 | $2.9e^3$ |
| Web Tr. | 500 | 80 | 1.393 | $5.9e^{-3}$ | 1.482 | $9.5e^3$ |
| Temp. Rain | 500 | 48 | 11.51 | $1.6e^{-1}$ | 5.69 | $5.3e^3$ |

**Comparison with Transformer.** To verify the effectiveness of UNITS structure, we compare the original Transformer with UNITS. The unified tokenization and co-training strategy are applied to both models. The results shown in Table 26 indicate that UNITS clearly outperforms the Transformer in both classification and forecasting tasks, suggesting that merely using a transformer structure is insufficient for achieving robust multi-task performance on time series datasets.

# G  Additional Results: UNITS for Zero-Shot Forecasting on New Datasets

**Setup.** When UNITS is trained with shared prompt and GEN tokens across all forecasting tasks, it acquires the ability to perform zero-shot forecasting on datasets with new lengths and variable numbers that were not part of its training domain. We evaluate UNITS in a zero-shot setting on five new forecasting tasks as referenced in Table 9. These tasks have varying forecasting lengths and numbers of variables compared to those seen by UNITS during pre-training. We benchmark against LLMTime [81], a model designed for zero-shot forecasting using LLMs. Following LLMTime, we utilize one sample from each dataset to manage the extensive inference costs. We exclude a related method, Time-LLM [47], from experiments. Time-LLM supports zero-shot learning but requires that the forecasting length and the number of variables/sensors for zero-shot prediction are the same as those used for training.

Table 26: Comparison between UNITS and Transformer structure. The unified tokenization and co-training strategy are applied to both models.

| | $\text{Acc}_{Avg}\uparrow$ | $\text{MSE}_{Avg}\downarrow$ | $\text{MAE}_{Avg}\downarrow$ |
|---|---|---|---|
| Transformer-network | 80.2% | 0.468 | 0.397 |
| **UNITS-network** | **81.6%** | **0.439** | **0.381** |

Table 27: Multi-task learning comparison with existing networks under 20 forecasting tasks and 18 classification tasks. UNITS handles all tasks with a unified model and no task-specific head. While baseline models have a shared backbone but task-specific input/output heads for each dataset/task. **Bold** indicates best-performing model for that dataset while underline is second-best.

| CLASSIFICATION DATASETS | UNITS-*SUP* ACCURACY↑ | UNITS-*PMT* ACCURACY↑ | iTRANSFORMER ACCURACY↑ | TIMESNET ACCURACY↑ | PATCHTST ACCURACY↑ | PYRAFORMER ACCURACY↑ | AUTOFORMER ACCURACY↑ | GPT4TS ACCURACY↑ |
|---|---|---|---|---|---|---|---|---|
| HEARTBEAT | 0.639 | 0.654 | 0.668 | **0.727** | 0.659 | **0.727** | 0.717 | 0.698 |
| JAPANESEVOWELS | 0.922 | 0.903 | 0.959 | **0.976** | 0.941 | 0.854 | 0.941 | 0.946 |
| PEMS-SF | 0.832 | 0.827 | 0.832 | 0.775 | **0.838** | 0.832 | 0.792 | 0.792 |
| SELFREGULATIONSCP2 | 0.489 | **0.572** | 0.489 | 0.528 | 0.489 | 0.567 | 0.45 | 0.456 |
| SPOKENARABICDIGITS | 0.968 | 0.955 | 0.978 | **0.987** | 0.975 | 0.921 | 0.973 | 0.975 |
| UWAVEGESTURELIBRARY | 0.822 | **0.853** | 0.822 | 0.844 | 0.819 | 0.722 | 0.422 | 0.819 |
| ECG5000 | 0.928 | 0.924 | 0.933 | 0.926 | **0.943** | 0.914 | 0.919 | 0.93 |
| NONINVASIVEFETALECGTHORAX1 | **0.896** | 0.808 | 0.882 | 0.889 | 0.865 | 0.214 | 0.217 | 0.897 |
| BLINK | **0.976** | 0.916 | 0.933 | 0.876 | 0.896 | 0.882 | 0.631 | 0.924 |
| FACEDETECTION | 0.654 | 0.58 | 0.66 | 0.662 | 0.639 | **0.673** | 0.592 | 0.661 |
| ELECTRICDEVICES | 0.622 | 0.624 | 0.573 | 0.495 | 0.595 | **0.654** | 0.561 | 0.629 |
| TRACE | 0.96 | **0.99** | 0.79 | 0.91 | 0.77 | 0.74 | 0.6 | 0.96 |
| FORDB | 0.759 | **0.78** | 0.727 | 0.689 | 0.614 | 0.553 | 0.664 | 0.777 |
| MOTIONSENSEHAR | 0.951 | **0.958** | 0.936 | 0.906 | 0.758 | 0.887 | 0.302 | 0.962 |
| EMOPAIN | 0.797 | **0.814** | 0.794 | 0.78 | 0.792 | 0.814 | 0.699 | 0.794 |
| CHINATOWN | **0.98** | **0.98** | 0.974 | 0.977 | 0.977 | 0.274 | 0.968 | 0.965 |
| MELBOURNEPEDESTRIAN | 0.876 | 0.839 | 0.893 | **0.957** | 0.804 | 0.523 | 0.75 | 0.94 |
| SHAREPRICEINCREASE | 0.618 | 0.638 | 0.619 | 0.65 | **0.68** | 0.631 | 0.615 | 0.637 |
| BEST COUNT | 3/18 | 7/18 | 0/18 | 4/18 | 3/18 | 4/18 | 0/18 | 2/18 |
| AVERAGE SCORE | **0.816** | 0.812 | 0.803 | 0.809 | 0.781 | 0.688 | 0.656 | 0.820 |
| FULLY SHARED MODEL | ✓ | ✓ | ✗ | ✗ | ✗ | ✗ | ✗ | ✗ |

**Results.** UNITS considerably surpasses LLMTime across most of the tested datasets, demonstrating superior performance in handling different forecasting lengths and variable numbers (Table 25). For example, UNITS achieves a 45.2% improvement in MSE over LLMTime (0.456 vs. 0.832) on River. Remarkably, UNITS exhibits an inference speed approximately $10^6$ times faster than LLMTime.

# H Additional Results: Relation among Prompt Tokens

We calculate the similarity between prompt tokens across datasets, as illustrated in Figure 7. Datasets within the same class, for instance, FaceDetection and SelfRegulationSCP2, which both consist of EEG data, demonstrate a higher similarity. While some out-of-domain datasets still exhibit strong similarities, indicating that they share certain similar requirements.

To compare the difference among tokens before and after training, beyond similarity comparison, we show UMAP plots generated with the prompt tokens before and after training, in Figure 8 and Figure 9. Before training, the prompt tokens from all datasets are dispersed. In contrast, the UMAP of prompt tokens after training reveals that tokens from the same datasets are clustered. However, some tokens from different datasets remain closely positioned, indicating that data from different domains share similar information.

# I Additional Results: Classification Performance Stratified by Datasets

We present the performance of multi-task classification on each dataset in Table 27.

Table 28: Full results of few-shot multi-task learning of block-wise imputation tasks on 6 datasets.

| Imputation | Mask Ratio | ECL MSE | ECL MAE | ETTh1 MSE | ETTh1 MAE | ETTh2 MSE | ETTh2 MAE | ETTm1 MSE | ETTm1 MAE | ETTm2 MSE | ETTm2 MAE | Weather MSE | Weather MAE | Avg MSE | Avg MAE | Best Count | Shared |
|---|---|---|---|---|---|---|---|---|---|---|---|---|---|---|---|---|---|
| TimesNet-*FT* | 25% | 0.245 | 0.339 | 0.369 | 0.403 | 0.193 | 0.292 | 0.442 | 0.418 | 0.119 | 0.229 | 0.106 | 0.152 | 0.246 | 0.305 | 0/12 | × |
|  | 50% | 0.258 | 0.350 | 0.412 | 0.420 | 0.211 | 0.302 | 0.607 | 0.485 | 0.140 | 0.247 | 0.125 | 0.171 | 0.292 | 0.329 | 0/12 | × |
| PatchTST-*FT* | 25% | 0.195 | 0.297 | 0.315 | 0.361 | 0.147 | 0.251 | 0.309 | 0.337 | 0.092 | 0.193 | 0.089 | 0.122 | 0.191 | 0.260 | 0/12 | × |
|  | 50% | 0.230 | 0.323 | 0.353 | 0.382 | 0.175 | 0.271 | 0.442 | 0.400 | 0.111 | 0.214 | 0.105 | 0.139 | 0.236 | 0.288 | 0/12 | × |
| iTrans-*FT* | 25% | 0.174 | 0.275 | 0.301 | 0.359 | 0.185 | 0.293 | 0.254 | 0.319 | 0.113 | 0.227 | 0.087 | 0.127 | 0.186 | 0.266 | 0/12 | × |
|  | 50% | 0.203 | 0.300 | 0.332 | 0.376 | 0.205 | 0.307 | 0.372 | 0.382 | 0.136 | 0.252 | 0.106 | 0.150 | 0.226 | 0.295 | 0/12 | × |
| UNITS-*PMT* | 25% | **0.117** | **0.231** | 0.281 | **0.339** | **0.177** | **0.281** | 0.247 | 0.308 | 0.095 | 0.198 | 0.075 | 0.113 | 0.165 | 0.245 | 5/12 | ✓ |
|  | 50% | **0.135** | **0.248** | 0.323 | 0.365 | **0.246** | 0.331 | 0.343 | 0.364 | 0.131 | 0.237 | **0.093** | 0.139 | 0.212 | 0.281 | 4/12 | ✓ |
| UNITS-*FT* | 25% | 0.143 | 0.255 | **0.277** | 0.341 | 0.194 | 0.284 | **0.204** | **0.281** | 0.088 | **0.186** | 0.074 | **0.105** | 0.163 | 0.242 | 7/12 | ✓ |
|  | 50% | 0.161 | 0.273 | **0.313** | **0.361** | 0.252 | **0.322** | **0.295** | **0.334** | **0.119** | **0.223** | 0.096 | **0.135** | **0.206** | **0.275** | 8/12 | ✓ |

# J  Additional Results: Direct Multi-step Forecasting on New Forecasting Lengths

**Average inference steps comparison.** In Table 6, we present a comparison of the average number of inference steps required by our direct multi-step inference method and the multi-step sliding window-based inference approach. Contrary to the direct multi-step inference, which is completed in a single step, the sliding window-based method necessitates multiple inference steps. Specifically, for the maximum extra inference length of 384, the sliding window-based approach demands, on average, 3.66 times more inference steps.

# K  Additional Results: Benchmarking in the Single-Task Regime

**Setup.** As we are the first work that focuses on time series multi-task learning with one model, to make fair comparisons with existing time series methods, we compare them with the single-task setting. In this setting, for each dataset, one model is independently trained with tuned hyperparameters. Following existing works [112, 67, 14], we tune the following hyperparameters, including number of channels, patch size, number of layers, learning rate, and dropout ratio. The baseline methods for time series forecasting, classification, anomaly detection, and imputation, are listed in Table 13. We following existing works [112, 67] to use 36 commonly used datasets for forecasting (Table 30), 10

Table 29: Full results of few-shot multi-task learning on 9 forecasting and 6 classification tasks on out-of-domain datasets. Ratio is the data ratio of the dataset used for training.

| **Classification** (Acc↑) (6 datasets) | 5% iTrans-*FT* | 5% UNITS-*PMT* | 5% UNITS-*FT* | 15% iTrans-*FT* | 15% UNITS-*PMT* | 15% UNITS-*FT* | 20% iTrans-*FT* | 20% UNITS-*PMT* | 20% UNITS-*FT* |
|---|---|---|---|---|---|---|---|---|---|
| ECG200 | 0.780 | 0.790 | 0.790 | 0.810 | 0.760 | 0.820 | 0.810 | 0.820 | 0.820 |
| Handwriting | 0.054 | 0.044 | 0.061 | 0.098 | 0.089 | 0.080 | 0.118 | 0.087 | 0.081 |
| SelfRegulationSCP1 | 0.928 | 0.816 | 0.758 | 0.679 | 0.648 | 0.672 | 0.771 | 0.676 | 0.737 |
| RacketSports | 0.375 | 0.316 | 0.487 | 0.546 | 0.474 | 0.618 | 0.546 | 0.539 | 0.586 |
| Epilepsy | 0.399 | 0.514 | 0.522 | 0.413 | 0.732 | 0.681 | 0.500 | 0.797 | 0.855 |
| StarLightCurves | 0.851 | 0.862 | 0.826 | 0.842 | 0.869 | 0.834 | 0.848 | 0.895 | 0.833 |
| Average | 0.564 | 0.557 | **0.574** | 0.565 | 0.595 | **0.618** | 0.599 | 0.636 | **0.652** |
| Best Count | 1/6 | 1/6 | 4/6 | 2/6 | 2/6 | 2/6 | 2/6 | 2/6 | 2/6 |

| **Forecast** (9 datasets) | 5% iTrans-*FT* MSE | MAE | 5% UNITS-*PMT* MSE | MAE | 5% UNITS-*FT* MSE | MAE | 15% iTrans-*FT* MSE | MAE | 15% UNITS-*PMT* MSE | MAE | 15% UNITS-*FT* MSE | MAE | 20% iTrans-*FT* MSE | MAE | 20% UNITS-*PMT* MSE | MAE | 20% UNITS-*FT* MSE | MAE |
|---|---|---|---|---|---|---|---|---|---|---|---|---|---|---|---|---|---|---|
| ETTh2$_{P96}$ | 0.554 | 0.500 | 0.397 | 0.406 | 0.414 | 0.419 | 0.441 | 0.440 | 0.390 | 0.404 | 0.400 | 0.409 | 0.418 | 0.426 | 0.387 | 0.403 | 0.396 | 0.407 |
| ETTh2$_{P192}$ | 0.440 | 0.438 | 0.385 | 0.399 | 0.390 | 0.401 | 0.398 | 0.410 | 0.390 | 0.403 | 0.376 | 0.393 | 0.395 | 0.407 | 0.394 | 0.406 | 0.378 | 0.395 |
| ETTh2$_{P336}$ | 0.478 | 0.467 | 0.425 | 0.434 | 0.431 | 0.434 | 0.436 | 0.441 | 0.434 | 0.436 | 0.425 | 0.430 | 0.431 | 0.438 | 0.425 | 0.435 | 0.420 | 0.428 |
| ETTh2$_{P720}$ | 0.483 | 0.480 | 0.438 | 0.451 | 0.431 | 0.444 | 0.438 | 0.453 | 0.442 | 0.452 | 0.427 | 0.444 | 0.431 | 0.449 | 0.428 | 0.448 | 0.424 | 0.442 |
| RiverFlow$_{P24}$ | 1.141 | 0.514 | 1.111 | 0.504 | 1.160 | 0.521 | 1.067 | 0.467 | 1.074 | 0.489 | 1.096 | 0.501 | 1.056 | 0.462 | 1.084 | 0.494 | 1.078 | 0.495 |
| ETTm1$_{P96}$ | 0.504 | 0.462 | 0.370 | 0.397 | 0.412 | 0.417 | 0.423 | 0.419 | 0.360 | 0.392 | 0.353 | 0.385 | 0.408 | 0.410 | 0.357 | 0.391 | 0.346 | 0.382 |
| ETTm1$_{P192}$ | 0.555 | 0.485 | 0.416 | 0.421 | 0.453 | 0.434 | 0.464 | 0.439 | 0.402 | 0.415 | 0.394 | 0.406 | 0.444 | 0.428 | 0.398 | 0.414 | 0.386 | 0.401 |
| ETTm1$_{P336}$ | 0.567 | 0.496 | 0.467 | 0.451 | 0.509 | 0.465 | 0.492 | 0.457 | 0.446 | 0.441 | 0.425 | 0.425 | 0.471 | 0.445 | 0.442 | 0.439 | 0.417 | 0.421 |
| ETTm1$_{P720}$ | 0.659 | 0.539 | 0.565 | 0.500 | 0.573 | 0.499 | 0.558 | 0.493 | 0.529 | 0.484 | 0.490 | 0.460 | 0.536 | 0.482 | 0.527 | 0.483 | 0.481 | 0.454 |
| Average | 0.598 | 0.487 | **0.508** | **0.440** | 0.530 | 0.448 | 0.524 | 0.447 | 0.496 | 0.435 | **0.487** | **0.428** | 0.510 | 0.438 | 0.494 | 0.435 | **0.481** | **0.425** |
| Best Count | 0/9 | 0/9 | 8/9 | 7/9 | 1/9 | 2/9 | 1/9 | 1/9 | 1/9 | 1/9 | 7/9 | 7/9 | 1/9 | 1/9 | 1/9 | 0/9 | 7/9 | 8/9 |

Table 30: Full results of the single-task long-term forecasting task where the model is separately trained on each dataset. The input time series sequence length is set to 96 to ensure fair comparisons. Baseline results are obtained from [67].

| Models | | UniTS-ST (Ours) | | iTransformer [67] | | RLinear [59] | | PatchTST [82] | | Crossformer [126] | | TiDE [21] | | TimesNet [112] | | DLinear [119] | | SCINet [64] | | FEDformer [128] | | Stationary [69] | | Autoformer [114] | |
|---|---|---|---|---|---|---|---|---|---|---|---|---|---|---|---|---|---|---|---|---|---|---|---|---|---|
| Metric | | MSE | MAE | MSE | MAE | MSE | MAE | MSE | MAE | MSE | MAE | MSE | MAE | MSE | MAE | MSE | MAE | MSE | MAE | MSE | MAE | MSE | MAE | MSE | MAE |
| ETTm1 | 96 | **0.310** | **0.351** | 0.334 | 0.368 | 0.355 | 0.376 | 0.329 | 0.367 | 0.404 | 0.426 | 0.364 | 0.387 | 0.338 | 0.375 | 0.345 | 0.372 | 0.418 | 0.438 | 0.379 | 0.419 | 0.386 | 0.398 | 0.505 | 0.475 |
| | 192 | **0.357** | **0.382** | 0.377 | 0.391 | 0.391 | 0.392 | 0.367 | 0.385 | 0.450 | 0.451 | 0.398 | 0.404 | 0.374 | 0.387 | 0.380 | 0.389 | 0.439 | 0.450 | 0.426 | 0.441 | 0.459 | 0.444 | 0.553 | 0.496 |
| | 336 | **0.392** | **0.408** | 0.426 | 0.420 | 0.424 | 0.415 | 0.399 | 0.410 | 0.532 | 0.515 | 0.428 | 0.425 | 0.410 | 0.411 | 0.413 | 0.413 | 0.490 | 0.485 | 0.445 | 0.459 | 0.495 | 0.464 | 0.621 | 0.537 |
| | 720 | **0.447** | **0.439** | 0.491 | 0.459 | 0.487 | 0.450 | 0.454 | 0.439 | 0.666 | 0.589 | 0.487 | 0.461 | 0.478 | 0.450 | 0.474 | 0.453 | 0.595 | 0.550 | 0.543 | 0.490 | 0.585 | 0.516 | 0.671 | 0.561 |
| | Avg | **0.377** | **0.395** | 0.407 | 0.410 | 0.414 | 0.407 | 0.387 | 0.400 | 0.513 | 0.496 | 0.419 | 0.419 | 0.400 | 0.406 | 0.403 | 0.407 | 0.485 | 0.481 | 0.448 | 0.452 | 0.481 | 0.456 | 0.588 | 0.517 |
| ETTm2 | 96 | **0.171** | **0.255** | 0.180 | 0.264 | 0.182 | 0.265 | 0.175 | 0.259 | 0.287 | 0.366 | 0.207 | 0.305 | 0.187 | 0.267 | 0.193 | 0.292 | 0.286 | 0.377 | 0.203 | 0.287 | 0.192 | 0.274 | 0.255 | 0.339 |
| | 192 | **0.238** | **0.298** | 0.250 | 0.309 | 0.246 | 0.304 | 0.241 | 0.302 | 0.414 | 0.492 | 0.290 | 0.364 | 0.249 | 0.309 | 0.284 | 0.362 | 0.399 | 0.445 | 0.269 | 0.328 | 0.280 | 0.339 | 0.281 | 0.340 |
| | 336 | **0.299** | **0.342** | 0.311 | 0.348 | 0.307 | 0.342 | 0.305 | 0.343 | 0.597 | 0.542 | 0.377 | 0.422 | 0.321 | 0.351 | 0.369 | 0.427 | 0.637 | 0.591 | 0.325 | 0.366 | 0.334 | 0.361 | 0.339 | 0.372 |
| | 720 | **0.393** | **0.395** | 0.412 | 0.407 | 0.407 | 0.398 | 0.402 | 0.400 | 1.730 | 1.042 | 0.558 | 0.524 | 0.408 | 0.403 | 0.554 | 0.522 | 0.960 | 0.735 | 0.421 | 0.415 | 0.417 | 0.413 | 0.433 | 0.432 |
| | Avg | **0.275** | **0.323** | 0.288 | 0.332 | 0.286 | 0.327 | 0.281 | 0.326 | 0.757 | 0.610 | 0.358 | 0.404 | 0.291 | 0.333 | 0.350 | 0.401 | 0.571 | 0.537 | 0.305 | 0.349 | 0.306 | 0.347 | 0.327 | 0.371 |
| ETTh1 | 96 | **0.367** | **0.393** | 0.386 | 0.405 | 0.386 | 0.395 | 0.414 | 0.419 | 0.423 | 0.448 | 0.479 | 0.464 | 0.384 | 0.402 | 0.386 | 0.400 | 0.654 | 0.599 | 0.376 | 0.419 | 0.513 | 0.491 | 0.449 | 0.459 |
| | 192 | **0.404** | 0.425 | 0.441 | 0.436 | 0.437 | 0.424 | 0.460 | 0.445 | 0.471 | 0.474 | 0.525 | 0.492 | 0.436 | 0.429 | 0.437 | 0.432 | 0.719 | 0.631 | 0.420 | 0.448 | 0.534 | 0.504 | 0.500 | 0.482 |
| | 336 | **0.405** | **0.422** | 0.487 | 0.458 | 0.479 | 0.446 | 0.501 | 0.466 | 0.570 | 0.546 | 0.565 | 0.515 | 0.491 | 0.469 | 0.481 | 0.459 | 0.778 | 0.659 | 0.459 | 0.465 | 0.588 | 0.535 | 0.521 | 0.496 |
| | 720 | **0.437** | **0.454** | 0.503 | 0.491 | 0.481 | 0.470 | 0.500 | 0.488 | 0.653 | 0.621 | 0.594 | 0.558 | 0.521 | 0.500 | 0.519 | 0.516 | 0.836 | 0.699 | 0.506 | 0.507 | 0.643 | 0.616 | 0.514 | 0.512 |
| | Avg | **0.403** | **0.424** | 0.454 | 0.447 | 0.446 | 0.434 | 0.469 | 0.454 | 0.529 | 0.522 | 0.541 | 0.507 | 0.458 | 0.450 | 0.456 | 0.452 | 0.747 | 0.647 | 0.440 | 0.460 | 0.570 | 0.537 | 0.496 | 0.487 |
| ETTh2 | 96 | **0.283** | **0.337** | 0.297 | 0.349 | 0.288 | 0.338 | 0.302 | 0.348 | 0.745 | 0.584 | 0.400 | 0.440 | 0.340 | 0.374 | 0.333 | 0.387 | 0.707 | 0.621 | 0.358 | 0.397 | 0.476 | 0.458 | 0.346 | 0.388 |
| | 192 | **0.367** | **0.389** | 0.380 | 0.400 | 0.374 | 0.390 | 0.388 | 0.400 | 0.877 | 0.656 | 0.528 | 0.509 | 0.402 | 0.414 | 0.477 | 0.476 | 0.860 | 0.689 | 0.429 | 0.439 | 0.512 | 0.493 | 0.456 | 0.452 |
| | 336 | **0.404** | **0.421** | 0.428 | 0.432 | 0.415 | 0.426 | 0.426 | 0.433 | 1.043 | 0.731 | 0.643 | 0.571 | 0.452 | 0.452 | 0.594 | 0.541 | 1.000 | 0.744 | 0.496 | 0.487 | 0.552 | 0.551 | 0.482 | 0.486 |
| | 720 | **0.411** | **0.434** | 0.427 | 0.445 | 0.420 | 0.440 | 0.431 | 0.446 | 1.104 | 0.763 | 0.874 | 0.679 | 0.462 | 0.468 | 0.831 | 0.657 | 1.249 | 0.838 | 0.463 | 0.474 | 0.562 | 0.560 | 0.515 | 0.511 |
| | Avg | **0.366** | **0.395** | 0.383 | 0.407 | 0.374 | 0.398 | 0.387 | 0.407 | 0.942 | 0.684 | 0.611 | 0.550 | 0.414 | 0.427 | 0.559 | 0.515 | 0.954 | 0.723 | 0.437 | 0.449 | 0.526 | 0.516 | 0.450 | 0.459 |
| ECL | 96 | **0.132** | **0.228** | 0.148 | 0.240 | 0.201 | 0.281 | 0.181 | 0.270 | 0.219 | 0.314 | 0.237 | 0.329 | 0.168 | 0.272 | 0.197 | 0.282 | 0.247 | 0.345 | 0.193 | 0.308 | 0.169 | 0.273 | 0.201 | 0.317 |
| | 192 | **0.158** | **0.252** | 0.162 | 0.253 | 0.201 | 0.283 | 0.188 | 0.274 | 0.231 | 0.322 | 0.236 | 0.330 | 0.184 | 0.289 | 0.196 | 0.285 | 0.257 | 0.355 | 0.201 | 0.315 | 0.182 | 0.286 | 0.222 | 0.334 |
| | 336 | **0.168** | **0.264** | 0.178 | 0.269 | 0.215 | 0.298 | 0.204 | 0.293 | 0.246 | 0.337 | 0.249 | 0.344 | 0.198 | 0.300 | 0.209 | 0.301 | 0.269 | 0.369 | 0.214 | 0.329 | 0.200 | 0.304 | 0.231 | 0.338 |
| | 720 | **0.192** | **0.287** | 0.225 | 0.317 | 0.257 | 0.331 | 0.246 | 0.324 | 0.280 | 0.363 | 0.284 | 0.373 | 0.220 | 0.320 | 0.245 | 0.333 | 0.299 | 0.390 | 0.246 | 0.355 | 0.222 | 0.321 | 0.254 | 0.361 |
| | Avg | **0.163** | **0.258** | 0.178 | 0.270 | 0.219 | 0.298 | 0.205 | 0.290 | 0.244 | 0.334 | 0.251 | 0.344 | 0.192 | 0.295 | 0.212 | 0.300 | 0.268 | 0.365 | 0.214 | 0.327 | 0.193 | 0.296 | 0.227 | 0.338 |
| Traffic | 96 | 0.416 | 0.272 | **0.395** | **0.268** | 0.649 | 0.389 | 0.462 | 0.295 | 0.522 | 0.290 | 0.805 | 0.493 | 0.593 | 0.321 | 0.650 | 0.396 | 0.788 | 0.499 | 0.587 | 0.366 | 0.612 | 0.338 | 0.613 | 0.388 |
| | 192 | 0.436 | 0.277 | **0.417** | **0.276** | 0.601 | 0.366 | 0.466 | 0.296 | 0.530 | 0.293 | 0.756 | 0.474 | 0.617 | 0.336 | 0.598 | 0.370 | 0.789 | 0.505 | 0.604 | 0.373 | 0.613 | 0.340 | 0.616 | 0.382 |
| | 336 | 0.444 | 0.290 | **0.433** | **0.283** | 0.609 | 0.369 | 0.482 | 0.304 | 0.558 | 0.305 | 0.762 | 0.477 | 0.629 | 0.336 | 0.605 | 0.373 | 0.797 | 0.508 | 0.621 | 0.383 | 0.618 | 0.328 | 0.622 | 0.337 |
| | 720 | 0.513 | 0.316 | **0.467** | **0.302** | 0.647 | 0.387 | 0.514 | 0.322 | 0.589 | 0.328 | 0.719 | 0.449 | 0.640 | 0.350 | 0.645 | 0.394 | 0.841 | 0.523 | 0.626 | 0.382 | 0.653 | 0.355 | 0.660 | 0.408 |
| | Avg | 0.452 | 0.289 | **0.428** | **0.282** | 0.626 | 0.378 | 0.481 | 0.304 | 0.550 | 0.304 | 0.760 | 0.473 | 0.620 | 0.336 | 0.625 | 0.383 | 0.804 | 0.509 | 0.610 | 0.376 | 0.624 | 0.340 | 0.628 | 0.379 |
| Weather | 96 | **0.149** | **0.198** | 0.174 | 0.214 | 0.192 | 0.232 | 0.177 | 0.218 | 0.158 | 0.230 | 0.202 | 0.261 | 0.172 | 0.220 | 0.196 | 0.255 | 0.221 | 0.306 | 0.217 | 0.296 | 0.173 | 0.223 | 0.266 | 0.336 |
| | 192 | **0.200** | **0.243** | 0.221 | 0.254 | 0.240 | 0.271 | 0.225 | 0.259 | 0.206 | 0.277 | 0.242 | 0.298 | 0.219 | 0.261 | 0.237 | 0.296 | 0.261 | 0.340 | 0.276 | 0.336 | 0.245 | 0.285 | 0.307 | 0.367 |
| | 336 | **0.257** | **0.286** | 0.278 | 0.296 | 0.292 | 0.307 | 0.278 | 0.297 | 0.272 | 0.335 | 0.287 | 0.335 | 0.280 | 0.306 | 0.283 | 0.335 | 0.309 | 0.378 | 0.339 | 0.380 | 0.321 | 0.338 | 0.359 | 0.395 |
| | 720 | **0.334** | **0.338** | 0.358 | 0.347 | 0.364 | 0.353 | 0.354 | 0.348 | 0.398 | 0.418 | 0.351 | 0.386 | 0.365 | 0.359 | 0.345 | 0.381 | 0.377 | 0.427 | 0.403 | 0.428 | 0.414 | 0.410 | 0.419 | 0.428 |
| | Avg | **0.235** | **0.266** | 0.258 | 0.278 | 0.272 | 0.291 | 0.259 | 0.281 | 0.259 | 0.315 | 0.271 | 0.320 | 0.259 | 0.287 | 0.265 | 0.317 | 0.292 | 0.363 | 0.309 | 0.360 | 0.288 | 0.314 | 0.338 | 0.382 |
| Solar-Energy | 96 | **0.188** | **0.225** | 0.203 | 0.237 | 0.322 | 0.339 | 0.234 | 0.286 | 0.310 | 0.331 | 0.312 | 0.399 | 0.250 | 0.292 | 0.290 | 0.378 | 0.237 | 0.344 | 0.242 | 0.342 | 0.215 | 0.249 | 0.884 | 0.711 |
| | 192 | **0.229** | **0.258** | 0.233 | 0.261 | 0.359 | 0.356 | 0.267 | 0.310 | 0.734 | 0.725 | 0.339 | 0.416 | 0.296 | 0.318 | 0.320 | 0.398 | 0.280 | 0.380 | 0.285 | 0.380 | 0.254 | 0.272 | 0.834 | 0.692 |
| | 336 | **0.233** | **0.260** | 0.248 | 0.273 | 0.397 | 0.369 | 0.290 | 0.315 | 0.750 | 0.735 | 0.368 | 0.430 | 0.319 | 0.330 | 0.353 | 0.415 | 0.304 | 0.389 | 0.282 | 0.376 | 0.290 | 0.296 | 0.941 | 0.723 |
| | 720 | **0.249** | **0.272** | 0.249 | 0.275 | 0.397 | 0.356 | 0.289 | 0.317 | 0.769 | 0.765 | 0.370 | 0.425 | 0.338 | 0.337 | 0.356 | 0.413 | 0.308 | 0.388 | 0.357 | 0.427 | 0.285 | 0.295 | 0.882 | 0.717 |
| | Avg | **0.225** | **0.254** | 0.233 | 0.262 | 0.369 | 0.356 | 0.270 | 0.307 | 0.641 | 0.639 | 0.347 | 0.417 | 0.301 | 0.319 | 0.330 | 0.401 | 0.282 | 0.375 | 0.291 | 0.381 | 0.261 | 0.381 | 0.885 | 0.711 |
| Best Count | | **28** | **27** | 4 | 4 | 0 | 1 | 0 | 0 | 0 | 0 | 0 | 0 | 0 | 0 | 0 | 0 | 0 | 0 | 0 | 0 | 0 | 0 | 0 | 0 |

datasets for classification(Table 31), 4 datasets for imputation (Table 10), and 5 datasets for anomaly detection (Table 11).

**Forecasting.** We compare the forecasting performance with the forecasting length of 96, 192, 336, and 720. To make fair comparisons with baseline methods under different look back windows, we have forecasting results in both fixed and optimal back windows. The full results for forecasting with a 96 look back window are shown in Table 30. The full results with optimal look back window ranging from 96 to 512 are shown Table 34.

**Classification.** Following [112], we use 10 multivariate datasets from the UEA dataset collection [4]. The full results for classification are shown in Table 31.

**Imputation.** Imputation aims to fill in the missing data points of the time series samples. We randomly mask data points of the time series samples with mask ratios of 12.5%, 25%, 37.5%, and 50%, and then make the model predict the missing points. The full results of the imputation task are shown in Table 33.

**Anomaly detection.** Anomaly detection identifies the anomalous data points in the time series samples. We present the complete results of anomaly detection in Table 32.

Table 31: Full results for the single-task classification task. ∗. in the Transformers indicates the name of ∗former. We report the classification accuracy (%) as the result.

| Datasets / Models | Classical methods | | | RNN | | TCN | | Transformers | | | | | | | | | MLP | | Freq. | |
|---|---|---|---|---|---|---|---|---|---|---|---|---|---|---|---|---|---|---|---|---|
| | DTW | XGBoost | Rocket | LSTM | LSTNet | LSSL | TCN | Trans. | Re. | In. | Pyra. | Auto. | Station. | FED. | ETS. | Flow. | DLinear | LightTS | TimesNet | UniTS-*ST* |
| | [6] | [15] | [24] | [41] | [53] | [39] | [30] | [104] | [51] | [127] | [65] | [114] | [69] | [128] | [111] | [113] | [119] | [122] | [112] | (Ours) |
| EthanolConcentration | 32.3 | 43.7 | 45.2 | 32.3 | 39.9 | 31.1 | 28.9 | 32.7 | 31.9 | 31.6 | 30.8 | 31.6 | 32.7 | 31.2 | 28.1 | 33.8 | 32.6 | 29.7 | 35.7 | 37.6 |
| FaceDetection | 52.9 | 63.3 | 64.7 | 57.7 | 65.7 | 66.7 | 52.8 | 67.3 | 68.6 | 67.0 | 65.7 | 68.4 | 68.0 | 66.0 | 66.3 | 67.6 | 68.0 | 67.5 | 68.6 | 70.5 |
| Handwriting | 28.6 | 15.8 | 58.8 | 15.2 | 25.8 | 24.6 | 53.3 | 32.0 | 27.4 | 32.8 | 29.4 | 36.7 | 31.6 | 28.0 | 32.5 | 33.8 | 27.0 | 26.1 | 32.1 | 29.7 |
| Heartbeat | 71.7 | 73.2 | 75.6 | 72.2 | 77.1 | 72.7 | 75.6 | 76.1 | 77.1 | 80.5 | 75.6 | 74.6 | 73.7 | 73.7 | 71.2 | 77.6 | 75.1 | 75.1 | 78.0 | 80.0 |
| JapaneseVowels | 94.9 | 86.5 | 96.2 | 79.7 | 98.1 | 98.4 | 98.9 | 98.7 | 97.8 | 98.9 | 98.4 | 96.2 | 99.2 | 98.4 | 95.9 | 98.9 | 96.2 | 96.2 | 98.4 | 97.8 |
| PEMS-SF | 71.1 | 98.3 | 75.1 | 39.9 | 86.7 | 86.1 | 68.8 | 82.1 | 82.7 | 81.5 | 83.2 | 82.7 | 87.3 | 80.9 | 86.0 | 83.8 | 75.1 | 88.4 | 89.6 | 93.1 |
| SelfRegulationSCP1 | 77.7 | 84.6 | 90.8 | 68.9 | 84.0 | 90.8 | 84.6 | 92.2 | 90.4 | 90.1 | 88.1 | 84.0 | 89.4 | 88.7 | 89.6 | 92.5 | 87.3 | 89.8 | 91.8 | 93.9 |
| SelfRegulationSCP2 | 53.9 | 48.9 | 53.3 | 46.6 | 52.8 | 52.2 | 55.6 | 53.9 | 56.7 | 53.3 | 53.3 | 50.6 | 57.2 | 54.4 | 55.0 | 56.1 | 50.5 | 51.1 | 57.2 | 61.1 |
| SpokenArabicDigits | 96.3 | 69.6 | 71.2 | 31.9 | 100.0 | 100.0 | 95.6 | 98.4 | 97.0 | 100.0 | 99.6 | 100.0 | 100.0 | 100.0 | 100.0 | 98.8 | 81.4 | 100.0 | 99.0 | 98.9 |
| UWaveGestureLibrary | 90.3 | 75.9 | 94.4 | 41.2 | 87.8 | 85.9 | 88.4 | 85.6 | 85.6 | 85.6 | 83.4 | 85.9 | 87.5 | 85.3 | 85.0 | 86.6 | 82.1 | 80.3 | 85.3 | 87.8 |
| Average Accuracy | 67.0 | 66.0 | 72.5 | 48.6 | 71.8 | 70.9 | 70.3 | 71.9 | 71.5 | 72.1 | 70.8 | 71.1 | 72.7 | 70.7 | 71.0 | 73.0 | 67.5 | 70.4 | 73.6 | **75.0** |

Table 32: Full results for the anomaly detection task. The P, R and F1 represent the precision, recall and F1-score (%) respectively. F1-score is the harmonic mean of precision and recall.

| Datasets | | SMD | | | MSL | | | SMAP | | | SWaT | | | PSM | | | Avg F1↑ |
|---|---|---|---|---|---|---|---|---|---|---|---|---|---|---|---|---|---|---|
| Metrics | | P↑ | R↑ | F1↑ | P↑ | R↑ | F1↑ | P↑ | R↑ | F1↑ | P↑ | R↑ | F1↑ | P↑ | R↑ | F1↑ | (%) |
| LSTM | [41] | 78.52 | 65.47 | 71.41 | 78.04 | 86.22 | 81.93 | 91.06 | 57.49 | 70.48 | 78.06 | 91.72 | 84.34 | 69.24 | 99.53 | 81.67 | 77.97 |
| Transformer | [104] | 83.58 | 76.13 | 79.56 | 71.57 | 87.37 | 78.68 | 89.37 | 57.12 | 69.70 | 68.84 | 96.53 | 80.37 | 62.75 | 96.56 | 76.07 | 76.88 |
| LogTrans | [57] | 83.46 | 70.13 | 76.21 | 73.05 | 87.37 | 79.57 | 89.15 | 57.59 | 69.97 | 68.67 | 97.32 | 80.52 | 63.06 | 98.00 | 76.74 | 76.60 |
| TCN | [30] | 84.06 | 79.07 | 81.49 | 75.11 | 82.44 | 78.60 | 86.90 | 59.23 | 70.45 | 76.59 | 95.71 | 85.09 | 54.59 | 99.77 | 70.57 | 77.24 |
| Reformer | [51] | 82.58 | 69.24 | 75.32 | 85.51 | 83.31 | 84.40 | 90.91 | 57.44 | 70.40 | 72.50 | 96.53 | 82.80 | 59.93 | 95.38 | 73.61 | 77.31 |
| Informer | [127] | 86.60 | 77.23 | 81.65 | 81.77 | 86.48 | 84.06 | 90.11 | 57.13 | 69.92 | 70.29 | 96.75 | 81.43 | 64.27 | 96.33 | 77.10 | 78.83 |
| Anomaly* | [116] | 88.91 | 82.23 | 85.49 | 79.61 | 87.37 | 83.31 | 91.85 | 58.11 | 71.18 | 72.51 | 97.32 | 83.10 | 68.35 | 94.72 | 79.40 | 80.50 |
| Pyraformer | [65] | 85.61 | 80.61 | 83.04 | 83.81 | 85.93 | 84.86 | 92.54 | 57.71 | 71.09 | 87.92 | 96.00 | 91.78 | 71.67 | 96.02 | 82.08 | 82.57 |
| Autoformer | [114] | 88.06 | 82.35 | 85.11 | 77.27 | 80.92 | 79.05 | 90.40 | 58.62 | 71.12 | 89.85 | 95.81 | 92.74 | 99.08 | 88.15 | 93.29 | 84.26 |
| LSSL | [39] | 78.51 | 65.32 | 71.31 | 77.55 | 88.18 | 82.53 | 89.43 | 53.43 | 66.90 | 79.05 | 93.72 | 85.76 | 66.02 | 92.93 | 77.20 | 76.74 |
| Station. | [69] | 88.33 | 81.21 | 84.62 | 68.55 | 89.14 | 77.50 | 89.37 | 59.02 | 71.09 | 68.03 | 96.75 | 79.88 | 97.82 | 96.76 | 97.29 | 82.08 |
| DLinear | [119] | 83.62 | 71.52 | 77.10 | 84.34 | 85.42 | 84.88 | 92.32 | 55.41 | 69.26 | 80.91 | 95.30 | 87.52 | 98.28 | 89.26 | 93.55 | 82.46 |
| ETSformer | [111] | 87.44 | 79.23 | 83.13 | 85.13 | 84.93 | **85.03** | 92.25 | 55.75 | 69.50 | 90.02 | 80.36 | 84.91 | 99.31 | 85.28 | 91.76 | 82.87 |
| LightTS | [122] | 87.10 | 78.42 | 82.53 | 82.40 | 75.78 | 78.95 | 92.58 | 55.27 | 69.21 | 91.98 | 94.72 | **93.33** | 98.37 | 95.97 | 97.15 | 84.23 |
| FEDformer | [128] | 87.95 | 82.39 | 85.08 | 77.14 | 80.07 | 78.57 | 90.47 | 58.10 | 70.76 | 90.17 | 96.42 | 93.19 | 97.31 | 97.16 | 97.23 | 84.97 |
| TimesNet* | [112] | 87.95 | 81.54 | 84.62 | 89.55 | 75.29 | 81.80 | 90.14 | 56.56 | 69.50 | 90.76 | 95.35 | 93.00 | 98.50 | 96.29 | 97.38 | 85.26 |
| **UniTS-*ST*** | Ours | 89.32 | 86.90 | **88.09** | 89.91 | 77.68 | 83.46 | 93.37 | 76.02 | **83.80** | 92.37 | 94.17 | 93.26 | 98.62 | 96.28 | **97.43** | **89.21** |

For fair comparisons, we follow the settings of [112] to only use reconstruction error for Anomaly Transformer.

TimesNet are reproduced from the `https://github.com/thuml/Time-Series-Library` to ensure fair comparisons.

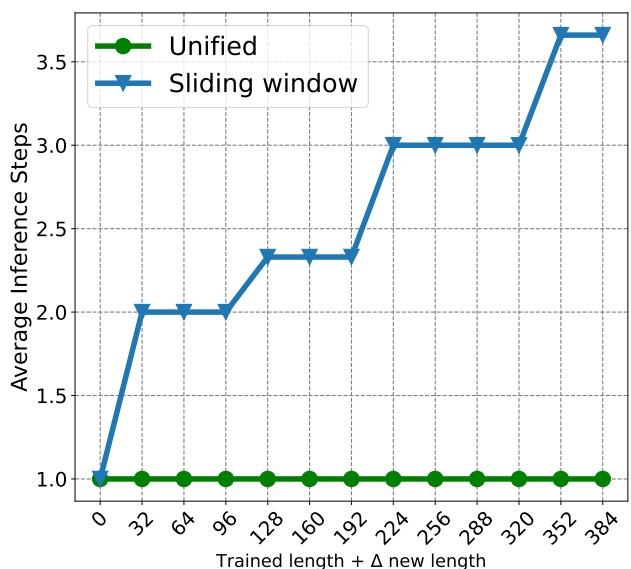

Figure 6: The comparison of average inference steps between our direct multi-step inference and multi-step sliding window-based inference for zero-shot forecasting on new lengths.

Table 33: Full results for the imputation task. We randomly mask 12.5%, 25%, 37.5% and 50% time points to compare the model performance under different missing degrees.

| Models | | UniTS-*ST* (Ours) | | TimesNet [112] | | ETS. [111] | | LightTS* [122] | | DLinear* [119] | | FED. [128] | | Stationary [69] | | Auto. [114] | | Pyra. [65] | | In. [127] | | LogTrans [57] | | Re. [51] | | LSTM [41] | | TCN [30] | | LSSL [39] | |
|---|---|---|---|---|---|---|---|---|---|---|---|---|---|---|---|---|---|---|---|---|---|---|---|---|---|---|---|---|---|---|---|
| Mask Ratio | | MSE | MAE | MSE | MAE | MSE | MAE | MSE | MAE | MSE | MAE | MSE | MAE | MSE | MAE | MSE | MAE | MSE | MAE | MSE | MAE | MSE | MAE | MSE | MAE | MSE | MAE | MSE | MAE | MSE | MAE |
| ETTm1 | 12.5% | **0.015** | **0.079** | 0.019 | 0.092 | 0.067 | 0.188 | 0.075 | 0.180 | 0.058 | 0.162 | 0.035 | 0.135 | 0.026 | 0.107 | 0.034 | 0.124 | 0.670 | 0.541 | 0.047 | 0.155 | 0.041 | 0.141 | 0.032 | 0.126 | 0.974 | 0.780 | 0.510 | 0.493 | 0.101 | 0.231 |
| | 25% | **0.017** | **0.082** | 0.023 | 0.101 | 0.096 | 0.229 | 0.093 | 0.206 | 0.080 | 0.193 | 0.052 | 0.166 | 0.032 | 0.119 | 0.046 | 0.144 | 0.689 | 0.553 | 0.063 | 0.180 | 0.044 | 0.144 | 0.042 | 0.146 | 1.032 | 0.807 | 0.518 | 0.500 | 0.106 | 0.235 |
| | 37.5% | **0.019** | **0.088** | 0.029 | 0.111 | 0.133 | 0.271 | 0.113 | 0.231 | 0.103 | 0.219 | 0.069 | 0.191 | 0.039 | 0.131 | 0.057 | 0.161 | 0.737 | 0.581 | 0.079 | 0.200 | 0.052 | 0.158 | 0.063 | 0.182 | 0.999 | 0.792 | 0.516 | 0.499 | 0.116 | 0.246 |
| | 50% | **0.024** | **0.097** | 0.036 | 0.124 | 0.186 | 0.323 | 0.134 | 0.255 | 0.132 | 0.248 | 0.089 | 0.218 | 0.047 | 0.145 | 0.067 | 0.174 | 0.770 | 0.605 | 0.093 | 0.218 | 0.063 | 0.173 | 0.082 | 0.208 | 0.952 | 0.763 | 0.519 | 0.496 | 0.129 | 0.260 |
| | Avg | **0.019** | **0.087** | 0.027 | 0.107 | 0.120 | 0.253 | 0.104 | 0.218 | 0.093 | 0.206 | 0.062 | 0.177 | 0.036 | 0.126 | 0.051 | 0.150 | 0.717 | 0.570 | 0.071 | 0.188 | 0.050 | 0.154 | 0.055 | 0.166 | 0.989 | 0.786 | 0.516 | 0.497 | 0.113 | 0.254 |
| ETTh1 | 12.5% | **0.032** | **0.118** | 0.057 | 0.159 | 0.126 | 0.263 | 0.240 | 0.345 | 0.151 | 0.267 | 0.070 | 0.190 | 0.060 | 0.165 | 0.074 | 0.182 | 0.857 | 0.609 | 0.114 | 0.234 | 0.229 | 0.330 | 0.074 | 0.194 | 1.265 | 0.896 | 0.599 | 0.554 | 0.422 | 0.461 |
| | 25% | **0.036** | **0.126** | 0.069 | 0.178 | 0.169 | 0.304 | 0.265 | 0.364 | 0.180 | 0.292 | 0.106 | 0.236 | 0.080 | 0.189 | 0.090 | 0.203 | 0.829 | 0.672 | 0.140 | 0.262 | 0.207 | 0.323 | 0.102 | 0.227 | 1.262 | 0.883 | 0.610 | 0.567 | 0.412 | 0.456 |
| | 37.5% | **0.047** | **0.142** | 0.084 | 0.196 | 0.220 | 0.347 | 0.296 | 0.382 | 0.215 | 0.318 | 0.124 | 0.258 | 0.102 | 0.212 | 0.109 | 0.222 | 0.830 | 0.675 | 0.174 | 0.293 | 0.210 | 0.328 | 0.135 | 0.261 | 1.200 | 0.867 | 0.628 | 0.577 | 0.421 | 0.461 |
| | 50% | **0.060** | **0.160** | 0.102 | 0.215 | 0.293 | 0.402 | 0.334 | 0.404 | 0.257 | 0.347 | 0.165 | 0.299 | 0.133 | 0.240 | 0.137 | 0.248 | 0.854 | 0.691 | 0.215 | 0.325 | 0.230 | 0.348 | 0.179 | 0.298 | 1.174 | 0.849 | 0.648 | 0.587 | 0.443 | 0.473 |
| | Avg | **0.043** | **0.136** | 0.078 | 0.187 | 0.202 | 0.329 | 0.284 | 0.373 | 0.201 | 0.306 | 0.117 | 0.246 | 0.094 | 0.201 | 0.103 | 0.214 | 0.842 | 0.682 | 0.161 | 0.279 | 0.219 | 0.332 | 0.122 | 0.245 | 1.225 | 0.873 | 0.621 | 0.571 | 0.424 | 0.481 |
| Electricity | 12.5% | **0.031** | **0.112** | 0.085 | 0.202 | 0.196 | 0.321 | 0.102 | 0.229 | 0.092 | 0.214 | 0.107 | 0.237 | 0.093 | 0.210 | 0.089 | 0.210 | 0.297 | 0.383 | 0.218 | 0.326 | 0.164 | 0.296 | 0.190 | 0.308 | 0.277 | 0.366 | 0.621 | 0.620 | 0.217 | 0.341 |
| | 25% | **0.035** | **0.119** | 0.089 | 0.206 | 0.207 | 0.332 | 0.121 | 0.252 | 0.118 | 0.247 | 0.120 | 0.251 | 0.097 | 0.214 | 0.096 | 0.220 | 0.294 | 0.380 | 0.219 | 0.326 | 0.169 | 0.299 | 0.197 | 0.312 | 0.281 | 0.369 | 0.559 | 0.585 | 0.219 | 0.341 |
| | 37.5% | **0.040** | **0.128** | 0.094 | 0.213 | 0.219 | 0.344 | 0.141 | 0.273 | 0.144 | 0.276 | 0.136 | 0.266 | 0.102 | 0.220 | 0.104 | 0.229 | 0.296 | 0.381 | 0.222 | 0.328 | 0.178 | 0.305 | 0.203 | 0.315 | 0.275 | 0.364 | 0.567 | 0.588 | 0.223 | 0.343 |
| | 50% | **0.046** | **0.138** | 0.100 | 0.221 | 0.235 | 0.357 | 0.160 | 0.293 | 0.175 | 0.305 | 0.158 | 0.284 | 0.108 | 0.228 | 0.113 | 0.239 | 0.299 | 0.383 | 0.228 | 0.331 | 0.187 | 0.312 | 0.210 | 0.319 | 0.273 | 0.361 | 0.581 | 0.597 | 0.229 | 0.347 |
| | Avg | **0.038** | **0.124** | 0.092 | 0.210 | 0.214 | 0.339 | 0.131 | 0.262 | 0.132 | 0.260 | 0.130 | 0.259 | 0.100 | 0.218 | 0.101 | 0.225 | 0.297 | 0.382 | 0.222 | 0.328 | 0.175 | 0.303 | 0.200 | 0.313 | 0.277 | 0.365 | 0.582 | 0.597 | 0.222 | 0.293 |
| Weather | 12.5% | **0.025** | **0.041** | 0.025 | 0.045 | 0.057 | 0.141 | 0.047 | 0.101 | 0.039 | 0.084 | 0.041 | 0.107 | 0.027 | 0.051 | 0.026 | 0.047 | 0.140 | 0.220 | 0.037 | 0.093 | 0.037 | 0.072 | 0.031 | 0.076 | 0.296 | 0.379 | 0.176 | 0.287 | 0.036 | 0.095 |
| | 25% | **0.026** | **0.044** | 0.029 | 0.052 | 0.065 | 0.155 | 0.052 | 0.111 | 0.048 | 0.103 | 0.064 | 0.163 | 0.029 | 0.056 | 0.030 | 0.054 | 0.147 | 0.229 | 0.042 | 0.100 | 0.038 | 0.074 | 0.035 | 0.082 | 0.327 | 0.409 | 0.187 | 0.293 | 0.042 | 0.104 |
| | 37.5% | **0.027** | **0.045** | 0.031 | 0.057 | 0.081 | 0.180 | 0.058 | 0.121 | 0.057 | 0.117 | 0.107 | 0.229 | 0.033 | 0.062 | 0.032 | 0.060 | 0.156 | 0.240 | 0.049 | 0.111 | 0.039 | 0.078 | 0.040 | 0.091 | 0.406 | 0.463 | 0.172 | 0.281 | 0.047 | 0.112 |
| | 50% | **0.029** | **0.049** | 0.034 | 0.062 | 0.102 | 0.207 | 0.065 | 0.133 | 0.066 | 0.134 | 0.183 | 0.312 | 0.037 | 0.068 | 0.037 | 0.067 | 0.164 | 0.249 | 0.053 | 0.114 | 0.042 | 0.082 | 0.046 | 0.099 | 0.431 | 0.483 | 0.195 | 0.303 | 0.054 | 0.123 |
| | Avg | **0.026** | **0.045** | 0.030 | 0.054 | 0.076 | 0.171 | 0.055 | 0.117 | 0.052 | 0.110 | 0.099 | 0.203 | 0.032 | 0.059 | 0.031 | 0.057 | 0.152 | 0.235 | 0.045 | 0.104 | 0.039 | 0.076 | 0.038 | 0.087 | 0.365 | 0.434 | 0.183 | 0.291 | 0.045 | 0.108 |
| Best Count | | **16** | **16** | 0 | 0 | 0 | 0 | 0 | 0 | 0 | 0 | 0 | 0 | 0 | 0 | 0 | 0 | 0 | 0 | 0 | 0 | 0 | 0 | 0 | 0 | 0 | 0 | 0 | 0 | 0 | 0 |

## L    Additional Results: Multi-task versus Single-task Learning

To verify the gap between multi-task and single-task learning under fair comparisons, we conduct a experiment to train the single-task models using the same hyper-parameters as the multi-task co-training. As shown in Table 35, multi-task learning achieves stronger performance on both forecasting and classification tasks. Interestingly, under the same hyper-parameters, some classification models fail to converge in the single-task setting, whereas the multi-task model does not have this issue, demonstrating the robustness of multi-task training.

Table 34: Full results of the long-term forecasting task where model is separately trained on each dataset. The input time series sequence length is set ranging from 96 to 512 to ensure fair comparisons. Baseline results are obtained from their original papers. "Extra Training Data" indicates whether the model uses training data beyond just time series data. "Multi-task Support" refers to whether the model can handle multiple tasks or is focused solely on a single task. Gray color represents LLM-reprogrammed models that reprogram pre-trained LLMs to time series domain and needs dataset/task-specific modules. For the best count, we only consider the purely time series models.

| Models | | UniTS-ST (Ours) | | MOMENT [36] | | TSMixer [14] | | TEMPO [10] | | TIME-LLM [47] | | LLM4TS [12] | | TEST [97] | | GPT4TS [129] | |
|---|---|---|---|---|---|---|---|---|---|---|---|---|---|---|---|---|---|
| Metric | | MSE | MAE | MSE | MAE | MSE | MAE | MSE | MAE | MSE | MAE | MSE | MAE | MSE | MAE | MSE | MAE |
| ETTm1 | 96 | **0.278** | **0.338** | 0.293 | 0.349 | 0.285 | 0.339 | 0.438 | 0.424 | 0.272 | 0.334 | 0.360 | 0.388 | 0.293 | 0.346 | 0.292 | 0.346 |
| | 192 | **0.319** | **0.364** | - | - | 0.327 | 0.365 | 0.461 | 0.432 | 0.310 | 0.358 | 0.386 | 0.401 | 0.332 | 0.369 | 0.332 | 0.372 |
| | 336 | **0.354** | 0.386 | - | - | 0.356 | **0.382** | 0.515 | 0.467 | 0.352 | 0.384 | 0.415 | 0.417 | 0.368 | 0.392 | 0.366 | 0.394 |
| | 720 | **0.397** | 0.416 | 0.405 | 0.416 | 0.419 | **0.414** | 0.591 | 0.509 | 0.383 | 0.411 | 0.470 | 0.445 | 0.418 | 0.420 | 0.417 | 0.421 |
| | Avg | **0.337** | 0.376 | 0.349 | 0.383 | 0.347 | **0.375** | 0.501 | 0.458 | 0.329 | 0.372 | 0.408 | 0.413 | 0.353 | 0.382 | 0.352 | 0.383 |
| ETTm2 | 96 | 0.167 | 0.258 | 0.181 | 0.269 | **0.163** | **0.252** | 0.185 | 0.267 | 0.161 | 0.253 | 0.184 | 0.265 | - | - | 0.173 | 0.262 |
| | 192 | 0.222 | 0.295 | - | - | **0.216** | **0.290** | 0.243 | 0.304 | 0.219 | 0.293 | 0.240 | 0.301 | - | - | 0.229 | 0.301 |
| | 336 | 0.270 | 0.325 | - | - | **0.268** | **0.324** | 0.309 | 0.345 | 0.271 | 0.329 | 0.294 | 0.337 | - | - | 0.286 | 0.341 |
| | 720 | **0.358** | **0.380** | 0.366 | 0.388 | 0.420 | 0.422 | 0.386 | 0.395 | 0.352 | 0.379 | 0.386 | 0.393 | - | - | 0.378 | 0.401 |
| | Avg | **0.254** | **0.315** | 0.274 | 0.329 | 0.267 | 0.322 | 0.281 | 0.328 | 0.251 | 0.314 | 0.276 | 0.324 | - | - | 0.284 | 0.339 |
| ETTh1 | 96 | **0.360** | 0.396 | 0.387 | 0.410 | 0.361 | **0.392** | 0.400 | 0.406 | 0.362 | 0.392 | 0.371 | 0.394 | 0.372 | 0.400 | 0.376 | 0.397 |
| | 192 | **0.401** | **0.416** | - | - | 0.404 | 0.418 | 0.426 | 0.421 | 0.398 | 0.418 | 0.403 | 0.412 | 0.414 | 0.422 | 0.416 | 0.418 |
| | 336 | 0.425 | 0.439 | - | - | **0.420** | **0.431** | 0.441 | 0.430 | 0.430 | 0.427 | 0.420 | 0.422 | 0.422 | 0.437 | 0.442 | 0.433 |
| | 720 | **0.434** | **0.454** | 0.454 | 0.472 | 0.463 | 0.472 | 0.443 | 0.451 | 0.442 | 0.457 | 0.422 | 0.444 | 0.447 | 0.467 | 0.477 | 0.456 |
| | Avg | **0.405** | **0.426** | 0.421 | 0.441 | 0.412 | 0.428 | 0.428 | 0.427 | 0.408 | 0.424 | 0.404 | 0.418 | 0.414 | 0.431 | 0.428 | 0.426 |
| ETTh2 | 96 | 0.277 | 0.346 | 0.288 | 0.345 | **0.274** | **0.341** | 0.301 | 0.353 | 0.268 | 0.328 | 0.269 | 0.332 | 0.275 | 0.338 | 0.285 | 0.342 |
| | 192 | **0.325** | **0.382** | - | - | 0.339 | 0.385 | 0.355 | 0.389 | 0.329 | 0.375 | 0.328 | 0.377 | 0.340 | 0.379 | 0.354 | 0.389 |
| | 336 | **0.347** | **0.398** | - | - | 0.361 | 0.406 | 0.379 | 0.408 | 0.368 | 0.409 | 0.353 | 0.396 | 0.329 | 0.381 | 0.373 | 0.407 |
| | 720 | **0.373** | **0.420** | 0.403 | 0.439 | 0.445 | 0.470 | 0.409 | 0.440 | 0.372 | 0.420 | 0.383 | 0.425 | 0.381 | 0.423 | 0.406 | 0.441 |
| | Avg | **0.331** | **0.387** | 0.346 | 0.392 | 0.355 | 0.401 | 0.361 | 0.398 | 0.334 | 0.383 | 0.333 | 0.383 | 0.331 | 0.380 | 0.355 | 0.395 |
| ECL | 96 | **0.130** | **0.224** | 0.138 | 0.242 | 0.131 | 0.229 | 0.178 | 0.276 | 0.131 | 0.224 | 0.128 | 0.223 | 0.132 | 0.223 | 0.139 | 0.238 |
| | 192 | **0.147** | **0.242** | - | - | 0.151 | 0.246 | 0.198 | 0.293 | 0.152 | 0.241 | 0.146 | 0.240 | 0.158 | 0.241 | 0.153 | 0.251 |
| | 336 | **0.160** | **0.260** | - | - | 0.161 | 0.261 | 0.209 | 0.309 | 0.160 | 0.261 | 0.163 | 0.258 | 0.163 | 0.260 | 0.169 | 0.266 |
| | 720 | **0.188** | **0.284** | 0.211 | 0.305 | 0.197 | 0.293 | 0.279 | 0.355 | 0.192 | 0.298 | 0.200 | 0.292 | 0.199 | 0.291 | 0.206 | 0.297 |
| | Avg | **0.156** | **0.253** | 0.175 | 0.274 | 0.160 | 0.257 | 0.216 | 0.308 | 0.159 | 0.253 | 0.159 | 0.253 | 0.163 | 0.253 | 0.167 | 0.263 |
| Traffic | 96 | 0.370 | 0.255 | 0.391 | 0.282 | 0.376 | 0.264 | 0.476 | 0.343 | 0.362 | 0.248 | 0.372 | 0.259 | 0.407 | 0.282 | 0.388 | 0.282 |
| | 192 | 0.390 | **0.263** | - | - | 0.397 | 0.277 | 0.496 | 0.355 | 0.374 | 0.247 | 0.391 | 0.265 | 0.423 | 0.287 | 0.412 | 0.290 |
| | 336 | 0.415 | **0.268** | - | - | **0.413** | 0.290 | 0.503 | 0.356 | 0.385 | 0.271 | 0.405 | 0.275 | 0.430 | 0.296 | 0.412 | 0.294 |
| | 720 | 0.461 | 0.326 | 0.450 | 0.310 | **0.444** | **0.306** | 0.538 | 0.376 | 0.430 | 0.288 | 0.437 | 0.292 | 0.463 | 0.315 | 0.450 | 0.312 |
| | Avg | 0.409 | **0.278** | 0.421 | 0.296 | **0.408** | 0.284 | 0.503 | 0.358 | 0.388 | 0.264 | 0.401 | 0.273 | 0.431 | 0.295 | 0.414 | 0.295 |
| Weather | 96 | **0.140** | **0.192** | 0.154 | 0.209 | 0.145 | 0.198 | 0.211 | 0.254 | 0.147 | 0.201 | 0.147 | 0.196 | 0.150 | 0.202 | 0.162 | 0.212 |
| | 192 | **0.185** | **0.237** | - | - | 0.191 | 0.242 | 0.254 | 0.298 | 0.189 | 0.234 | 0.191 | 0.238 | 0.198 | 0.246 | 0.204 | 0.248 |
| | 336 | **0.234** | **0.278** | - | - | 0.242 | 0.280 | 0.292 | 0.332 | 0.262 | 0.299 | 0.241 | 0.277 | 0.245 | 0.286 | 0.254 | 0.286 |
| | 720 | **0.306** | **0.330** | 0.315 | 0.336 | 0.320 | 0.336 | 0.370 | 0.379 | 0.304 | 0.316 | 0.313 | 0.329 | 0.324 | 0.342 | 0.326 | 0.337 |
| | Avg | **0.216** | **0.259** | 0.235 | 0.273 | 0.225 | 0.264 | 0.282 | 0.316 | 0.226 | 0.258 | 0.223 | 0.260 | 0.229 | 0.269 | 0.237 | 0.271 |
| Best Count | | 21/28 | 19/28 | 0/28 | 0/28 | 7/28 | 9/28 | - | - | - | - | - | - | - | - | - | - |
| Extra Training Data | | No | | No | | No | | Yes | | Yes | | Yes | | Yes | | Yes | |
| Multi-task Support | | Yes | | No | | No | | No | | No | | No | | No | | No | |

Table 35: Compare UNITS trained by multi-task learning with that trained by single-task learning under same hyper-parameters.

| UNITS | $\text{Acc}_{Avg}\uparrow$ (Classification) | $\text{MSE}_{Avg}\downarrow$ (Forecasting) |
|---|---|---|
| **Multi-task** | 81.6% | 0.439 |
| **Single-task** | 65.3% | 0.464 |

# M   Limitations and Future Directions

The datasets collected by this work do not yet cover all available time series datasets, such as some of the univariate datasets in UCR dataset collections [23] and the more physiologic time series signals from PhysioNet [34]. We will explore using larger dataset collections to further improve UNITS.

UNITS primarily aims to unify predictive and generative tasks within a single multi-task model. We demonstrate this by showcasing its adaptability to new data and tasks through prompt learning and few-shot learning. While adapting to new time series data differs fundamentally from generalizing to entirely new data, we will further explore UNITS's generalization ability for zero-shot learning.

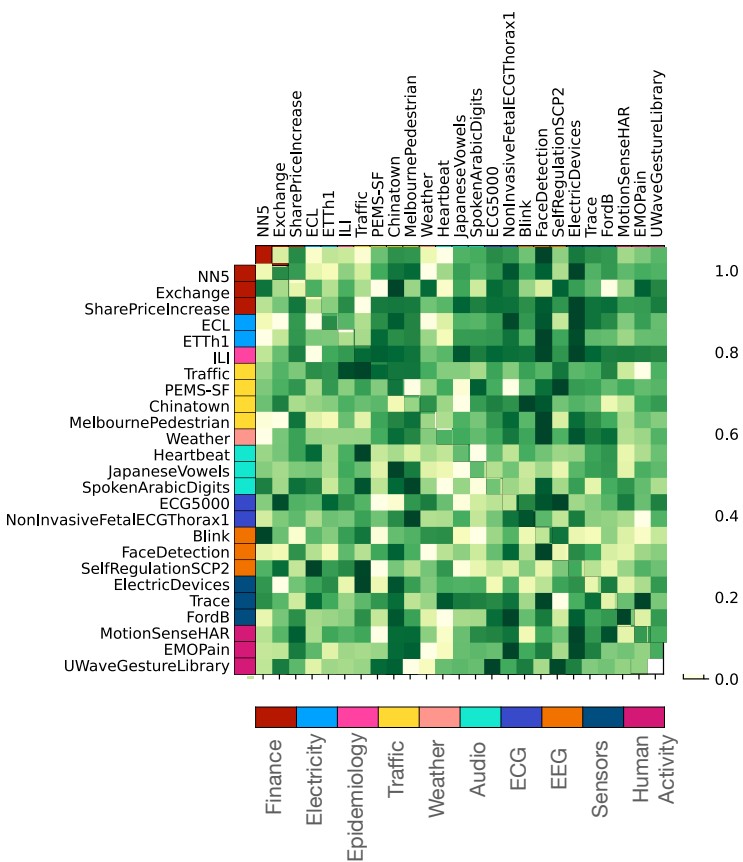

Figure 7: The similarity of prompt tokens among datasets.

# N Impact Statement

This paper focuses on analyzing time series sequences from various domains and introduces a versatile machine-learning approach designed for this purpose. While our research has numerous potential societal impacts, we believe none require specific emphasis in this context.

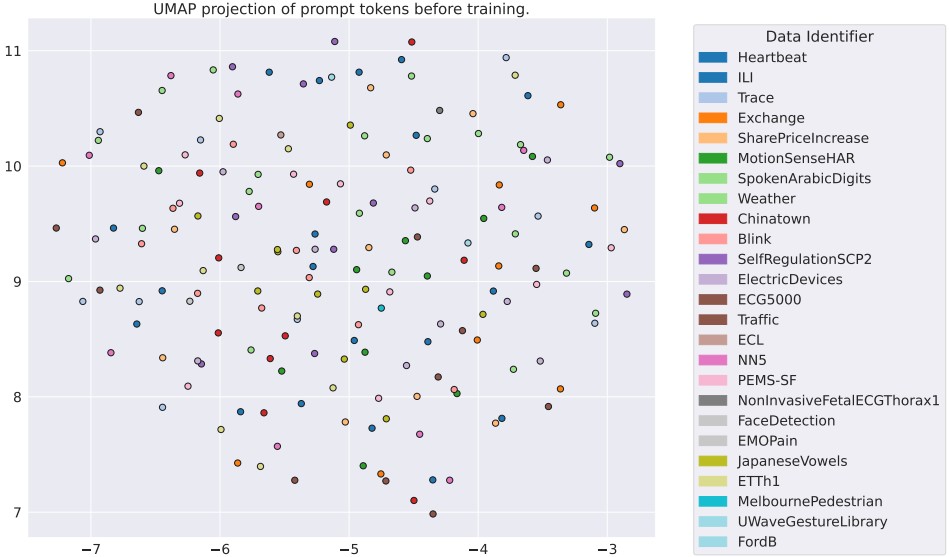

Figure 8: UMAP of untrained prompt tokens in UNITS. This plot illustrates that there is no significant organization (clustering) of prompt tokens prior to UNITS training.

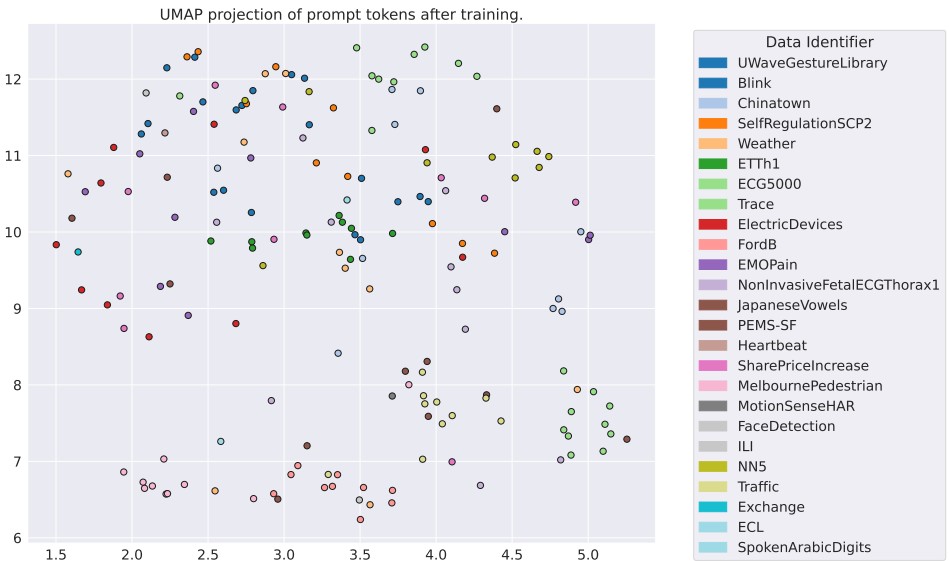

Figure 9: UMAP of trained prompt tokens in UNITS. Unlike Figure 8 above, this plot illustrates the meaningful organization (clustering) of prompt tokens by dataset domain category when trained by UNITS.

