# OpenReview forum: "UniTS: A Unified Multi-Task Time Series Model"
_NeurIPS.cc/2024/Conference — NeurIPS 2024 poster_

### Official Review · Reviewer_bSMF · 2024-07-09

**Soundness:** 3
**Presentation:** 3
**Contribution:** 2
**Rating:** 6
**Confidence:** 4

**Summary:**

This paper presents UNITS, a multi-task time series model that uses task tokenization to express predictive and generative tasks within a single model. UNITS can process heterogeneous time series with diverse variables and lengths without modifying the network architecture. Experiments show that UNITS demonstrates effective few-shot and prompt learning capabilities when evaluated on new data domains and tasks.

**Strengths:**

1. The paper  is well-written and easy to follow.
2. The paper introduces a unified multi-task time series model that handles a broad range of time series tasks.
3. The proposed architecture achieve state-of-the-art performance on various time series tasks and can be adapted to new tasks via parameter-efficient prompt learning.

**Weaknesses:**

Although the idea of a multi-task model is somewhat novel, it seems that the UNITS on Single-Task Learning outperforms Multi-Task Learning.

**Questions:**

1. Is the experiment setting in Section 5.2 consistent with that in Section 5.1? Does GPT4TS also use the same fully supervised multi-task training as UNITS?

**Limitations:**

yes

---

> ### Author Rebuttal · Authors · 2024-08-07
>
> # Response to Reviewer bSMF Part I (Part I of II)
> Thank you for your helpful feedback\! We appreciate your acknowledgment of the quality and novelty of our method and the state-of-the-art performance achieved by UniTS. Below, we address each of your concerns, provide further details, and present new experiments.
>
> We hope our responses have addressed your concerns and kindly request you to consider raising your score. If you still have questions after reviewing our responses, we would greatly appreciate any further guidance on how we can improve our work to meet your expectations. Thank you again\!
>
> ---
>
> ### W1: It seems UniTS on Single-Task Learning outperforms Multi-Task Learning.
>
> Thank you for pointing this out. We address your concerns in two parts.
>
> **First**, **we have added new experiments to show that multi-task learning of UniTS outperforms single-task learning under the same hyperparameters.** Following common practice \[ICLR2024itrans, ICLR2023patch, ICLR2023times\], results of single-task training are obtained by training the model on each dataset by selecting optimal hyperparameters for each dataset. However, the multi-task model can only use a single set of hyperparameters that are the same across all tasks/datasets. To ensure fair comparisons and to answer your question, we conducted a new experiment, where we trained the single-task models using the same hyperparameters as the multi-task co-training. Results show in the table below, multi-task learning achieves stronger performance on both forecasting and classification tasks. Interestingly, under the same hyperparameters, some classification models fail to converge in the single-task setting, whereas the multi-task model does not have this issue, demonstrating the robustness of multi-task training.
> | UniTS | Acc↑ (Classification) | MSE↓ (Forecasting) |
> | :---- | :---- | :---- |
> | **Multi-task** (same hyperparameters) | 81.6% | 0.439 |
> | **Single-task** (same hyperparameters) | 65.3% | 0.464 |
>
> **Second, we show the advantage of multi-task learning over single-task learning**. The advantages of multi-task learning in UniTS over single-task learning are: 1\) enhanced performance without requiring task-specific hyperparameter tuning, and 2\) improved generalization capabilities, including few-shot and zero-shot learning.
>
> - **Multi-task UniTS outperforms hyperparameters-tuned single-task DLinear:** We also compare the multi-task performance of UniTS with a popular single-task model DLinear, as summarized in the table below (full results in Tables 1 and 2, page 7 and 8). Despite DLinear being able to optimize and select dataset-specific hyperparameters for each dataset, UniTS, co-trained in a multi-task setting, consistently outperforms DLinear. Specifically, across 32 experiments (rows in the table below) across 16 datasets, we find that UniTS outperforms DLinear in 27 out of 32 experiments, showing that multi-task learning across data domains is an effective approach.
> | Best count on 16 datasets using MSE/MAE metrics | UniTS (Multi-task) | DLinear (Single-task) |
> | :---- | :---- | :---- |
> | One-model | **Yes** | No |
> | Best Count | **27/32** | 5/32 |
>
>  - **Multi-task UniTS introduces new abilities beyond single-task models**. Using a single task model is insufficient to handle tasks that require strong generalization ability, such as few-shot learning and zero-shot learning.
>     - For few-shot learning, the table below shows UniTS clearly outperforms the best performing single-task model iTransformer, on forecasting, classification, imputation, and anomaly detection tasks. (Full comparisons with other methods are shown in Tables 3,4,5,23,24.)
> | Method/Best count | Forecasting  (9 datasets) | Classification  (6 datasets) | Imputation  (6 datasets) | Anomaly detection  (5 datasets) |
> | :---- | :---- | :---- | :---- | :---- |
> | iTransformer | 0/9 | 1/6 | 0/6 | 0/5 |
> | UniTS | **9/9** | **5/6** | **6/6** | **4/5** |
>     - For zero-shot learning, the table below shows UniTS considerably surpasses LLMTime, a model designed for zero-shot forecasting using LLM, across most of the tested datasets, demonstrating superior performance in handling different forecasting lengths and variable numbers.  For example, UniTS achieves a considerable improvement in MSE over LLMTime (0.030 vs.0.265) on Solar. Remarkably, UniTS exhibits an inference speed approximately 10e6 times faster than LLMTime. We also show in Figure 3 of Figure PDF (Figure 3 of the manuscript, page 8\) that UniTS can generalize to new forecasting lengths not seen during training.
> | Solar dataset | MSE | Infer. Time (seconds) |  | 5 datasets (Full results in Table 21) | Best count | Var. number | Pred length |
> | :---- | :---- | :---- | :---- | :---- | :---- | :---- | :---- |
> | LLM-Time | 0.265 | 2.0e3 | - | LLM-Time | 1/5 | 1 to 767 | 16 to 128 |
> | UniTS | **0.030** | **6.8e−3** | - | UniTS | **4/5** | 1 to 767 | 16 to 128 |

---

> ### Author Response · Authors · 2024-08-07
> **Response to Reviewer bSMF Part II (Part II of II)**
>
> # Response to Reviewer bSMF Part II (Part II of II)
>
> ### Q1.1: Are experiment settings of Sec.5.1 and Sec.5.2 the same?
>
> The settings in Sec. 5.1 represents a single-task setting, following existing works (e.g., \[CLR2024itrans, ICLR2023patch, ICLR2023times\]), where the model is trained separately on each dataset with unique training settings for each dataset. The settings in Sec. 5.2 represents a multi-task learning setting, where one model is co-trained on all datasets.
>
> ### Q1.2: Does GPT4TS also use the same fully supervised multi-task training as UniTS?
>
> For the single-task setting, the results of GPT4TS are obtained using the training setup outlined in the GPT4TS paper. In the multi-task learning setting, since GPT4TS doesn’t inherently support multi-task learning, we added data and task-specific heads to GPT4TS and employed the same fully supervised multi-task training approach as UniTS. We have updated the implementation details in the appendix to include this. Compared to GPT4TS, UniTS offers the following advantages: 1\) Unlike GPT4TS, which is built on the GPT-2 model trained with large-scale data (such as the internet-scale Pile dataset), UniTS is not trained on additional datasets beyond those described in our papers. 2\) GPT4TS is 48 times larger than UniTS (164.5M vs. 3.4M parameters) in terms of model scale. Despite the significant differences in data volume and model scale, UniTS still performs favorably compared to GPT4TS. On forecasting tasks, UniTS-SUP even outperforms GPT4TS by 2.2% (0.439 vs. 0.449 MSE).

---

> ### Comment · Reviewer_bSMF · 2024-08-10
>
> Thank you for the responses. Considering other reviewers' comments and Author Rebuttal as well, I have updated my score.

---

### Official Review · Reviewer_EsEH · 2024-07-10

**Soundness:** 3
**Presentation:** 2
**Contribution:** 2
**Rating:** 6
**Confidence:** 3

**Summary:**

The authors propose a unified model trained over multiple datasets to solve multiple tasks such as forecasting, imputation, anomaly detection, and classification. In the paper, the authors demonstrate their model abilities through extensive empirical results comparing with a large variety of baseline methods and suggesting new benchmark to compare multi-task results.

**Strengths:**

- The paper focuses on a foundational model for time series, which is unified for multiple datasets and tasks. The authors propose novel architecture and method to overcome the challenges that come with the problem.

- Although it is not the main focus of the paper, the proposed method achieves state-of-the-art results for training per-task per-dataset via UniTS-ST.

**Weaknesses:**

- The details for comparing with previous baseline methods are missing, and thus it is hard to assess how well the model performs in the unified set-up in contrast to previous methods. (see questions)


- Although most parts of the paper are clearly written and described, it is hard to follow the prompt learning part of the methods’ methodology. What exactly are the prompt tokens? The authors perform an extensive ablation study and include the number of prompts as part of it but do not describe or give an example of such prompts. (Is it just weights?)


- The empirical evidence lacks comparison to similar methods that employ unified training or prompt-based approaches, as in [1,2,3]. Although the author’s unified multi-task model is indeed interesting, it still remains unclear whether the given approach is at least comparable to other unified forecasting models that use similar benchmarks such as unified training, few-shot, zero-shot, etc.


[1]  TEMPO: Prompt-based Generative Pre-trained Transformer for Time Series Forecasting

[2]  UniTime: A Language-Empowered Unified Model for Cross-Domain Time Series Forecasting

[3]  Time-LLM: Time Series Forecasting by Reprogramming Large Language Models

**Questions:**

- Eq. 2 is slightly confusing, $\textbf{z}_c$ is defined as an input to the model but at the same time is used for the final class choice. Isn’t it should be the output of the model compared to each $\textbf{z}_e(i)$?


- In Eq. 4, $W_{\text{Interp}}$ should be defined as $l_{\text{out}} \times l_{\text{in}}$ instead of $l_{\text{in}} \times l_{\text{out}}$, otherwise the DyLinear operations does not compile in terms of dimensions.

- The implementation details of the baseline methods should be extended. Specifically, it is hard to understand what exactly was done to train the models for the new benchmark compared to UniTS-PM and UniTS-SUP. As it does not support unified training, in the appendix part, the authors mention: “For multi-task learning settings, models are configured with 3 blocks…” “The input and output heads of baseline methods are duplicated for each task to create data/task-specific heads tailored for each data source and task.” How is it different from the original set-up of those baseline methods? what do the authors mean by models configured with 3 blocks?
- How do you perform zero-shot? Now that you do not have the corresponding prompt tokens, what do you feed to the network instead of those tokens?
- The bold and underline in Table 2 for the ILI row seem to be incorrect as GPT4TS outperforms the given results.

**Limitations:**

See the Weaknesses and Questions sections.

---

> ### Author Rebuttal · Authors · 2024-08-07
>
> # Response to Reviewer EsEH Part I (Part I of II)
> Thank you for your valuable feedback\! We appreciate your recognition of our novel architecture and methods, as well as the state-of-the-art results achieved in our work. We have carefully addressed each of your questions, expanded on implementation details, and provided additional explanations following your suggestions.
>
> We hope our response addresses your concerns and kindly ask you to consider raising your score. If you still have reservations after reviewing our responses, we would greatly appreciate further guidance on how we can improve our work to achieve a better score in your review. Thank you again\!
>
> ---
>
> ###  W1 and Q3: Implementation details of the baseline methods are missing.
>
> We have revised the implementation details of the baseline methods for better understanding, here are the explanations per your two concerns:
>
> * **Make baselines support unified training**: Unlike UniTS, which can handle diverse data and tasks within a single model, baseline methods cannot be directly used for unified training because: 1\) To accommodate data with varying numbers of variables, baseline methods typically use a data-specific input head to project features from the variable count to a fixed number of embedding dimensions. 2\) Similarly, to manage different tasks, such as classification with various classes and forecasting with different lengths, baseline methods employ task-specific output heads to transform the features into the appropriate task outputs.
>   Since baseline methods are designed for single-task training, in their original setting, data/task-specific heads are used for each data and task. To make baseline methods support unified training, we add separate input heads to project data into a shared embedding space and separate output heads to convert the shared model output into task-specific outputs. However, using separate input and output heads makes it hard to generalize to new datasets and tasks.
> * **Meaning of 3 blocks**: Existing networks and UniTS are built by stacking multiple basic blocks, e.g. transformer blocks and our designed block. “Configured with 3 blocks” means the network is stacked with 3 blocks. For fair comparisons, we stack “3 blocks” for all methods.
>
> ###  W2: Explanation of prompt learning methodology.
>
> The prompt tokens are learnable embedding weights with the shape $p \times v \times d$, where $p$ is the number of prompt tokens, $v$ is the number of variables in the current data sample, and $d$ is the number of embedding channels. Each dataset has its own unique group of prompt tokens. For each sample in the dataset, these prompt tokens are appended to the tokenized sample and sent to the network to provide context information about the current sample. We have revised the manuscript to clearly describe the prompt tokens.
>
> ###  W3: Comparison with similar methods such as TEMPO, UniTime, and Time-LLM.
>
> As we discussed in the related work of the manuscript (page 3), the notable works you mentioned that utilize unified training or prompt-based approaches rely on LLMs pre-trained with extensive textual training data. We have compared the forecasting performance of these LLM-based methods, such as TEMPO, UniTime, and Time-LLM, in the table below (full results in Table 29 of the manuscript, page 32). Remarkably, even without utilizing the additional textual training data employed by other methods, UniTS demonstrates superior performance across most datasets. Additionally, unlike these methods that focus solely on forecasting, UniTS supports both generative tasks (e.g., forecasting, imputation, anomaly detection) and predictive tasks (e.g., classification). We have updated the manuscript to include a discussion of these notable works.
>
> |  | UniTS | TEMPO | UniTime | TIME-LLM |
> | :---- | :---- | :---- | :---- | :---- |
> | Additional data necessary beyond the training set | **No** | Yes | Yes | Yes |
> | Multi-task support | **Yes** | No | No | No |
> | ETTm1 | 0.337/0.376 | 0.501/0.458 | 0.385/0.399 | 0.329/0.372 |
> | ETTm2 | 0.254/0.315 | 0.281/0.328 | 0.293/0.334 | 0.251/0.314 |
> | ETTh1 | 0.405/0.426 | 0.428/0.427 | 0.442/0.448 | 0.408/0.424 |
> | ETTh2 | 0.331/0.387 | 0.361/0.398 | 0.378/0.403 | 0.334/0.383 |
> | ECL | 0.156/0.253 | 0.216/0.308 | 0.216/0.305 | 0.159/0.253 |
> | Traffic | 0.409/0.278 | 0.503/0.358 | \- | 0.388/0.264 |
> | Weather | 0.216/0.259 | 0.282/0.316 | 0.253/0.276 | 0.226/0.258 |
>
> ###  Q1: Explanation of Eq. 2.
>
> In Eq. 2, $z_c$ represents the input to the model, while in Eq. 3,  $z_c$ represents the output of the model. Since the feature shapes remain unchanged after being processed by each block in UniTS, we initially omitted the layer index for simplicity. For better clarity and understanding, we have now revised the manuscript to include the layer index.
>
> ###   Q2: Revise Eq.4,
>
> Thank you for the reminder. We have addressed this in the latest revision.
>
> ###  Q4: How do you perform zero-shot?
>
> We conducted two zero-shot experiments where the data or tasks were not encountered during training: 1\) zero-shot multi-task learning on forecasting tasks, and 2\) direct multi-step forecasting to varying time lengths.
>
> * For zero-shot forecasting on new datasets, we ask the model to make predictions on these new datasets. During pre-training, we use shared prompt tokens and the GEN token across all datasets for generative tasks. This allows the model to use the shared prompt and GEN tokens to perform predictions on new data.
> * For direct multi-step forecasting to varying time lengths, we have the model forecast for time lengths not seen during training. Thanks to our unified task tokenization, we can easily predict new lengths by concatenating a new number of GEN tokens as the model input.

---

> ### Author Response · Authors · 2024-08-07
> **Response to Reviewer EsEH Part II (Part II of II)**
>
> # Response to Reviewer EsEH Part II (Part II of II)
>
> ###  Q5: Incorrect bold and underline in Table 2.
>
> Thank you for pointing this out. For fair comparisons, we mark GPT4TS with gray and exclude it from model ranking for the following reasons: 1\) GPT4TS is based on the GPT-2, which is pre-trained with large-scale extra-textual data  (such as the internet-scale Pile dataset). In contrast, UniTS is not trained on additional datasets beyond those described in our papers. 2\) GPT4TS is 48 times larger than UniTS (164.5M vs. 3.4M parameters) in terms of model scale.
>
> Despite the significant differences in data volume and model scale, UniTS still performs favorably compared to GPT4TS. On forecasting tasks, UniTS-SUP even outperforms GPT4TS by 2.2% (0.439 vs. 0.449 MSE).

---

> ### Comment · Reviewer_EsEH · 2024-08-10
>
> Thank you for the responses, clarifications, and additional results. Considering other reviewers' comments and the author's rebuttal, I have updated my score.

---

### Official Review · Reviewer_YrSc · 2024-07-11

**Soundness:** 3
**Presentation:** 2
**Contribution:** 2
**Rating:** 5
**Confidence:** 4

**Summary:**

This paper proposes UNITS, a multi-task time series model that handles multiple predictive and generative tasks within a single model. UNITS uses the mask modeling pre-training framework. To handle multiple downstream tasks, two new sets of tokens are concatenated with data tokens: 1)prompt tokens indicating the task corresponding to each dataset and are used for future prompt learning; 2)task tokens to generate generative/discriminative outputs. A modified transformer block is also proposed. Experiments of single-task and multi-task learning on forecasting, anomaly detection, imputation, and classification demonstrate the effectiveness of the proposed model.

**Strengths:**

1. The issue studied in this work, i.e. using one shared model to model data from various domains and deal with multiple downstream tasks is of great importance.

2. The experiments conducted in this paper are comprehensive, including single-task, multi-task and few-shot learning, etc.

**Weaknesses:**

1. The writing needs improvement. 1) Section 4.3 requires a more detailed description with formal equations. Actually, it is hard to get the role of three kinds of tokens in pre-training; 2) Experiment setting in Section 5 also requires re-organization to clarify what experiments are conducted and which datasets are used for which tasks. Moreover, several datasets are generated from the same dataset with different sliding windows and thus should be addressed in the paper.

2. The modified model structure is orthogonal to the tokens and training methods proposed in the paper, and they are not well-coupled. For instance, training can still be conducted using the original Transformer.

3. Compared with traditional single-task single-dataset supervised training models (Table 25), the proposed multi-datasets multi-task pre-training model (Table 2) does not show a better performance. As training a simple model like DLinear for each dataset and task is efficient, this calls into question whether joint training across multiple domains of data can provide real benefits.

4. Some details of some experimental setups are not rigorous: for example in Table 4 (few-shot learning for imputation), datasets ECL, ETTh1 and Weather are generated from the same datasets used for pre-training, which the model can get access to, so these settings can not be viewed as few-shot learning.

**Questions:**

1. In Equation 5, what is the role of task tokens $z_m$? Compared with the left side, the right side encourages $H_{cls}(z_{pred}) = z_p$. How can it be discriminative enough for classification?

2. How is $\lambda_i$ in line 270 be determined?

3. Please compare UNITS trained by multi-task learning with that trained by single-task learning to confirm the necessity of unified multi-task training.

**Limitations:**

No potential negative societal impact.

---

> ### Author Rebuttal · Authors · 2024-08-07
>
> # Response to Reviewer YrSc Part I (Part I of III)
>
> Thank you for your valuable feedback\! We appreciate your recognition of the importance of multi-task learning on time series and the comprehensive experimental results in our work. We have carefully addressed each of your concerns, clarified our descriptions of the methodology and analyses, and added experiments based on your suggestions.
>
> We hope our response answers all your questions, and we kindly ask you to consider raising your score. If, after reviewing our responses, you still have concerns about our work, please let us know, and we will work on addressing any additional questions to achieve a better score in your review. Thank you again\!
>
> ---
>
> ###  W1: Writing needs improvement.
>
> Thank you for the suggestions. We have made the following improvements to make the paper easier to read. We are also committed to further refining the writing in the camera-ready version of the paper to make it easy to follow.
>
> ###  W1.1: More detailed description with formal equations in Section 4.3.
>
> In Section 4.3 and Equation 5, we have explained the roles of prompt and task tokens during pre-training. For each time-series sample, a handful of sample tokens get masked and replaced with GEN tokens. This masked tokenized representation of the input time-series sample is then concatenated with prompt tokens and CLS tokens and fed into the UniTS model. The UniTS model involves three task towers used during pre-training:
>
> - The GEN tower processes the prompt tokens, GEN tokens, and time-series sample tokens from the model output. The GEN tokens are used to predict the masked sequences.
> - The CLS tower processes the prompt tokens, CLS tokens, GEN tokens, and time-series sample tokens from the model output, where the CLS token is returned from the CLS tower.
> - Another GEN tower processes the GEN tokens and sequence tokens from the model output, along with the CLS token from the CLS tower output. The GEN tokens are again used to predict masked time-series samples.
>
> ###  W1.2: Experiment setting in Section 5\.
>
> Due to the limited space in the paper, we have included detailed experiment setups in Table 7, 8, 9, 10, 11 (page 19),  Section B (page 18\) and Section D (page 22\) of the appendix to explain the experimental settings and datasets used in experiments. Following your suggestions, we have further revised the implementation details in the revision to clarify the experiment settings.
>
> ###  W1.3: Several datasets are generated from the same dataset with different sliding windows.
>
> For subsets of a dataset such as ETTh1, we start by splitting the data into training and testing sets based on distinct time intervals of a long time series sequence. Within these training and testing intervals, we generate samples using various sliding windows, ensuring that there is no data leakage between the training and testing sets. We have included an explanation in the appendix detailing how various sub-datasets are generated using different sliding windows.
>
>
> ###  W2: The modified model structure is orthogonal to the tokens and training methods. Training can be conducted with the original transformer.
>
> Thank you for this question. Our response has two parts:
>
> **First**, the UniTS model architecture is strongly coupled with our multi-task co-training strategy in the following sense:
>
> - The unified tokenization strategy in the UniTS model integrates predictive and generative tasks, making it sufficiently flexible to support the co-training of various types of tasks in a single model without substantial task-specific model components.
> - The Variable/Time MHSA separately manages features with varying numbers of variables and time slots, enabling the model to co-train with data from various domains.
> - The Dynamic FFN adapts its weight shapes to capture shared temporal representations across different data types, whereas the original transformer cannot capture this with the point-wise FFN.
> - Gate module modulates feature scales to reduce interference in the latent representation space caused by multi-domain and multi-task datasets, which is critical to multi-task learning.  Traditional transformers do not account for data and task interference.
>
> We carried out an ablation study of the UniTS model architecture below to show the effectiveness of each of the above-mentioned components in the UniTS model (Full results in Tables 15, 16, 17, 18, 19, and 20 of the appendix).
>
> |  | Acc↑ (Classification) | MSE↓ (Forecasting) |
> | :---- | :---- | :---- |
> | UniTS | 81.6 | 0.439 |
> | UniTS w/o Time MHSA | 80.7  | 0.449 |
> | UniTS w/o Variable MHSA | 80.8 | 0.444 |
> | UniTS w/o Dynamic FFN  | 80.8 | 0.465 |
> | UniTS w/o Gate module  | 81.1  | 0.459 |
>
> **Second**, following your suggestion, we added new experiments comparing the original Transformer with UniTS. The unified tokenization and co-training strategy were applied to both models. Results shown in the table below suggest that simply using a transformer structure is insufficient for strong multi-task performance on time series datasets.
>
> |  | Acc↑ (Classification) | MSE↓ (Forecasting) | MAE↓ (Forecasting) |
> | :---- | :---- | :---- | :---- |
> | Transformer-network | 80.2% | 0.468 | 0.397 |
> | **UniTS-network** | **81.6%** | **0.439** | **0.381** |

---

> ### Author Response · Authors · 2024-08-07
> **Response to Reviewer YrSc Part II (Part II of III)**
>
> # Response to Reviewer YrSc Part II (Part II of III)
>
> ###  W3: The performance gain of the multi-datasets multi-task pre-training model over single-task single-dataset supervised training models. A DLinear for each dataset and task is efficient. Whether joint training across multiple domains of data can provide real benefits?
>
> Thank you for your great question\! The advantages of multi-task learning in UniTS over single-task learning include: 1\) enhanced performance without the need for task-specific hyperparameter tuning, and 2\) improved generalization capabilities, such as few-shot and zero-shot learning. The detailed evidence is presented in the following three parts:
> * **For UniTS, multi-task learning outperforms single-task learning under the same hyperparameters:** Following common practice\[ICLR2024itrans, ICLR2023patch, ICLR2023times\], the single-task results shown in Table 25 are obtained by training on each dataset with separately tuned hyperparameters. To ensure fair comparisons, we conduct a new experiment to train the single-task models using the same hyperparameters as the multi-task co-training. As shown in the table below, multi-task learning achieves stronger performance on both forecasting and classification tasks. Interestingly, under the same hyperparameters, some classification models fail to converge in the single-task setting, whereas the multi-task model does not have this issue, demonstrating the robustness of multi-task training.
> | UniTS | Acc↑ (Classification) | MSE↓ (Forecasting) |
> | :---- | :---- | :---- |
> | **Multi-task** (same hyperparameters) | 81.6% | 0.439 |
> | **Single-task** (same hyperparameters) | 65.3% | 0.464 |
> * **Multi-task UniTS outperforms hyperparameters-tuned single-task DLinear:** We compare multi-task performance of UniTS with single-task performance of DLinear, as summarized in the table below (full results in Tables 1 and 2). Despite DLinear being able to optimize and select dataset-specific hyperparameters for each dataset, UniTS, co-trained in a multi-task setting, consistently outperforms DLinear. Specifically, across 32 experiments (rows in the table below) across 16 datasets, we find that UniTS outperforms DLinear in 27 out of 32 experiments, showing that joint training across data domains is an effective approach.
> | Dataset | UniTS (Multi-task; MSE/MAE) | DLinear (Single-task; MSE/MAE) |
> | :---- | :---- | :---- |
> | ECL_P96 | **0.157/0.258** | 0.197/0.282 |
> | ECL_P192 | **0.173/0.272** | 0.196/0.285 |
> | ECL_P336 | **0.185/0.284** | 0.209/0.301 |
> | ECL_P720 | **0.219/0.314** | 0.245/0.333  |
> | Traffic_P96 | **0.465/0.298** | 0.650/0.396  |
> | Traffic_P192 | **0.484/0.306** | 0.598/0.370  |
> | Traffic_P336 | **0.494/0.312** | 0.605/0.373 |
> | Traffic_P720 | **0.534/0.335** | 0.645/0.394 |
> | ETTH1_P96 | 0.390/0.411 | **0.386/0.400** |
> | ETTH1_P192 | **0.432**/0.438 | 0.437/**0.432** |
> | ETTH1_P336 | **0.480**0.460 | 0.481/**0.459** |
> | ETTH1_P720 | 0.542/**0.508** | **0.519**/0.516  |
> | Weather_P96 | **0.157/0.206** | 0.196/0.255  |
> | Weather_P192 | **0.208/0.251** | 0.237/0.296 |
> | Weather_P336 | **0.264/0.291** | 0.283/0.335 |
> | Weather_P720 | **0.344/0.344** | 0.345/0.381  |
> | Multi-task training regime | **Yes** | No |
> | Best Count | **27/32** | 5/32 |
> * **Multi-task UniTS introduces new abilities beyond the single-task model**: Using a single-task model is insufficient to handle tasks that require strong generalization ability, such as few-shot learning and zero-shot learning.
>   - For few-shot learning, the table below shows UniTS clearly outperforms the best-performing single-task model iTransformer, on forecasting, classification, imputation, and anomaly detection tasks. (Full comparisons with other methods are shown in Tables 3,4,5,23,24.)
> | Method/Best count | Forecast  (9 sets) | Classification  (6 sets) | Imputation  (6 sets) | Anomaly detection  (5 sets) |
> | :---- | :---- | :---- | :---- | :---- |
> | iTrans | 0/9 | 1/6 | 0/6 | 0/5 |
> | UniTS | **9/9** | **5/6** | **6/6** | **4/5** |
>    - For zero-shot learning, the table below shows UniTS considerably surpasses LLMTime, a model designed for zero-shot forecasting using LLM, across most of the tested datasets, demonstrating superior performance in handling different forecasting lengths and variable numbers.  For example, UniTS achieves a considerable improvement in MSE over LLMTime (0.030 vs.0.265) on Solar. Remarkably, UniTS exhibits an inference speed approximately 10e6 times faster than LLMTime. We also show in Figure 3 of Figure PDF (Figure 3 of the manuscript, page 8\) that UniTS can generalize to new forecasting lengths not seen during training.
> | Solar dataset | MSE | Infer. Time (seconds) |  | 5 datasets (Full results in Table 21) | Best count | Var. number | Pred length |
> | :---- | :---- | :---- | :---- | :---- | :---- | :---- | :---- |
> | LLM-Time | 0.265 | 2.0e3 | - | LLM-Time | 1/5 | 1 to 767 | 16 to 128 |
> | UniTS | **0.030** | **6.8e−3** | - | UniTS | **4/5** | 1 to 767 | 16 to 128 |

---

> ### Author Response · Authors · 2024-08-07
> **Response to Reviewer YrSc Part III (Part III of III)**
>
> # Response to Reviewer YrSc Part III (Part III of III)
>
> ###  W4: Some datasets in few-shot learning for imputation are generated from the same datasets used for pre-training, thus these settings can not be viewed as few-shot learning.
>
> Thank you for pointing this out. We have revised the term "few-shot learning for imputation" to "downstream imputation task with few fine-tuning examples." The reason for using “few-shot learning” in the initial version is that 1\)  The imputation task is not seen during pretraining, making it a new task that requires a few fine-tuning examples. 2\) Following established practice in recent studies, e.g., \[ICLR2023times\], we use four datasets for imputation, in which the ETTm1 dataset is not used during pre-training.
>
> ###  Q1: Clarification of Equation 5.
>
> We have clarified Equation 5 in the revision for better understanding.
>
> * Role of task token $z_m$ (GEN token): Tokens of a full input time series sample $x$ are randomly masked, i.e., replaced with GEN tokens $z_m$, resulting in the sample token $z_x$ as shown in Equation 5\.
> * For the right part of the loss, instead of using information from the prompt token to assist with mask reconstruction, we use the feature $H_{CLS}(z_{Pred})$ to enforce the mask reconstruction. This approach trains the CLS token and $H_{CLS}$ head to learn semantic information related to classification. As demonstrated in the table below (with full results in Table 17), removing the right part of the loss results in a significant drop in classification performance (from 78.0% to 33.1%), indicating that this part of the loss is crucial for maintaining discriminative power in classification.
>
> | UniTS | Unified pretraining | Remove the right part of the loss | Remove the left part of the loss |
> | :---- | :---- | :---- | :---- |
> | Acc ↑ | 78.0 |  33.1 | 76.8 |
>
> ###  Q2: How to set $𝜆_𝑖$?
>
> In this work, we set $𝜆_𝑖$ to 1 for simplicity. In future research, we will explore more effective methods for determining $𝜆_𝑖$, as the contribution of different datasets may vary.
>
> ###  Q3: Compare UniTS trained by multi-task learning with that trained by single-task learning.
>
> We conduct new experiments to show that multi-task learning with UniTS outperforms single-task learning under the same hyperparameters. Following established protocols \[ICLR2024itrans, ICLR2023patch, ICLR2023times\], results for single-task models shown in Table 25 are obtained by training on each dataset with separately tuned hyperparameters, meaning that hyperparameters are optimally tuned for each dataset. To ensure fair comparisons, we train the single-task models using the same hyperparameters as the multi-task co-training. As shown in the table below, multi-task learning achieves stronger performance on both forecasting and classification tasks. Interestingly, under the same hyperparameters, some classification models fail to converge in the single-task setting, whereas the multi-task model does not have this issue, demonstrating the robustness of multi-task training.
>
> | UniTS | Acc↑ (Classification) | MSE↓ (Forecasting) |
> | :---- | :---- | :---- |
> | **Multi-task** (same hyperparameters) | 81.6% | 0.439 |
> | **Single-task** (same hyperparameters) | 65.3% | 0.464 |

---

> ### Author Response · Authors · 2024-08-09
> **Follow-up on reponses**
>
> Dear Reviewer YrSc,
>
> We sincerely appreciate your valuable comments! We have provided detailed responses to each of your concerns and made the necessary revisions. We kindly ask if you are satisfied with our revisions and if you have any further concerns or feedback. We would be more than happy to continue the discussion and address any additional points you may have.
>
> Thank you for your time and consideration.
>
> Best regards,
>
> The Authors

---

> > ### Comment · Reviewer_YrSc · 2024-08-10
> >
> > Thanks for your response. The rebuttal addressed my concern about Multi-task vs. Single-task learning. I have raised my score to 5.
> >
> > However, I do suggest authors improve the representation in the final version, including: 1) giving detailed descriptions of the pre-training process, better to provide a visualization to show the role of two towers in pre-training; 2) reorganizing the experiment section to clarify what experiments are conducted in a more structured way.

---

### Official Review · Reviewer_h6vS · 2024-07-13

**Soundness:** 3
**Presentation:** 3
**Contribution:** 3
**Rating:** 6
**Confidence:** 4

**Summary:**

This work aims to present a pre-trained foundation model for time series. They proposed a model called UNITS, that performs multi-task learning (both generative and discriminative) on time series datasets. Specifically, they embed a transformer with prompt tokens and task tokens to perform prompt tuning or few-shot learning. The proposed method is pre-trained on multi-domain datasets, and showed transferability across multiple downstream datasets with diverse task specifications and domains. The authors performed experimental results that show UNITS performs favorably, or comparably against many state-of-the-art models, and can be used in few-shot and prompt tuning ways.

**Strengths:**

- Overall, the work has strong experimental results.
- Great engineering efforts to perform cross-dataset training on multiple tasks.

**Weaknesses:**

Major concern 1: Overall, the contributions from the paper are overclaimed.
- The authors claim "generalizability to new tasks", yet (1) the proposed model is pre-trained on all datasets that are later experimented on; (2) for all available tasks, the authors perform prompt tuning on an aggregate dataset-centric loss value that includes all tasks that are later considered. If the model does not see a new dataset or new tasks during test time, why claim generalizability to new tasks?

Major concern 2: Lack of analytical or theoretical understanding of how multi-dataset training benefits performance.
- Lack of ablation experiments. For example: How much does the amount of pre-training datasets help (in terms of training steps or amount of datasets)? More systematic evaluation of the interrelationship between pre-training tasks (beyond Table 17)? How does the performance vary when the number of prompt tuning parameters change?
- Lack of better understanding of how to handle varying channel-wise relationships. One of the biggest challenges in building foundational model for time series is that while temporal dynamics may be generalizable, the channel-wise relationships differ significantly for each dataset. How does the proposed method address this issue through pre-training, or does it just see all possible channel-wise relationships, that are going to appear at test time, during the pre-training process?
- Lack of understanding of the learned tokens. How do the sample tokens capture the dynamics of the data, can they be visualized or categorized? How do the task tokens or prompt tokens differ from other task tokens or prompt tokens after training beyond similarity comparison? Does the similarity comparison show the intrinsic dynamical properties of the datasets?

Minor concern 1: In experimental results, the baseline numbers seem to be slightly lower than that are reported in other papers.

Minor concern 2: Lack of methodological novelty. Due to the lack of ablations, it is challenging for me to dissect which parts of the proposed model are critical to the results. The multi-task parts seem to be the work that tries to show that the more datasets one uses to train their model, the better the performance will be, as long as one uses the right combination of existing methods.

Minor concern 3: Can the authors compare the number of prompt tuning parameters against other baseline models?

**Questions:**

See above.

**Limitations:**

Can be more comprehensive.

---

> ### Author Rebuttal · Authors · 2024-08-07
>
> # Response to Reviewer h6vS Part I (Part I of IV)
>
> Thank you for your detailed comments and valuable feedback. We appreciate your recognition of experimental results in our work. We have carefully addressed each of your concerns and have added new experiments and analyses based on your suggestions.
>
> We hope that our response and new experiments address your concerns, and we kindly ask that you consider raising your score. If, after reviewing our responses, you still have concerns about our work, we would greatly appreciate further guidance on how we can improve it to achieve a better score in your review. Thank you again\!
>
> ---
>
> ### Major concern 1.1: Generalizability to new tasks: the proposed model is pre-trained on all datasets that are later experimented on.
>
> We'd like to clarify this is a misunderstanding. We are confident and have confirmed that there is no information leakage between the training set in pre-training and the testing set in fine-tuning. Furthermore, we consider the generalization to new tasks and generalization to new datasets that have never been seen during pre-training. Moveover, we even consider the generalization to new data domains (e.g., traffic, weather-\>spacecraft, web). The details of these generalization settings are in the following:
>
> * **Few-shot learning on new data and tasks**: 1\) The datasets for few-shot classification and forecasting tasks are not used during pre-training, demonstrating generalization to new datasets (Tables 3, 24, Page 8, 29). 2\) For few-shot imputation, the imputation task is not performed during pre-training, demonstrating generalization to new tasks (Tables 4, 23, Page 9, 28). The ETTm1 dataset is not used for pre-training, ensuring generalization to new data and new tasks (Tables 23, Page 28). 3\) For anomaly detection, both the task and datasets are not used during pre-training, ensuring generalization to new data and new tasks (Tables 5, Page 9).
> * **Direct multi-step forecasting to varying time lengths**: UniTS generalizes to new forecasting lengths not seen during training, which demonstrates task generalization (Figure 3 of Figure PDF).
> * **Zero-shot forecasting**: UniTS conducts zero-shot forecasting on new datasets that are only seen during testing (Tables 21, Page 26).
> * **Generalization to new domains:** UniTS is evaluated not only on tasks and datasets that were never encountered during pre-training or fine-tuning, but our experiments also consider generalization to new domains. These include the domains of server machine data (SMD, PSM), spacecraft (MSL, SMAP), and infrastructure (SWaT) in anomaly detection tasks (Table 11, Page 20), as well as the domains of healthcare (Hospital) and web data (Web Traffic) for zero-shot forecasting (Table 9, Page 20), none of which are seen during pre-training.
>
> ### Major concern 1.2: Why claim generalizability to new tasks?
>
> UniTS has two levels of generalizability, 1\) **Zero-shot learning** where the model is presented with new datasets or new tasks at test time that it has not encountered during training. 2\) **Few-shot/prompt learning** where a pre-trained model is adapted to a new task via fine-tuning using only a small part of a specific dataset.
>
> * For zero-shot learning, we benchmark against LLMTime, a model designed for zero-shot forecasting using LLM. Table below shows UniTS considerably surpasses LLMTime across most of the tested datasets, demonstrating superior performance in handling different forecasting lengths and variable numbers.  For example, UniTS achieves a considerable improvement in MSE over LLMTime (0.030 vs.0.265) on Solar. Remarkably, UniTS exhibits an inference speed approximately 10e6 times faster than LLMTime. We also show in Figure 3 of Figure PDF (Figure 3 of the manuscript, page 8\) that UniTS can generalize to new forecasting lengths not seen during training.
> | Solar dataset | MSE | Infer. Time (seconds) |  | 5 datasets (Full results in Table 21) | Best count | Var. number | Pred length |
> | :---- | :---- | :---- | :---- | :---- | :---- | :---- | :---- |
> | LLM-Time | 0.265 | 2.0e3 | - | LLM-Time | 1/5 | 1 to 767 | 16 to 128 |
> | UniTS | **0.030** | **6.8e−3** | - | UniTS | **4/5** | 1 to 767 | 16 to 128 |
>
> * For few-shot learning, the table below shows UniTS clearly outperforms the best performing baselines iTransformer on forecasting, classification, imputation, and anomaly detection tasks. (Full comparison with other methods are shown in Tables 3,4,5,23,24.)
> | Method/Best count | Forecast  (9 datasets) | Classification  (6 datasets) | Imputation  (6 datasets) | Anomaly detection  (5 datasets) |
> | :---- | :---- | :---- | :---- | :---- |
> | iTrans. | 0/9 | 1/6 | 0/6 | 0/5 |
> | UniTS | **9/9** | **5/6** | **6/6** | **4/5** |

---

> ### Author Response · Authors · 2024-08-07
> **Response to Reviewer h6vS Part II (Part II of IV)**
>
> # Response to Reviewer h6vS Part II (Part II of IV)
>
> ### Major concern 2.1: Lack of ablation experiments of pre-training.
>
> Thank you for your great suggestion\! Following your suggestion, we conduct new experiments of pre-training UniTS by varying the size of the pre-training dataset and the amount of training epochs.
>
> - **Pre-training with different numbers of epochs**: As demonstrated in the table below, increasing the number of pre-training epochs improves performance on both forecasting and classification tasks.
>
>   *New experiments of UniTS under different pre-training epochs, average performance on 20 forecasting and 18 classification are reported.*
>    | Pre-training steps | 1 epoch | 3 epochs | 5 epochs | 8 epochs | 10 epochs |
>    | :---- | :---- | :---- | :---- | :---- | :---- |
>    | Acc↑  (Cls.) | 75.1 | 76.8 | 78.2 | 77.0 | 79.0 |
>    | MSE↓ (Fore.) | 0.493 | 0.479 | 0.484 | 0.473 | 0.460 |
>    | MAE↓ (Fore.) | 0.410 | 0.391 | 0.389 | 0.386 | 0.383 |
>
> - **Pre-training with different data sizes**: Similarly, increasing the size of pre-training dataset improves performance on both forecasting and classification tasks, as shown in the table below.
>
>    *New experiments of UniTS under different pre-training epochs, average performance on 20 forecasting and 18 classification are reported.*
>   | Pre-training data size | 10% of the total training set | 30% | 50% | 80% | 100% |
>   | :---- | :---- | :---- | :---- | :---- | :---- |
>   | Acc↑  (Cls.) | 74.2 | 76.3 | 77.6 | 78.8 | 79.0 |
>   | MSE↓ (Fore.) | 0.502 | 0.462 | 0.483 | 0.465 | 0.460 |
>   | MAE↓ (Fore.) | 0.417 | 0.385 | 0.391 | 0.384 | 0.383 |
>
> ### Major concern 2.2: More systematic evaluation of the interrelationship between pre-training tasks.
>
> We have added three new experiments to systematically evaluate interrelationship and interdependencies between pre-training tasks:
>
> * **Results on scaling pre-training data sizes and training steps**: We examine the impact of considering different sizes of pre-training datasets and the amount of model training in our response to the major concern 2.1.
> * **Results on cross-task pre-training**: We evaluate the effect of cross-task pre-training by pre-training a model using our pre-training strategy on either generative tasks (forecasting) or predictive tasks (classification). The table below shows that UniTS, pre-trained solely on forecasting datasets, achieves similar performance to the model pre-trained on both forecasting and classification data. Despite not encountering any classification datasets during pre-training, it still performs well on classification tasks. When the model is pre-trained exclusively on classification datasets, performance on both classification and forecasting tasks drops significantly compared to the model pre-trained on both types of data. Given that the data amount of forecasting datasets is larger than classification datasets (22920 vs. 5022 iterations per epoch), this suggests that the larger amount of data plays a more crucial role in pre-training effectiveness than the data type.
>
>   *New experiment on cross-task pre-training evaluation on UniTS, average performance on 20 forecasting and 18 classification tasks are reported.*
>   | Pre-training data type | Acc↑ (Cls.) | MSE↓ (Fore.) | MAE↓ (Fore.) |
>   | :---- | :---- | :---- | :---- |
>   | 20 forecasting datasets | 78.5 | 0.454 | 0.379 |
>   | 18 classification datasets | 74.1 | 0.583 | 0.807 |
>   | Full 38 datasets | 79.0 | 0.460 | 0.383 |
>
> * **Results on cross-domain pre-training:** We evaluate the effect of cross-domain data pre-training, where the model is pre-trained on either Weather-domain datasets or Traffic-domain datasets. In the table below, compared to joint pre-training on both domains, the performance decreases with single-domain pre-training, where pre-training is conducted solely on the downstream dataset's domain, showing the advantage of joint pretraining. For instance, the MSE on Weather datasets goes from 0.253 to 0.259. Compared to single-domain pre-training, cross-domain pre-training leads to larger performance drops, e.g., pre-training on Traffic datasets and then evaluating on Weather datasets results in an MSE increase from 0.259 to 0.289. Interestingly, pre-training on Weather datasets achieves better performance across both domains, suggesting that data from certain domains might be more beneficial for pre-training.
>
>   *New experiment on cross-domain pre-training evaluation on UniTS, average performance on 4 Weather or Traffic dataset domains are reported.*
>   | Evaluation data➡️ | Weather datasets (4 sets) | Traffic datasets (4 sets) |
>   | :---- | :---- | :---- |
>   | Pre-training data⬇️ | MSE↓/MAE↓ (Fore.) | MSE↓/MAE↓ (Fore.) |
>   | Weather domain (4 datasets) | 0.259/0.287 | 1.338/0.768 |
>   | Traffic domain (4 datasets) | 0.289/0.314 | 0.680/0.438 |
>   | Weather \+ Traffic domains (8 sets) | 0.253/0.282 | 0.511/0.320 |

---

> ### Author Response · Authors · 2024-08-07
> **Response to Reviewer h6vS Part III (Part III of IV)**
>
> # Response to Reviewer h6vS Part III (Part III of IV)
>
> ### Major concern 2.3 How does the performance vary when the number of prompt tuning parameters change?
>
> We have performed an ablation study on the number of prompt tokens in table below (Table 15 of the manuscript, page 24). We find that larger numbers of prompt tokens lead to better performance, and this performance gains levels off with the increased number of prompt tokens. However, the performance remains consistently high across different token counts. For instance, using 10 versus 5 tokens results in only a \+0.1 increase in accuracy and a \-0.016 decrease in MSE, indicating that the results are robust to variations in this hyperparameter.
> | Prompt token (Count) | Acc↑ (Classification) | MSE↓ (Forecasting) |
> | :---- | :---- | :---- |
> | 0 | 81.0  | 0.460 |
> | 5 | 81.5 | 0.455 |
> | 10 | **81.6** | **0.439** |
>
> ### Major concern 2.4: How to handle varying channel-wise relationships?
>
> Handling the varying channel-wise (variable-wise) relationships is a key issue we addressed in designing UniTS. We tackle this challenge by designing UniTS such that it can handle arbitrary numbers of variables. UniTS specifies a unified masked reconstruction pre-training task to ensure the model can capture channel/variable-wise relationships without assuming a fixed number of variables/channels across all pre-training/fine-tuning datasets.
>
> At the architecture level:
>
> * The input to the network is tokenized to features that have independent channel and variable dimensions.
> * The UniTS network is designed to handle a varying number of variables. For example, we use the Variable MHSA to get the relation among variables without the need to use linear layers with a fixed number of channels to learn relation among variables.
> * Prompt tokens in UniTS are variable-dependent, enabling the network to adapt to data with varying numbers of variables seamlessly.
>
> At the model pre-training level:
>
> * Based on the unified network that handles a varying number of variables, our pre-training approach is designed to be independent of the number of variables. It employs a unified masked reconstruction pre-training task to randomly mask input tokens. The model is forced to learn general and shared representations of the relationships among variables to reconstruct missing tokens. This ensures that the model can capture the essential relational dynamics without being tied to a specific number of variables.
> * The prompt tokens for datasets are also pretrained, making the prompt tokens learn the conditions that can prompt the network to capture data-specific variable-wise relationships during inference.
>
> ### Major concern 2.5:  How do the sample tokens capture the dynamics of the data, can they be visualized or categorized?
>
> In UniTS, the sample tokens are extracted from the input sample to capture data dynamics using time/variable MHSA, DynamicFFN, and Gate modules. These tokens are then transformed into sample sequences via the GEN tower, allowing visualization as time series sequences. Additionally, the CLS token enables categorization by transferring sample tokens to the classification classes in the CLS tower.
>
> ### Major concern 2.6: How do tokens differ from other tokens after training beyond similarity comparison?
>
> To compare the difference among tokens before and after training, beyond similarity comparison, we add new UMAP plots generated with the prompt tokens before and after training, in Figure 1 and Figure 2 of Figure PDF file. Before training, the prompt tokens from all datasets are dispersed. In contrast, the UMAP of prompt tokens after training reveals that tokens from the same datasets are clustered. However, some tokens from different datasets remain closely positioned, indicating that data from different domains share similar information.
>
> ### Major concern 2.7: Does similarity comparison show the intrinsic dynamical properties of the datasets?
>
> We categorize the datasets and visualize the similarity among learned prompt tokens for datasets in Figure 6 of the manuscript (page 27). Datasets within the same class, for instance, FaceDetection and SelfRegulationSCP2, which both consist of EEG data, demonstrate a higher similarity. While some out-of-domain datasets still exhibit strong similarities, indicating that they share certain similar requirements.

---

> ### Author Response · Authors · 2024-08-07
> **Response to Reviewer h6vS Part IV (Part IV of IV)**
>
> # Response to Reviewer h6vS Part IV (Part IV of IV)
>
> ### Minor concern 1: Slightly lower baseline numbers.
>
> We have two settings: a single-task setting and a multi-task setting. For the single-task setting, we report results from previous works to ensure fair comparisons. For the multi-task setting, we reimplement the baseline to support multi-task capabilities. The multi-task setting is intrinsically more challenging than the single-task setting, so the results for baseline methods are lower than those for the single-task setting.
>
> ### Minor concern 2: Methodological novelty and ablations. How does multi-task work?
>
> We show in Table 2 that UniTS outperforms existing methods in the multi-task setting, and it’s not a combination of existing methods. Instead, we propose new designs to handle multi-task learning as follows:
>
> * The unified tokenization strategy unifies the predictive and generative tasks, which is the first time that two types of tasks have been unified in one model, as far as we know.
> * The Variable/Time MHSA separately handles the features with varying numbers of variables and time slots, enabling one model to handle different datasets, while existing works have to define data-specific input heads to support each dataset.
> * The Dynamic FFN adapts the weight shape to capture the shared temporal representations of different data.
> * The Gate module modulates the feature scales to mitigate interference in the latent representation space caused by multi-domain and multi-task datasets.
>
> Due to the limited space in the manuscript, we show the ablations in Tables 15, 16, 17, 18, 19, and 20 of the appendix. From these tables, we summarize the ablation results of key modules in UniTS in the following table. For classification, both Time MSHA, Variable MSHA, and Dynamic FFN play similar roles in performance, indicating that time/variable/channel level features are all important to the time series classification. For forecasting, the Dynamic FFN and Gate module plays more important roles than the Time/Variable MHSA. Still, Time/Variable MHSA is important to forecasting performance.
>
> |  | Acc↑ (Classification) | MSE↓ (Forecasting) |
> | :---- | :---- | :---- |
> | UniTS | 81.6 | 0.439 |
> | UniTS w/o Time MHSA | 80.7  | 0.449 |
> | UniTS w/o Variable MHSA | 80.8 | 0.444 |
> | UniTS w/o Dynamic FFN  | 80.8 | 0.465 |
> | UniTS w/o Gate module  | 81.1  | 0.459 |
>
> ### Minor concern 3: Compare the number of prompt tuning parameters against other baseline models.
> The table below compares the prompt tuning parameters of UniTS against strong baseline methods. The tokens used for prompt tuning in UniTS are only about 1.4% of parameters used in the LLM-based GPT4TS. Additionally, compared to pure time series models like TimesNet and PatchTST, the parameters for prompt tuning in UniTS are still smaller.
>
> | Model | Prompt-tuning parameters in UniTS | GPT4TS | TimesNet | PatchTST |
> | :---- | :---- | :---- | :---- | :---- |
> | Trainable parameters | 2.4M | 164.5M | 5.0M | 3.9M |

---

> ### Author Response · Authors · 2024-08-09
> **Follow up on rebuttal**
>
> Dear Reviewer h6vS,
>
> We sincerely appreciate your insightful comments and the time you have taken to review our work. In response, we have carefully addressed each of your concerns and made the necessary revisions. We kindly ask if these revisions meet your expectations and if there are any further concerns or feedback you would like to discuss. We are more than happy to continue the conversation and address any additional points you may have.
>
> Thank you once again for your time and thoughtful consideration.
>
> Best regards,
> The Authors

---

> ### Author Response · Authors · 2024-08-12
> **Reminder of our new response to Reviewer h6vS's concerns.**
>
> Thank you for your valuable feedback!
> We've provided a detailed response to your new comments. In case the OpenReview system didn’t notify you, we wanted to bring it to your attention here as well.
>
> If you find our response unsatisfactory, please don't hesitate to share your concerns with us. We are more than happy to continue the discussion.
>
> Thank you for your time and consideration.
>
> Best regards,
>
> The Authors

---

### Author Rebuttal · Authors · 2024-08-07

Thank you to all the reviewers for thoughtful and insightful feedback\! We appreciate Reviewers acknowledging our contributions. Reviewers emphasized the importance of the issue studied, noting that **“using one shared model to model data from various domains and deal with multiple downstream tasks is of great importance”** \[YrSc\]. Reviewers recognize the novelty of our approach, stating that the **“authors propose novel architecture and methods to overcome the challenges that come with the problem”** \[EsEH\]. Reviewers highlight that our work **“has strong experimental results”** \[h6vS\] and appreciate the comprehensive nature of our experiments, which include **“single-task, multi-task, and few-shot learning”** \[YrSc\]. Furthermore, reviewers commend our **“great engineering efforts to perform cross-dataset training on multiple tasks”** \[h6vS\]. Additionally, reviewers acknowledge the achievement of **“state-of-the-art performance on various time series tasks”** while being adaptable to new tasks through **“parameter-efficient prompt learning”** \[bSMF\]. Even though it is not the primary focus, our method achieves **“state-of-the-art results for training per-task per-dataset via UniTS-ST”** \[EsEH\]. They also praise the clarity and readability of the paper, noting that it is **“well-written and easy to follow”** \[bSMF\].


Reviewers also raised several key points, which we address in our point-by-point responses to each reviewer. Here, in the general response, we briefly highlight new experiments added in this rebuttal and address a few important points raised by reviewers.

In response to reviewers’ comments, we conducted seven new groups of experiments to achieve the following:

* Study the pre-training strategy of UniTS with varying amounts of data and training steps.
* Systematically evaluate the interrelationship between pre-training tasks, including cross-task and cross-domain pre-training.
* Analyze the learned tokens by generating new UMAP plots of prompt tokens before and after training.
* Compare prompt tuning parameters of UniTS against strong baseline methods.
* Compare UniTS architecture to original Transformer architecture.
* Compare single-task models with multi-task models using the same hyperparameters.
* Compare UniTS with similar methods, such as UniTime.

These new experiments and analyses, together with initial results across 38 datasets (spanning human activity sensors, healthcare, engineering, and finance domains), 12 forecasting models, 20 classification models, 18 anomaly detection models, and 16 imputation models, have strengthened our confidence in the effectiveness of UniTS. We address key points raised by the reviewers:

* We demonstrate the advantages of multi-task learning over single-task learning, particularly in terms of generalizability to new tasks through zero-shot and few-shot learning \[h6vS, YrSc, bSMF\].
* We provide detailed explanations and show the effectiveness of the proposed unified tokenization and UniTS network structure \[h6vS, YrSc, EsEH\].
* We clarify the implementation details and experimental settings of our method and baselines, and we add more experiments to offer a thorough analysis of UniTS \[YrSc, EsEH, YrSc, bSMF\].

### General Notes to Reviewers:
We abbreviate weaknesses by “W\#”  and questions by “Q\#.” We are not allowed to update the revised manuscript during the rebuttal phase, but we assure the reviewers that comments have been incorporated into our revisions. Figure PDF refers to the one-page PDF we upload to the open review system.

---

*If you feel we have not sufficiently addressed your concerns to motivate increasing your score, we would love to hear from you further on what points of concern remain and how we can improve the work in your eyes. Thank you again\!*

---

### Comment · Reviewer_h6vS · 2024-08-12
**Excessive parameters trained during tuning & channel-wise relationship**

Thank you for the careful response. While the response did clarify some of my questions, I think it also further highlighted my doubts about the work. I want to highlight two points below:

1. UNITS trains **2.4M** parameters during the prompt tuning stage, while the near-SOTA task-specific model TimesNet only has **3.9M** parameters in total. The new parameters, by themselves, are sufficient to train a brand-new model to achieve decent performance. It would be unfair to compare against huge models that attempt to engineer LLMs into TS models because their objective is to adapt a very large model to be useful in this case.
2. Being adaptive to new channel-wise relationships is fundamentally different from being generalizable to new channel-wise relationships. I understand that the model could be used on data with varying amounts of channels, and many works have different ways to achieve that (e.g. PatchTST). However, as the model claims itself to be a foundation model for time series that learns tasks in a zero-shot way, being merely adaptive is not sufficient. The authors have to provide the fundamental logic of using the pre-trained model on a completely unseen channel-wise relationship. Otherwise, it is an overclaim.

Overall, I like the engineering perspective of the work, yet I found the overclaiming components very concerning, and the limitations of the proposed approach are not very well addressed/discussed in the paper. I'd give a much higher score if the paper does not overclaim or lead the readers to assume beyond what is told by experimental results. Given my current understanding I'd adjust my score from 6 to 5.

---

> ### Author Response · Authors · 2024-08-12
> **New response to Reviewer h6vS**
>
> # New response to Reviewer h6vS Part I (Part I of II)
> Thank you for pointing out these two important questions\! We totally understand your concerns and we are clarifying your concerns about the parameters in UniTS and we will tune down our claim about the zero-shot ability of UniTS.  We will further add these points you mentioned in the limitation section to prompt future work to explore the unified time series model. However, we would also like to clarify a misunderstanding on the first point that we respond to below.
>
> If you think our response is still unsatisfactory, please share your concerns with us, we are more than happy to have further discussion with you.
>
> ---
>
> ### Parameters of UniTS
>
> * Thank you for your comment, but we would like to kindly point out a misunderstanding. **The total number of parameters in UniTS is 3.4M**, including 1.0M for the backbone network and 2.4M for the tokens, **which is smaller than the 3.9M parameters in TimesNet.** **Again, 3.4M is the TOTAL number of parameters in UNiTS, not only the number of trainable parameters in the prompt tuning stage** ***(PLEASE SEE THE 5TH COLUMN IN THE TABLE BELOW)*.**
> * **The number of tokens can be adjusted to balance performance and parameter count.** We use 10 prompt tokens for each dataset, but as shown in table below (Table 14, page 24), reducing the number of prompt tokens or even eliminating them can significantly decrease the parameter count while still outperforming TimesNet. For example, reducing the number of prompt tokens from 10 to 5 decreases the token parameters to 1.5M with only a minor performance drop, such as a 0.1% drop in accuracy. **Even without prompt tokens, UniTS, with 0.7M token parameters and 1.7M total parameters, still outperforms TimesNet, which has 3.9M parameters (with only 43% of TimesNet's parameters), by a wide margin in forecasting tasks**, e.g., 0.460 vs. 0.525 in MSE.
> | Method | Prompt token Num. | All Token parameters | Model network parameters | Total number of parameters (tokens+network) | Acc ↑ | MSE ↓ | MAE↓ |
> | :---- | :---- | :---- | :---- | :---- | :---- | :---- | :---- |
> | TimesNet | \- | \- | 3.9M | 3.9M | 80.9 | 0.525  | 0.412  |
> | **UniTS** | **0** | **0.7M** | **1.0M** | **1.7M** | **81.0** | **0.460** | **0.391** |
> | UniTS | 5 | 1.5M | 1.0M | 2.5M | 81.5 | 0.455 | 0.387  |
> | UniTS | 10 | 2.4M | 1.0M | 3.4M | 81.6 | 0.439 | 0.381 |
>
> * **We want to clarify that UniTS is not based on LLMs; instead, it is trained exclusively on time series data.** As shown in Table 29 (page 32), UniTS demonstrates superior performance across most datasets, even without utilizing the GPT-2 model (1,500M parameters), which is trained with additional textual data and used by other LLM-based methods. Furthermore, unlike these methods that focus solely on forecasting, UniTS supports both generative tasks (e.g., forecasting, imputation, anomaly detection) and predictive tasks (e.g., classification).
> |  | UniTS | TEMPO | UniTime | TIME-LLM |
> | :---- | :---- | :---- | :---- | :---- |
> | Additional data necessary beyond the training set | **No** | Yes | Yes | Yes |
> | Multi-task support | **Yes** | No | No | No |
> | ETTm1 | 0.337/0.376 | 0.501/0.458 | 0.385/0.399 | 0.329/0.372 |
> | ETTm2 | 0.254/0.315 | 0.281/0.328 | 0.293/0.334 | 0.251/0.314 |
> | ETTh1 | 0.405/0.426 | 0.428/0.427 | 0.442/0.448 | 0.408/0.424 |
> | ETTh2 | 0.331/0.387 | 0.361/0.398 | 0.378/0.403 | 0.334/0.383 |
> | ECL | 0.156/0.253 | 0.216/0.308 | 0.216/0.305 | 0.159/0.253 |
> | Traffic | 0.409/0.278 | 0.503/0.358 | \- | 0.388/0.264 |
> | Weather | 0.216/0.259 | 0.282/0.316 | 0.253/0.276 | 0.226/0.258 |
>
> **Please see more content in PART II**

---

> ### Author Response · Authors · 2024-08-12
> **New response to Reviewer h6vS Part II**
>
> # New response to Reviewer h6vS Part II (Part II of II)
> ### Claim about zero-shot ability
> We fully understand your concerns about adapting to new channel-wise relationships rather than generalizing, i.e. few-shot versus zero-shot learning. We agree that zero-shot learning is significantly more challenging than few-shot learning. **Our work primarily focuses on few-shot learning, with some initial exploration of zero-shot learning. We will emphasize this distinction and moderate our claims regarding zero-shot capabilities to avoid any overclaim.** Below, we provide a detailed explanation of our initial attempts in zero-shot learning, our work in few-shot learning, and the difference between UniTS and PatchTST.
>
> * **For initial attempts in zero-shot learning**, the table below shows UniTS considerably surpasses LLMTime, a model designed for zero-shot forecasting using LLM, across most of the tested datasets, demonstrating superior performance in handling unseen channel-wise relationship, different forecasting lengths, and different variable numbers. Note that the testing datasets are never seen during pre-training.  For example, UniTS achieves a considerable improvement in MSE over LLMTime (0.030 vs.0.265) on Solar. Remarkably, UniTS exhibits an inference speed approximately 10e6 times faster than LLMTime. We also show in Figure 3 of Figure PDF (Figure 3 of the manuscript, page 8\) that UniTS can generalize to new forecasting lengths not seen during training. We agree that these attempts are only initial explorations of using the pre-trained model for zero-shot learning, and we will stress this in our manuscript.
> | Solar dataset | MSE | Infer. Time (seconds) |  | 5 datasets (Full results in Table 21\) | Best count | Var. number | Pred length |
> | :---- | :---- | :---- | :---- | :---- | :---- | :---- | :---- |
> | LLM-Time | 0.265 | 2.0e3 | \- | LLM-Time | 1/5 | 1 to 767 | 16 to 128 |
> | UniTS | **0.030** | **6.8e−3** | \- | UniTS | **4/5** | 1 to 767 | 16 to 128 |
>
> * **For few-shot learning that UniTS mainly focuses on**, the table below shows UniTS clearly outperforms the best-performing single-task model iTransformer, on forecasting, classification, imputation, and anomaly detection tasks. (Full comparisons with other methods are shown in Tables 3,4,5,23,24.)
> | Method/Best count | Forecast  (9 datasets) | Classification  (6 datasets) | Imputation  (6 datasets) | Anomaly detection  (5 datasets) |
> | :---- | :---- | :---- | :---- | :---- |
> | iTrans | 0/9 | 1/6 | 0/6 | 0/5 |
> | UniTS | **9/9** | **5/6** | **6/6** | **4/5** |
>
> * The difference between our work and PatchTST in handling varying amounts of channels is that PatchTST doesn’t consider the inter-relationship among channels, treating all channels independently, while we design a new architecture and pretraining scheme to enable the model to learn these inter-relationships, which can generalize to new data, as we responded to you in Major Concern 2.4.
>
> ### Overclaiming on the UniTS components
> Thank you for highlighting this important point. We would like to clarify that we do not claim UniTS to be a foundational model for time series that learns tasks. The word "foundation" or "foundational" appears 10 times in the main paper and supplementary materials. Five instances are in the Reference list, and the remaining five instances occur in the background sections of the abstract, introduction, and related work. We are careful not to claim that UniTS is a foundational model, and we have tried to carefully pace our claims and support them by experimental results (e.g., UniTS is a multi-task model unifying predictive and generative tasks, as reflected in our evaluations).
>
> ---
>
> That said, understanding your excellent comment, we will additionally adjust our claims regarding the zero-shot capability of UniTS. We will also incorporate the points you raised into the limitations section to guide future work that would bring us closer towards a foundation model for time series.
>
> If you find our response unsatisfactory, please don't hesitate to share your concerns with us. We are more than happy to continue the discussion. Given this new understanding and clarification on misunderstanding wrt point 1, we kindly ask that you reconsider the adjustment of your initial score.

---

> > ### Comment · Reviewer_h6vS · 2024-08-12
> > **Thanks**
> >
> > Thanks for your reply. It is a detailed plan for tuning down the claims. Please make sure they are included in the revision.

---

### Author Response · Authors · 2024-08-12
**Thank you!**

Dear Reviewers,

We sincerely appreciate your thoughtful feedback and the time you invested in reviewing our paper! We will carefully revise the manuscript based on your valuable comments.

Thank you again for your time and dedication!

Best regards,

The Authors

---

### Decision · Program_Chairs · 2024-09-25

**Decision:**

Accept (poster)

**Comment:**

This paper proposes a unified pre-trained model for time series data called UNITS, designed to handle multiple predictive and generative tasks across different datasets. The model employs a transformer architecture enhanced with prompt and task tokens, enabling it to perform prompt tuning and few-shot learning. UNITS is pre-trained on multiple datasets from various domains and is evaluated on its transferability across different downstream tasks, demonstrating comparable or superior performance to state-of-the-art models.

**Strengths:**
- Strong experimental results, showing effective cross-dataset training on multiple tasks.
- The issue of handling diverse domains and multiple tasks within a single model is of great importance.
- UNITS achieves state-of-the-art results in several tasks and can adapt to new tasks via prompt learning.
- The paper is generally well-written and easy to follow.

**Weaknesses:**
- Overclaimed contributions, particularly regarding the generalizability to new tasks that were seen during pre-training.
- Lack of theoretical understanding and ablation studies on the impact of multi-dataset training.
- The model’s novelty is questioned due to insufficient methodological explanations and the orthogonality of the model's structure to the proposed tokens and training methods.
- Issues with writing clarity, especially in explaining the roles of the tokens and experimental setups.
- Missing comparisons with similar models using unified training or prompt-based approaches.

Some concerns have been addressed by the authors during the rebuttal period.

Reviewers generally lean positive for this paper. One of the reviewers had concerns about some of the overclaim, downgrading the rating from 6 to 5, but the concern seemed later addressed by the authors, and the reviewer adjusted the rating back to 6. Overall, I agree with the reviewers that this is a good contribution to the community and therefore recommend acceptance.